# Understanding terrestrial water storage variations in northern latitudes across scales

Tina Trautmann[1,2], Sujan Koirala[1], Nuno Carvalhais[1,3], Annette Eicker[4], Manfred Fink[5], Christoph Niemann[1,5], Martin Jung[1]

[1]Department of Biogeochemical Integration, Max-Planck-Institute for Biogeochemistry, Jena, 07745, Germany
[2]International Max Planck Research School for Global Biogeochemical Cycles, Jena, 07745, Germany
[3]CENSE, Departamento de Ciências e Engenharia do Ambiente, Faculdade de Ciências e Tecnologia, Universidade Nova de Lisboa, Caparica, 2829-516, Portugal
[4]HafenCity University, Hamburg, 20457, Germany
[5]Department of Geography, Friedrich-Schiller University, Jena, 07743, Germany

*Correspondence to*: Tina Trautmann (ttraut@bgc-jena.mpg.de)

**Abstract.** The GRACE satellites provide signals of total terrestrial water storage (TWS) variations over large spatial domains at seasonal to inter-annual time scales. While the GRACE data have been extensively and successfully used to assess spatio-temporal changes in TWS, little effort has been made to quantify the relative contributions of snow pack, soil moisture and other components to the integrated TWS signal across northern latitudes, which is essential to gain a better insight into the underlying hydrological processes. Therefore, this study aims to assess which storage component dominates the spatio-temporal patterns of TWS variations in the humid regions of northern mid-to-high latitudes.

To do so, we constrained a rather parsimonious hydrological model with multiple state-of-the-art Earth observation products including GRACE TWS anomalies, estimates of snow water equivalent, evapotranspiration fluxes, and gridded runoff estimates. The optimized model demonstrates good agreement with observed hydrological spatio-temporal patterns and was used to assess the relative contributions of solid (snow pack) versus liquid (soil moisture, retained water) storage components to total TWS variations. In particular, we analysed whether the same storage component dominates TWS variations at seasonal and inter-annual temporal scales, and whether the dominating component is consistent across small to large spatial scales.

Consistent with previous studies, we show that snow dynamics control seasonal TWS variations across all spatial scales in the northern mid-to-high latitudes. In contrast, we find that inter-annual variations of TWS are dominated by liquid water storages at all spatial scales. The relative contribution of snow to inter-annual TWS variations, though, increases when the spatial domain over which the storages are averaged becomes larger. This is due to a stronger spatial coherence of snow dynamics, that are mainly driven by temperature, as opposed to spatially more heterogeneous liquid water anomalies, that cancel out when averaged over a larger spatial domain. The findings first highlight the effectiveness of our model-data fusion approach that jointly interprets multiple Earth observation data streams with a simple model. Secondly, they reveal that the determinants of TWS variations in snow-affected northern latitudes are scale dependent. In particular, they seem to be not merely driven by snow variability, but rather are determined by liquid water storages on inter-annual time scales. We

conclude that inferred driving mechanisms of TWS cannot simply be transferred from one scale to another, which is of particular relevance for understanding the short and long-term variability of water resources.

## 1 Introduction

Since the start of the mission in 2002, measurements from the Gravity Recovery and Climate Experiment (GRACE) provide unprecedent estimates of changes in the terrestrial water storage (TWS) across large spatial domains (Tapley et al., 2004;Wahr et al., 2004). Due to its global coverage and independence from surface conditions, the data represents a unique opportunity to quantify spatio-temporal variations of the Earth's water resources (Alkama et al., 2010;Werth et al., 2009). Therefore, GRACE data has widely been used to diagnose patterns of hydrological variability (Seo et al., 2010a;Rodell et al., 2009;Ramillien et al., 2006;Feng et al., 2013), to validate and improve model simulations (Doll et al., 2014;Güntner, 2008;Werth and Güntner, 2010;Chen et al., 2017;Eicker et al., 2014;Girotto et al., 2016;Schellekens et al., 2017), and to enhance our understanding of the water cycle on regional to global scales (Syed et al., 2009;Felfelani et al., 2017).

Despite the high potential of GRACE data for hydrological applications (Döll et al., 2015;Werth et al., 2009), the measured signal vertically integrates over all water storages on and within the land surface, which challenges the interpretation of the driving mechanism behind TWS variations. To facilitate insight into the underlying processes, hydrological models are frequently used to separate the measured TWS into its different components such as groundwater, soil moisture, and snow pack (Felfelani et al., 2017). However, as a consequence of uncertain model structure, forcing and parametrization, model-based partitioning is ambiguous (Güntner, 2008), and may lead to diverging conclusions especially on regional scale (Long et al., 2015;Schellekens et al., 2017).

While the uncertainties of catchment-scale hydrological models are commonly reduced by calibrating the model parameters against discharge measurements, the majority of macro-scale models relies on a priori parametrization. So far, only few models used to assess hydrological processes on continental to global scales are constrained by observations, and if so, they are mainly calibrated against observed discharge of large river basins (Long et al., 2015;Döll et al., 2015). Recently, several studies showed the benefits of additionally including GRACE TWS data in model calibration (Werth and Güntner, 2010;Xie et al., 2012;Chen et al., 2017) or by means of data assimilation (Eicker et al., 2014;Forman et al., 2012;Kumar et al., 2016). However, although these approaches improve model simulations, they do not reduce the uncertainty in partitioning of TWS due to the parameter equifinality problem (Güntner, 2008). Therefore, it is desirable to include multiple observations, ideally of several hydrological storages and fluxes, to constrain model results (Syed et al., 2009).

Nowadays, the increasing number and quality of Earth Observation based products provides valuable information on a variety of hydrological variables over large scales, and thus facilitates constraining model simulations with multiple data streams simultaneously. While this can provide a more robust understanding of how variations of water storages translate into the observed TWS (Werth and Güntner, 2010), it is very challenging in practice and has rarely been implemented.

On the one hand, this is due to the limitations and inherent uncertainties of each Earth Observation based product that need to be considered when comparing simulations and observations. For example, satellite-based soil moisture retrievals only capture the upper 5 cm of soil under snow-free conditions and therefore are difficult to compare to modelled soil water (Lettenmaier et al., 2015), while large scale observations of snow mass based on passive microwave sensors are known to suffer from uncertainties in deep and wet snow conditions (Niu et al., 2007) and multispectral sensors solely provide estimates of snow cover in the absence of clouds (Lettenmaier et al., 2015).

Besides, the application of multi-criteria calibration approaches is limited by the increasing complexity of most macro-scale hydrological models over time (Döll et al., 2015). This high model complexity is not only associated with conceptual issues related to overparameterization (Jakeman and Hornberger, 1993) and large computational demand, but also has shown to not necessarily improve model performance (Orth et al., 2015). Therefore, it is desirable to implement a rather parsimonious model structure (Sorooshian et al., 1993), especially in multi-criteria model-data fusion approaches.

Applying multiple observational constraints is in particular beneficial in regions, where hydrological dynamics are poorly understood and thus their representation in models varies widely. This is the case for snow-dominated regions as the northern high-latitudes (Schellekens et al., 2017), which are among the areas most prone to the impacts of climate change (Tallaksen et al., 2015). These regions have been experiencing the strongest surface warming over the last century globally (IPCC, 2014), a trend which is expected to exacerbate in the future and to significantly change hydrological patterns (AMAP, 2017). Therefore, solid understanding of present hydrological processes and variations is crucial, yet the effect of complex snow dynamics on other storages and water resources is relatively unknown (van den Hurk et al., 2016;Kug et al., 2015). While it has been shown that snow mass is the primary component of seasonal variations of TWS in large northern basins (Niu et al., 2007;Rangelova et al., 2007), it is not known what drives the TWS variations on inter-annual or longer time scales in these regions. Moreover, most analysis so far focus on individual river basins and do not provide a comprehensive picture over large spatial scale.

In this study, we therefore aim to investigate the contributions of snow compared to other (liquid) water reservoirs to spatio-temporal variations of TWS in the northern mid-to-high latitudes. To do so, we establish a model-data-fusion approach that integrates multiple Earth observation based data streams including GRACE TWS along with estimates of snow water equivalent (SWE), evapotranspiration and runoff into a rather simple hydrological model. This model is designed as a combination of standard model formulations yet aims to maintain low complexity in order to facilitate multi-criteria calibration and to focus on variables that can be constrained by observations.

First, we explain the applied methods including the implemented model, the used data, and the multi-criteria calibration approach. The following section presents and discusses the results obtained with the optimized model. In the results, we describe the calibrated model parameters and evaluate the model performance with respect to observed patterns of TWS and SWE. Subsequently, the relative contributions of snow and liquid water storages to TWS variations are assessed on seasonal

and inter-annual scales. Thereby we first focus on spatially integrated values across the study domain, and secondly on the composition on local grid scale. Finally, we summarize our findings and draw the conclusions.

## 2 Data and methods

The following section provides an overview on the experimental set up, followed by a more detailed description of the
model, the input data as well as the methods for model calibration and analysis.

### 2.1 Experiment design

To assess the composition of TWS variations in northern mid-to-high latitudes, we optimized a simple hydrological model on daily time steps at a 1° x 1° latitude/longitude resolution. We defined the area of interest as humid land surface north of 40° N, excluding Greenland as well as grids with > 90 % permanent snow cover, and > 50 % water fraction. Humid areas are
derived based on an aridity index AI ≥ 0.65, which was calculated as the ratio of precipitation and potential evapotranspiration (United Nations Environment, 1992). Therefore, we used the same precipitation and potential evapotranspiration data as for model forcing (see 2.3). To mask out grids with > 90 % permanent snow cover and > 50 % water fraction, we applied the SYNMAP land cover classification (Jung et al., 2006). This dataset has an original resolution of 1 km and was used to determine the fraction of land cover classes within each 1° x 1° grid cell.
Forced with global observation-based climate data, the model parameters were constrained for a subset of the study domain by multiple Earth observation data products using a multi-criteria calibration approach. These products include terrestrial water storage anomalies as seen by the GRACE satellites (Watkins et al., 2015;Wiese, 2015), measurements of snow water equivalent obtained in the GlobSnow project (Luojus et al., 2014), evapotranspiration fluxes based on FLUXCOM (Tramontana et al., 2016) and runoff estimates for Europe from E-RUN based on E-OBS (Gudmundsson and Seneviratne,
2016). Once the model parameters were calibrated, we evaluated the model against the same data, taking into account the entire study domain. Finally we applied the calibrated model to quantify the contributions of snow and liquid water storages to the integrated TWS. Thereby we considered different spatial domains (local grid cell and spatially aggregated) and temporal scales (mean seasonal and inter-annual variations).

Due to the differences in the temporal coverage of the observational data streams, model calibration and evaluation were
conducted for the period 2002–2012, while analysis of TWS components cover the whole period of 2000–2014.

An overview on the experiment design and the selected time periods is provided by Fig. 1, while the following sections give a detailed description of the individual steps.

## 2.2 Model description

We designed a conceptual hydrological model with low complexity and a total number of 10 adjustable parameters. The model considers major hydrological fluxes as snow melt, sublimation, infiltration, evapotranspiration, and (delayed) runoff, and includes water storages in the snow pack, the soil, and due to delay in runoff (Fig. 2). It is forced by precipitation (P), air temperature (T) and net radiation (Rn) and calculates all hydrological processes on daily time steps for individual grid cells. A simple schematic diagram of the model is shown in Fig. 2, while a detailed description of modelled processes is provided in S1.

In the first step, precipitation P is partitioned into liquid precipitation (rain fall) and snow fall based on a temperature threshold of 0° C. Accumulating snow fall increases the snow pack represented by the snow water equivalent SWE [mm], which depletes by sublimation and melt if T exceeds 0 °C. We calculate sublimation based on the GLEAM model (Miralles et al., 2011c), and apply an extended day-degree approach to estimate snow melt (Kustas et al., 1994). Since the presence of snow can be highly variable in one grid cell, we model the fractional snow cover [-] following Balsamo et al. (2009) which is used to scale snow melt and sublimation.

Similar to the WaterGAP model (Döll et al., 2002), incoming water from rain and snow melt is allocated to soil moisture (SM) and land runoff (Qs) depending on soil moisture conditions (Bergström, 1991). SM is represented by a one-layer bucket storage that depletes by evapotranspiration (ET). We calculate ET as the minimum of demand-limited potential ET following the Priestley-Taylor formula (Priestley and Taylor, 1972b) and supply-limited ET following Teuling et al. (2006). As land runoff results from an effective soil water recharge formulation, the calculated runoff is essentially all the water that cannot be stored in the soil. Thus, it implicitly contains both, surface and subsurface runoff as well the percolation to deeper water storages such as groundwater, as well as contributions from surface water bodies. To account for runoff contributions from slow-varying storages, we calculate runoff from each grid cell (Q) by applying an exponential delay function on Qs (Orth et al., 2013). Based on mass balance, we derive the amount of retained land runoff (RW), which implicitly accounts for the effects of several water pools that are not explicitly represented in the model (groundwater, lakes, wetlands and the river storage). The sum of RW and SM is then taken as the total liquid water storage (W). Frozen soil water is not explicitly included in the model, Further, the model does not account for lateral flow of water among grid cells and does not consider river routing explicitly. While the effect of the routing can be significant in large river basins of humid regions (Kim et al., 2009), it is negligible on the spatial scale of a grid cell (as also shown by small influence of the delayed storage component), and at the temporal scale of monthly aggregated values. To ensure that the model calibration is not affected by river routing, we do not compare simulated runoff to measured river discharge of large basins in our model-data fusion approach.

Finally, the sum of liquid water storage and snow is taken as the modelled terrestrial water storage (TWSmod) of a grid cell for the given time step. Since the delayed runoff contribution is minor at the monthly time scale, we, for simplicity, only focus on the contributions of SWE and total W to TWS in this study.

## 2.3 Input Data

As meteorological forcing we used globally available, daily cumulated gridded precipitation sums [mm d$^{-1}$], average air temperature [°C] and net radiation [MJ m$^{-2}$] from March 2000 to December 2014.

Precipitation values originate from the 1° daily precipitation product version 1.2 of the Global Precipitation Climatology Project (GPCP-1DD) (Huffman et al., 2000;Huffman and Bolvin, 2013), that combines remotely-sensed precipitation and observations from gauges. Temperature was obtained from the CRUNCEP version 6.1 dataset (Viovy, 2015), which is a

merged product of Climate Research Unit (CRU) TS.3.23 observation-based monthly climatology (1901-2013) (New et al., 2000) and the National Center for Environmental Prediction (NCEP) 6-hourly reanalysis data (1948-2014) (Kalnay et al., 1996). Net radiation is based on radiation fluxes of the SYN1deg Ed3A data product of the Clouds and the Earth's Radiant Energy Systems (CERES) program of the United States' National Aeronautics and Space Administration (NASA) (Wielicki et al., 1996).

Rather than using a single data stream, e.g. discharge measurements at the outlet of large continental catchments as used in traditional large-scale hydrological studies, we calibrated the model against multiple observation-based data streams on the grid scale. The integrated datasets include terrestrial water storage anomalies (TWSobs) [mm], snow water equivalent (SWEobs) [mm], evapotranspiration (ETobs) [mm d$^{-1}$], and gridded runoff estimates for Europe (Qobs) [mm d$^{-1}$].

TWSobs is derived from the GRACE Tellus Mascon product version 2 based on the GRACE gravity fields Release 05 processed at NASA's Jet Propulsion Laboratory (JPL) (Watkins et al., 2015;Wiese, 2015). The GRACE solutions were corrected for geocentric motion coefficients, according to Swenson et al. (2008) and for variations in Earth's oblateness (C20 coefficient) obtained from Satellite Laser Ranging (Cheng et al., 2013). The Glacial isostatic adjustment has been accounted for using the model by A et al. (2013). The dataset provides monthly anomalies of equivalent water thickness relative to the

January 2004–December 2009 time-mean baseline for the period 2002–2016. Unlike previous GRACE products based on spherical harmonic coefficients, the JPL RL05M dataset uses equal area 3° x 3° spherical cap mass concentration blocks (mascons) to solve for monthly gravity field variation. To ensure a clean separation along coastlines within land/ocean mascons, a Coastline Resolution Improvement (CRI) filter has been applied (Watkins et al., 2015). For each mascon, uncertainties were estimated by scaling the formal covariance matrix. To enable hydrological analysis at sub-mascon

resolution, we used the provided gain factors to scale the original TWSobs values.

To gain confidence in partitioning of the integrated TWS, we additionally used SWE estimates from the European Space Agency's (ESA) GlobSnow SWE v2.0 product (Luojus et al., 2014). The dataset provides daily SWE values [mm] for the non-alpine Northern Hemisphere based on assimilating passive microwave satellite data and observed snow depth from

weather stations by applying a semi-empirical snow emission model. Compared to data from stand-alone remote sensing approaches, GlobSnow SWE shows superior performance, even though validation against ground based measurements still reveals a systematic underestimation of SWE under deep snow conditions due to a change in the microwave behaviour of the snow pack (Derksen et al., 2014;Takala et al., 2011;Luojus et al., 2014).

The ET product is based on FLUXCOM (www.fluxcom.org), i.e. upscaled estimates of latent energy that were derived by integrating local eddy covariance measurements of FLUXNET sites, remote sensing, and meteorological data using the Random Forest (Breiman, 2001) machine learning algorithm (Tramontana et al., 2016). In this study, we apply the Random Forest (Breiman, 2001) realization of FLUXCOM-RS+METEO (see Tramontana et al. 2016 for details). While the product captures seasonality and spatial patterns of mean annual fluxes well, predictions of inter-annual variations remain highly

uncertain (Tramontana et al., 2016). In addition, the performance of FLUXCOM ET was found to be lower in extreme environments that are not well represented by FLUXNET sites such as the arctic. An underestimation in the order of 10– 20 % of ET can be expected owing to missing energy balance correction prior to upscaling for this respective FLUXCOM ET realization. To calculate ETobs [mm d$^{-1}$], we assume a constant latent heat of vaporization of 2.45 MJ kg$^{-1}$.

Similar to TWS that represents the vertically integrated water storage, observations of river discharge spatially integrate

hydrological processes within a basin. Thus, they provide an invaluable tool for model validation at large scales. However, it is desirable to apply gridded products to evaluate model performance at local (grid) scale. Therefore, we used the observation-based gridded runoff product E-RUN version 1.1 (Gudmundsson and Seneviratne, 2016) as constraint for runoff processes. This dataset is based on observed river flow from 2771 small European catchments that was spatially disaggregated to upstream grid cells using a machine learning approach. The data provides mean monthly runoff rates per

unit area for each grid, so that river routing is not necessary to directly compare runoff estimates with modelled runoff. Similar to the ET data, gridded runoff estimates show high accuracy for the mean seasonal cycle across Europe, and poorer agreement regarding monthly time series and inter-annual variations (Gudmundsson and Seneviratne, 2016).

Table 1 summarizes the main features of the data used in this study. If required, the data streams were resampled from their

original resolution to a consistent 1° x 1° latitude/longitude grid and common daily (meteorological forcing) respectively monthly (calibration data) time steps. Data preparation further included extraction of the relevant, overlapping time period and area under consideration.

**2.4 Multi-criteria calibration**

In this study, calibration is intended to identify the set of 10 model parameters (Table 2) that achieves the best fit between

simulations and observations for all grids cells and regarding all observational data simultaneously. Thereby, we aimed to exploit the strength of each data stream, while considering known uncertainties and biases. For this purpose, we defined a cost function that takes into account the weakness of each observed variable and evaluates the overall model fit with one value of total cost (see subsequent section). To minimize total costs and thus find the optimal parameter values, we applied

the Covariance Matrix Evolution Strategy (CMAES) (Hansen and Kern, 2004) search algorithm. The CMAES, as an evolutionary algorithm, is a stochastic, derivative-free method for non-linear, non-convex optimization problems. Compared to gradient-based approaches, it performs superior on rough response surfaces with discontinuities, noise, local optima and/or outliers, and is a reliable tool even for global optimization (Hansen, 2014). Additionally, CMAES' guided search in the parameter space makes the algorithm less computationally demanding than other global optimization approaches which enumerate a large number of possible solutions (e.g. Monte Carlo Markov Chain methods) (Bayer and Finkel, 2007).

In order to keep computational demands low and to avoid overfitting by a very small sample size, we perform calibration for a subset of 1000 randomly chosen grid cells. Within this iterative process, the model simulations are carried out on daily time steps, while costs are calculated based on monthly values. Further, each model run includes an initialization based on 10 random years that were selected a priori.

**Cost function**

To objectively describe the goodness of fit, we defined a cost function based on model efficiency (Nash and Sutcliffe, 1970), but with explicit consideration of the uncertainty $\sigma_i$ of the observed data stream as:

$$\text{cost} = \frac{\sum_{i=1}^{n} \frac{(x_{obs,i} - x_{mod,i})^2}{\sigma_i}}{\sum_{i=1}^{n} \frac{(x_{obs,i} - \bar{x}_{obs})^2}{\sigma_i}} \ , \tag{1}$$

where $x_{obs,i}$ is the observed data, $\bar{x}_{obs}$ the average of $x_{obs}$, and $x_{mod,i}$ the modelled data of each space-time point $i$, respectively. Similar to model efficiency, the criterion reflects the overall fit in terms of variances and biases, yet with an optimal value of 0 and a range from 0–∞. Costs are calculated for each variable separately, considering only grid cells and time steps with available observations, which vary for the different data streams. Additionally, to overcome the sensitivity to outliers arising from data uncertainties or inconsistencies, we adopted a 5 percentile outlier removal criterion (Trischenko, 2002), i.e. the data points with the highest 5 % residuals $x_{obs} - x_{mod}$ were excluded in the cost function.

The costs of each observed variable and its modelled counterpart are then added equally to derive a single value of total cost (Eq. (2)). Since a perfect simulation would yield a total cost of 0, calibration aims to find the global minimum of $\text{cost}_{total}$.

$$\text{cost}_{total} = \text{cost}_{TWS} + \text{cost}_{SWE} + \text{cost}_{ET} + \text{cost}_{Q} \ , \tag{2}$$

As the uncertainty $\sigma$ of observational data in Eq. (1) is adapted to best reflect the strength of the individual data stream, we preselected the strongest aspect of the data to be included in the cost function. Owing to the larger uncertainties of ET$_{obs}$ and Q$_{obs}$ on inter-annual scales, we only employed the grid's mean seasonal cycles, while the full monthly time series of gridded TWS$_{obs}$ and SWE$_{obs}$ were taken into account.

As ETobs and Qobs do not explicitly provide uncertainty estimates, we assume an uncertainty of 10 % and minimal 0.1 mm, respectively. In order to define $\sigma$ of TWSobs we utilized the spatially and temporally varying uncertainty information provided with the GRACE data. Additionally, the monthly values of observed and modelled TWS datasets were translated as anomalies to a common time-mean baseline of their overlapping period 01.01.2002–31.12.2012 before calculating the cost

for TWS.

For SWE, we applied an absolute uncertainty of 35 mm based on reported differences to ground-measurements (Liu et al., 2014;Luojus et al., 2014). Since GlobSnow SWE saturates above approx. 100 mm (Luojus et al., 2014), we do not penalize model simulations when both, SWEobs and SWEmod, are larger than 100 mm in order to prevent the propagation of data biases to calibrated model parameters.

For maps of the temporal average uncertainties see S2.

**2.5 Evaluation of model performance**

Once the parameters were optimized, we applied the model for the entire study domain, and evaluated its performance regarding all grid cells (6050) in terms of Pearson correlation coefficient r and root mean square error RMSE for each variable with observational data, respectively. On the one hand, the overall performance at local scale was assessed by

calculating r and RMSE for the monthly time series of each grid individually. On the other hand, the model performance over the entire study domain was evaluated by comparing the seasonal and inter-annual dynamics of the regional average. Therefore, we defined inter-annual variation (IAV) as the deviation of the monthly values from the mean seasonal cycle (MSC). As with the calibration, we focused on the common time period 2002–2012, and considered only the grid cells and time steps with available observations.

In order to benchmark our model against current state-of-the-art hydrological models, we compared its simulations with the multi-model ensemble of the global hydrological and land surface models of the eartH2Observe dataset (Schellekens et al. 2017). This ensemble includes HTESSEL-CaMa (Balsamo et al., 2009), JULES (Best et al., 2011;Clark et al., 2011), LISFLOOD (van der Knijff et al., 2010), ORCHIDEE (Krinner et al., 2005;Ngo-Duc et al., 2007;d'Orgeval et al., 2008), SURFEX-TRIP (Alkama et al., 2010;Decharme et al., 2013), W3RA (van Dijk and Warren, 2010;van Dijk et al., 2014),

WaterGAP3 (Flörke et al., 2013;Döll et al., 2009), PCR-GLOBWB (van Beek et al., 2011;Wada et al., 2014) and SWBM (Orth et al., 2013). For consistency, we processed the model estimates in the same manner as our model simulations to directly compare modelled SWE and TWS to observations from GlobSnow and GRACE, respectively. While each model provides simulated SWE, they vary in the representation of other storage components. We calculated modelled TWS for each model by summing up the available water storage components, respectively. Thus, the variables contributing to

modelled TWS vary between the models, which impedes detailed comparison. Additionally, we calculated the multi-model mean of SWE and TWS simulations.

## 2.6 Analysis of TWS variations and composition

Finally, the contribution of snow and liquid water to seasonal and inter-annual TWS variability was quantified across spatial scales. For this, we ran the model with optimized parameters for the entire study domain from 2000 to 2014, and translated simulated storages as anomalies to the time-mean baseline. As in the model evaluation, the MSC and IAV of SWEmod, W and TWSmod anomalies were calculated at local scale for each grid individually and as spatial average over all grid cells. To assess storage variability, the variance in the MSC and the IAV of each storage component was computed. Assuming negligible covariance of snow and liquid water (see S8), their relative contribution to TWS variance was calculated as the contribution ratio CR:

$$CR = \frac{var(W)}{var(TWSmod)} - \frac{var(SWEmod)}{var(TWSmod)} \, , \tag{3}$$

While CR = 0 indicates equal contribution of snow and liquid water to TWS variability, positive (negative) values of CR imply that variations of TWSmod mainly result from variations in liquid water (snow pack), with CR = +1 meaning that all variation is explained by liquid water and CR = -1 suggests determination solely by snow.

From Eq. (3) and the assumption that $var(W) + var(SWE) = var(TWS)$, the percentage contribution of liquid water storages to the variability of TWS can be inferred as CW:

$$CW = \frac{var(W)}{var(TWSmod)} = \frac{CR+1}{2}, \tag{4}$$

As this study intends to analyse the effects of storage components on TWS at different spatial scales (local grid scale and large (regional) spatial averages), the difference in spatial heterogeneities of these components is considered. Some storage components, e.g., soil moisture anomalies, have much larger spatial variability than others. Due to this high small-scale heterogeneity, the effect on larger regional scale might be smaller than expected, as different local scale heterogeneities compensate each other when the regional averages are calculated (Jung et al., 2017). Thus, we assessed the spatial coherence of simulated patterns of SWE and W by calculating the proportion of total positive and total negative covariances among grid cells (Eq. (4,5) in Jung et al. (2017)). If the sum of positive covariances outweighs the sum of negative covariances, it implies some degree of spatial coherence of the anomalies. Spatial coherence of anomalies then causes a larger variance of the averaged anomalies compared to the sum of the variances of individual grid cells. This assessment of spatial coherence of SWE and W anomalies allows for understanding different contributions of SWE and W to TWS variability at local scale compared to the regional scale.

## 3 Results and discussion

The following sections present and discuss the results obtained with the calibrated model. First, we review the calibration approach and the optimized parameter values. Then the model is validated with respect to its overall performance at grid scale, as well as the reproduction of average seasonal (MSC) and inter-annual (IAV) dynamics. Subsequently, we assess the driving component of spatially integrated TWS variations and the relative contributions of snow and liquid water to TWS variability on local scale. Finally, we summarize the results across spatio-temporal scales.

### 3.1 Model optimization

Optimization of the model identifies the parameter values listed in Table 2 to be most suitable regarding all data constraints simultaneously. The CMAES search algorithm converged after 3272 function evaluations as no further improvement of costs$_{\text{total}}$ could be achieved, which suggests a reliable estimate of the global optimal parameter set. The individual cost terms obtained with default and optimized parameter values are contrasted in Table S1. Overall, this parameter set obtained for a subset of 1000 random grids is reasonable with respect to reported 'plausible' parameter ranges, with none of them reaching their physically and/or technically defined upper and lower calibration bounds.

In detail, snow fall is reduced by $p_{sf}$ to 67 % of precipitation occurring at T < 0 °C. This reduction agrees with Behrangi et al. (2016), who found GPCP to overestimate snowfall over Eurasian high latitudes by about 20 % compared to other precipitation products. Similar, overestimation of precipitation undercatch correction in GPCP has been reported by Swenson (2010). Taking into account the mismatch in temporal and spatial domain, as well as the experimental definitions, reducing GPCP snow fall in our study by 33 % is roughly consistent with to both studies. Therefore, $p_{sf}$ allows to reduce inconsistencies between the precipitation forcing and the water storages as given by GlobSnow SWE and GRACE TWS.

Further, each grid is assumed to be completely covered by snow if SWE ≥ 80 mm ($sn_c$). On the one hand, the snow pack can be reduced by sublimation, with $sn_a$ = 0.44 indicating relatively high sublimation resistance, compared to a default of $sn_a$ = 0.95 proposed by Miralles et al. (2011b). The divergence probably results from interaction with snow melt, as net radiation also contributes to melt with 0.9 mm MJ$^{-1}$ ($m_r$) if T exceeds 0 °C. Nevertheless, melt is mainly induced by temperature, as the estimated day degree factor ($m_t$) is 2.63 mm K$^{-1}$, which is close to typical values of 3 mm K$^{-1}$ (Müller Schmied et al., 2014;Stacke, 2011). These parameter interactions underline an equifinality issue between modelled snow melt and sublimation due to missing data constraints, resulting in larger parameter uncertainties for $sn_a$, $m_r$ and $m_t$. However, for the objective of this study it's not primarily relevant whether sublimation or radiation induced melt decreases the snow pack, as the total snow loss amount remains relatively unchanged for different parameter combinations.

The maximum soil water holding capacity is set to 515 mm after calibration, a comparatively high value that is likely to include storages in surface water bodies such as lakes and wetlands within our study domain. The optimized value of $s_{exp}$ is 1.46, which suggests a non-linear relationship between soil moisture storage and runoff generation. For the same amount of

incoming water (rain fall and snow melt), the non-linear relationship produces a smaller runoff and larger infiltration than a linear relationship ($s_{exp} = 1$).

Regarding evapotranspiration, the alpha coefficient ($et_a$) in the Priestley-Taylor formula is generally taken as 1.26 for well-watered crops based on experimental observations (Priestley and Taylor, 1972b;Eichinger et al., 1996). Thus, the optimized value of 1.20 for $et_a$ reflects a plausible value. Further, $et_{sup}$ indicates that 2 % of the available soil moisture can evaporate per day (including transpiration), which lies within the range of site-specific ET sensitivities from $0.001 - 0.5$ d$^{-1}$ and is close to the median value (5 %) (Teuling et al., 2006).

Finally, the calibrated recession time scale that delays land runoff is 13 days ($q_t$). Compared to much smaller alpine catchments for which Orth et al. (2013) reported $q_t$ of 2 days, the longer delay coefficients are reasonable at a spatial resolution of 1° x 1° grids, because the elevation gradients are much smaller within a large spatial area. At first glance, 13 days appear to be quite a short effective time period, as the delay is supposed to comprise contributions from much slower depleting reservoirs, such as lakes and deep groundwater. However, implementing and calibrating a simple groundwater storage, that is recharged with some proportion of land runoff and linearly depletes over time, led to similar retardation times.

The uncertainty in the optimized parameter vector was estimated by quantifying each parameter's standard error as the square root of the product between the diagonal elements of the parameters' covariance matrix (calculated from the Jacobian matrix) and the sum of residual squares according to Omlin and Reichert (1999) and Draper and Smith (1981). The resulting relative parameter uncertainty is particularly instructive for comparing how well individual parameters could be constrained. Most parameters were well constrained (Table 2) suggesting that our model-data fusion method, fed by multiple observation streams, succeeded in reducing the initial theoretical parameter ranges (up to 500 %) to much narrower ranges. Nonetheless, some parameters have a larger uncertainty range than others (e.g. $q_t$, $sn_c$, $m_t$), which may highlight a limitation in suitable observations to constrain them, as well as a lower sensitivity of the model results and the cost function used. Further, given that the model only considers the spatial variability of climate, the uncertainty in global parameters obtained from inversion may reflect the natural variations in these parameters that arise from differences in local land surface characteristics such as topography or land cover.

We adopted the calibrated parameter values as global constants for model simulations over the entire study domain. Even though the globally uniform parameters may not provide perfect simulation for all grids over a large study domain, this approach represents a compromise between a priori parametrization of the model and its calibration at local or regional (e.g. basin) scale. While local and regional model calibration enables good adaption to geographic characteristics, it easily leads to overfitting of the model and thus propagates the constraints' inherent errors and uncertainties to the modelling result. As these uncertainties often vary in space, globally uniform parameter values diminish overfitting uncertainties. In addition, calibration for several independent grids is computationally demanding and subsequently requires a parameter

regionalization approach (He et al., 2011). Since such approaches are not commonly accepted (Sood and Smakhtin, 2015;Bierkens et al., 2015), macro-scale models mostly apply a priori parameters based on empirical values or on expert knowledge, that yet may lead to suboptimal simulations (Beck et al., 2016;Sood and Smakhtin, 2015).

## 3.2 Model performance

For model validation, we used the optimized parameter values to simulate hydrological fluxes and states of the 2002–2012 period over the entire study domain, and evaluated the model results against the observation-based data of TWS, SWE, ET and Q.

In general, all observed patterns are reproduced very well, taking into account the specific data weaknesses. We achieve a 'near perfect' correlation of 0.99 and 0.94 for mean seasonal variations of ET and Q, respectively. The median RMSE of

mean seasonal ET is 11 mm month$^{-1}$ and 9.5 mm month$^{-1}$ for Q, which represent 15 % resp. 17 % of the average observed annual amplitude. At the inter-annual scale, though, larger discrepancies exist, which at least partly arise from larger uncertainties in ETobs resp. Qobs (S4). Thus, we assume high confidence in modelled ET and Q fluxes and subsequently focus on evaluation of the water storages TWS and SWE.

### 3.2.1 Performance on local grid scale

Overall, the model performs well compared to the observations of monthly time series of SWE and TWS (Fig. 3). More than half of the grid cells obtain correlation values higher than 0.74 between SWEobs and SWEmod. In general, the median RMSE is 20 mm, which is smaller than the average uncertainty of 35 mm in SWEobs. The correlation reduces in lower latitudes where seasonal snow accumulation and thus variability is small. Further, the correlation is also relatively weaker in arctic North America and the Rocky Mountains, while larger deviations between observed and modelled snow quantities

center around mountainous and coastal regions (e.g. Rocky Mountains, Kamchatka), and regions with the largest seasonal snow accumulation (Labrador Peninsula, North Siberian Lowland and northern West Siberian Plain). There are several reasons for this relatively poorer performance. First, the GlobSnow measurements do not cover mountainous areas due to the sub-grid variability of snow depth and high uncertainties in the microwave measurements in complex alpine terrains (Takala et al., 2011). As the resampling and the coarse resolution of each grid in this study compound a distinct alpine/non-alpine

classification, these uncertainties leak to the surrounding areas. Second, neither the input forcing data nor our model include the sub-grid scale heterogeneity of climate (e.g., precipitation and temperature) and hydrological processes, that may be significant in mountain-near or coastal regions. Additionally, the accuracy of observed large snow accumulation is limited as the radar-retrieval methods tend to saturate at large SWEobs values, which then leads to large RMSE of the model simulation.

Similar to SWE, more than half of the grid cells show a strong correlation of 0.71 between TWSobs and TWSmod, which reflects a realistic temporal variation in the model simulation. Compared to SWE, the RMSE of TWS is somewhat higher, yet the median of 43 mm still reflects the range of ± 22 mm average uncertainty in GRACE TWSobs of the study domain

(Wiese, 2015). However, when comparing GRACE TWS with model simulations, several aspects have to be considered. First, TWSobs as an integrated signal comprises all water storages, not all of which are (sufficiently) represented in the model structure. Second, although GRACE TWS passed through various pre-processing steps, the models to account e.g. for postglacial rebound or leakage between neighbouring grid cells introduce their own uncertainties and do not remove the effects completely. Further, with a native resolution of 3°, uncertainties remain for grids that comprise large variability at sub-grid scale and depend on the model used to estimate GRACE scaling factors (Wiese et al., 2016). This together is reflected in higher RMSE in arctic regions (e.g. surrounding the Hudson Bay), as well as in heterogeneous coastal and mountainous regions. Additionally, our model shows a weaker performance in subarctic and arctic wetlands, and in central North America and Eastern Eurasia. The latter both are relatively dry regions that are rather dominated by inter-annual TWS variations (Humphrey et al., 2016). Discrepancies between TWSobs and TWSmod thus relate to a low signal-to-noise ratio in TWSobs due to small seasonal TWS variations. On the other hand, the anthropogenic influence for irrigational withdrawal is very large in these regions, yet such processes are not considered in our model. We also lack explicit surface water storages (including wetland dynamics), which may be the reason for poorer performance especially in North American wetland regions.

### 3.1.2 Performance of the spatially integrated simulations

Since the aim of this study is to analyse the composition of TWS across temporal scales, we additionally evaluated average (spatially integrated) MSC and IAV of SWE and TWS (Figure 4). While the mean seasonal variations of both observational data streams are relatively robust and have been used for model evaluation before (Alkama et al., 2010;Döll et al., 2014;Schellekens et al., 2017;Zhang et al., 2017), their inter-annual variations are more uncertain and contain considerable noise. This clearly reduces the information content in the observational data, so that we evaluate the IAV in more qualitative terms.

As with the comparison at grid scale, the spatially averaged SWEmod compares well to SWEobs, with a correlation of 0.95 suggesting a good reproduction of seasonal snow accumulation and ablation processes (Fig. 4a). Owing to the high uncertainty of SWEobs peaks due to signal saturation, the higher amplitude of SWEmod seems reasonable. Although inter-annual variations are not as well represented as the MSC, general tendencies, e.g. increasing/decreasing positive/negative anomalies, coincide.

Similar to SWE, the spatial average TWS shows high correlation of 0.91 for seasonal variations, with positive anomalies from December to May/June and negative anomalies during summer and autumn months (Fig. 4b). Even though the modelled amplitude is slightly larger than the observed one, it stays within the uncertainty range of TWSobs for most months, suggesting reliable simulations. However, TWSmod precedes TWSobs on average by one month, reaching the maximum in March instead of April, and the minimum in August instead of September. A similar phase shift of one month between GRACE TWS and modelled TWS has been reported by several state-of-the-art global models (Döll et al., 2014;Schellekens et al., 2017). It should be noted that some areas such as East North America, Kamchatka, Scandinavia and

Western Europe do not show phase differences, while the lag in South East Eurasia is even larger, as already suggested by lower overall correlation (Fig. S5). In general, the disagreement in timing is attributed to the lack of sufficient water storages and delay mechanism within the model, so that the modelled system reacts too fast (Schellekens et al., 2017;Döll et al., 2014;Schmidt et al., 2008). Thus, we implemented model variants with an explicit groundwater storage to delay depletion of TWS, with spatially varying soil properties to better represent heterogeneous infiltration and runoff rates, as well as a variant that applied a more sophisticated approach to calculate snow dynamics based on energy balance. Despite the efforts, we achieved no improvement in terms of reducing the phase shift. Therefore, the question arose, whether it is not primarily the model formulation that prevents correction of the temporal delay, but rather the combination of forcing data respectively observational constraints. To further preclude possible errors due to such data inconsistencies e.g. between GRACE TWS and GlobSnow SWE, we excluded GlobSnow SWE data from calibration. Although this could slightly improve the agreement of TWS' MSC, it led to unrealistic behaviour of snow dynamics, and thus did not offer any advantages. Besides, we found no major differences in the magnitude or spatial distribution of the phase shift resulting from the precipitation forcing (GPCP vs. WFDEI) or compared to other GRACE solutions (S6). Further, the lag in TWS simulation can occur due to several mechanisms and processes that are not yet considered in the current model structure such as lateral flow and surface storages (wetland and lakes), vegetation processes, glacier melt, and human influence with dams and reservoirs. However, we don't observe a general or systematic relationship with either elevation, land cover type, soil properties, and the occurrence of lakes and wetlands. There is a tendency that larger negative lags occur more frequently in regions with sporadic permafrost, but the ranges of permafrost fractions are large for both, short and long lags in TWS, suggesting a complex interaction between permafrost extent and its effect on lag in seasonal TWS dynamics. Finally, potential biases in timing of ET due to snow cover and/or vegetation processes may also affect the timing of depletion of SM and TWS. Additionally, high uncertainties of the precipitation forcing and GlobSnow SWE product in mountain (near) regions, as well as leakage errors in the GRACE signal influence the accuracy of both, TWSobs and TWSmod. Although these shortcomings should be kept in mind, we assumed that they do not affect our results regarding to the relative contributions of snow and liquid water to TWS significantly.

In terms of inter-annual variations, the variance in monthly TWSobs values is highly underestimated by modelled TWS, which on the one hand relates to noise within the GRACE signal, but on the other hand may again reflect missing process representation in the model. To reduce the noise, we applied a three-month-moving-average filter on the monthly time series. The smoothed time series then shows fairly good agreement of inter-annual dynamics, with correlation r = 0.68 (Figure 4b, solid lines). Solely the amplitude of the large negative anomaly in 2003 is not captured by the model. While the spatial pattern of this negative TWS anomaly can be simulated, the forcing data doesn't show large anomalies in 2003, so that the model fails to reproduce the magnitude of observed TWS, especially in North America. Issues with the precipitation forcing are further suggested by a negative SWE anomaly of on average 5 mm (see Fig.4 a) indicated in the GlobSnow data, that is not captured by the model, either. The reason why this snow anomaly is not captured by the forcing remains unclear at this

point – it persists when using the WFDEI forcing data set. Besides, the agreement between GRACE and modelled TWS IAV gets substantially better when isolating inter-annual variations by removing the trends in both TWS time series (increase in correlation r from to 0.77). This suggests that the trend in GRACE TWS is to some extent either subject to observational issues or represents a process that is not captured by the model.

### 3.1.3 Comparison with the eartH2Observe model ensemble

Compared to the model ensemble of the eartH2Observe dataset, we achieve equally good or better performance for the spatially integrated SWE and TWS on both, MSC and IAV, scales (Fig. 5, S6). Besides, the majority of the model ensemble obtains similar spatial patterns of performance criteria for SWE as well as for TWS (not shown).

The average observed MSC of SWE is captured with a correlation in the range of 0.79 (PCR-GLOBWB) to 0.99 (ORCHIDEE), whereby only ORCHIDEE shows a better correlation than our model (r = 0.95). However, modelled snow accumulation exceeds that of SWEobs for the majority of the models, which also reflects in higher RMSE (Fig. S6, Fig. S7, Fig. S8). On IAV scales, the correlation in general is lower, yet again we obtain a better fit (r = 0.39) than the model ensemble (r = 0.12 (ORCHIDEE) to 0.28 (LISFLOOD)). However, it remains uncertain, whether the discrepancies between SWEobs and SWEmod represent model deficiencies or evolve from issues related to the GlobSnow SWE retrieval (Schellekens et al., 2017).

Regarding average seasonal TWS variations, our model performs as well as the model ensemble (r = 0.91), with the range of the eartH2Observe ensemble spanning from r = 0.83 (ORCHIDEE) to r = 1.00 (PCR-GLOBWB). The amplitudes in the MSC of TWSmod (95 to 156 mm) are comparable to the observed amplitude of 118 mm, except for SWBM, whose amplitude is twice as large as that of TWSobs. This discrepancy is reflected in relatively high RMSE values for SWBM (Fig. S8). The model ensemble precedes observed seasonal TWS variations by 1 to 1.4 months, similar to our estimates of TWSmod (-1.1 month). Only PCR-GLOBWB, with a higher correlation than other models, shows a smaller average lag of less than 1 month (-0.3 months). This minor difference results from balancing out of preceding and succeeding in different regions over the study domain. Additionally, Schellekens et al. (2017) found that PCR-GLOBWB shows unrealistic snow accumulation over time in Europe and boreal North America. Except for PCR-GLOBWB, the majority of the models obtains comparable spatial pattern of preceding TWS, with small differences at regional scales. Even though the difference in the MSC is commonly attributed to the lack or inadequate size and number of water storages (Kim et al., 2009b), a relationship between model performance and model complexity is not obvious. Relatively complex models, such as HTESSEL, SURFEX, and JULES, show similar phase differences as simpler models, such as SWBM and our model (-1.0 resp. -1.1 months). Since Schellekens et al. (2017) found the largest phase differences in cold regions, they postulate the implementation of processes associated with snow as important factor for this phase lag. In this context, constraining the model with snow observations as done in our study should increase confidence in the representation of snow processes. Nevertheless, we obtain a similar phase difference, which points to the importance of other hydrological processes and storages even in snow-affected regions.

Although compared to the model simulations in the EartH2Observe ensemble, our modelling framework assimilates information from more data streams, e.g. GRACE and GlobSnow data, we only used a subset of 1000 random grid cells to constrain the model parameters. Despite, our model performs better than EartH2Observe ensemble over the whole domain (6050 grids). This improvement in model performance is also consistent among several modelled variables and not limited to storage components only. This suggests that remote sensing data, with larger spatial coverage than site measurements, have a large potential in improving hydrological simulations over a large domain. In addition, remote sensing data also hold potentials beyond the use as an observational constraint and can provide information on identifying and formulating functional relationships across several spatial and temporal scales, that should be addressed in future efforts.

All in all, we conclude that our simple model with a global uniform parameter set achieves considerably good performance regarding observed patterns, especially compared to well-established, more complex models, and with respect to its simplicity and given uncertainties of forcing and calibration data. Thus, we found the model results to be suitable to analyse the composition of TWS across spatial and temporal scales.

### 3.3 TWS variation and composition

### 3.3.1 Spatially integrated

To assess the average composition of seasonal and inter-annual TWS variations, we spatially integrated modelled TWS anomalies as well as modelled anomalies of snow (SWE) and liquid water storages (W) across all grids of the study domain (Fig. 6).

Regarding the MSC, all water storages show a clear seasonal pattern. The maximum $TWS_{mod}$ in March coincides with the maximum in $SWE_{mod}$. On contrary, W remains nearly constant throughout the winter, as related to the lack of evapotranspiration losses and missing infiltration due to prevailing solid precipitation. Starting from March, snow melt decreases $SWE_{mod}$, and thus $TWS_{mod}$, progressively. Thereby $TWS_{mod}$ declines with some delay, as positive W anomalies in April and May suggest buffering of melt water in the soil and on the surface before being transferred to runoff or evapotranspirated. During the summer months, snow is absent, while W decreases due to higher summertime evapotranspiration, and preceding runoff of temporarily stored water. With W and $SWE_{mod}$ being at their minimum in August/September, $TWS_{mod}$ reaches its minimum, too, before starting to increase again in September/October. This rise relates to dropping evapotranspiration rates in combination with more precipitation input (increasing W) and beginning snow accumulation (increasing $SWE_{mod}$). Despite the interplay of $SWE_{mod}$ and W on seasonal variations of the integrated $TWS_{mod}$, the amplitude of W (62 mm) is considerably lower than the one of $SWE_{mod}$ (92 mm) and $TWS_{mod}$ (144 mm). Thus, the seasonal accumulation of snow largely determines the magnitude of $TWS_{mod}$. Additionally, W anomalies at least partly result from snow melt, whereas liquid water does not influence the snow pack. Thus, we conclude that average seasonal TWS variations in northern mid-to-high latitudes are mainly driven by annual snow accumulation and ablation

processes. The Contribution Ratio CR (Eq.(3)) based on the spatially averaged MSC underlines this, as CR = -0.26 indicates that variations in SWEmod explain 63 % of seasonal TWSmod variability (CW = 37 %). This agrees with previous findings of Güntner et al. (2007), who found that SWE contributes to 68 % of seasonal TWS variations in cold regions using the WaterGAP model.

On IAV scales, the pattern seems less homogeneous (Figure 6). In contrast to the MSC, CR = 0.25 suggests larger influence of liquid water anomalies (CW = 62.5 %) than snow anomalies to inter-annual TWS variations. Thereby, we found the main contributor to TWSmod anomalies being dependent on the phase of previous precipitation anomalies, in that they define whether snow fall anomalies lead to anomalies in the SWEmod, or whether rain anomalies directly influence W. Additionally, precipitation input shows larger inter-annual variability than evapotranspiration or runoff losses, and thus

dominates the change in water storages on IAV scales (not shown). Large TWSmod anomalies, such as in 2005, 2010 and 2012, follow anomalies in wintertime precipitation and go along with anomalies in SWEmod (Figure 6). On contrary, summertime anomalies related to W are usually less pronounced in their magnitude (e.g. 2003, 2006). We attribute this to accumulating effects of snow storage anomalies over the cold period, as they integrate all anomalies of previous cold months while the impact of evapotranspiration and runoff is reduced. Accordingly, largest TWSmod anomalies are obtained in early

spring before snow melt starts and when snow accumulation is highest. Nevertheless, since W is influenced by the quantity of snow melt, anomalies in SWEmod implicate subsequent changes in W. Additionally, snowpack anomalies are eliminated each summer, while W anomalies dominate the summer. As a result, W anomalies in any case affect TWSmod variability on IAV scales when analysing the spatial average composition.

Besides, Güntner et al. (2007) demonstrated a shift from short-term storages with high seasonality such as SWE on MSC

scales towards storages with longer delay times on IAV scales. Although modelled W mainly represents soil moisture, it implicitly includes surface and ground water storages. Thus, its contribution of CW = 62.5 % to inter-annual TWS variations is roughly comparable to 55 % contribution from soil moisture (27 %) and surface water (28 %) in cold regions found by Güntner et al. (2007). Despite, the relatively large influence of surface water bodies shown by Güntner et al. (2007) suggests that the lack of explicit surface water storages in this study may contribute to remaining discrepancies with GRACE and the

lower magnitude of modelled inter-annual TWS variability compared to GRACE estimates.

### 3.3.2 Local grid scale

Based on CR (Eq.(3)), Fig. 7 shows the relative contribution of SWEmod and W variances to total TWSmod variability on MSC and IAV time scales for each grid. Thereby, blue colours represent prevailing SWEmod variations as indicated by CR < 0, while red colours show dominance of variations in W (CR > 0).

Accordingly, variations in the MSC of TWSmod are mainly influenced by snow in northern regions, with the mean CR = -0.30 indicating that on average 65 % of seasonal TWSmod variability can be explained by SWEmod (CW = 35 %) (Fig. 7Figure 7a). The contribution of variation in liquid water in general increases southwards and prevails seasonal TWSmod variability south of approximately 50° latitude. An exception is Europe, where the influence of W reaches up to 60° latitude,

and where the transition to snow dominated regions is more gradual. Since the calculated variations in RW are low, the majority of modelled W represents variability in SM.

This obtained pattern confirms earlier studies, that showed dominance of snow in northern latitudes in North America (Rangelova et al., 2007), and prevailing soil moisture dynamics further South e.g. in the Mississippi basin (Ngo-Duc et al., 2007;Güntner et al., 2007). Besides, the north extent of dominating W reflects the temperature gradient in North America and Eurasia. Comparison with average annual temperature suggests, that for T > 10 °C variability of W dominates, while for T < 0°C SWEmod dynamics prevail. This is plausible, as temperature determines annual snow accumulation, and the relative contribution of liquid water increases in the absence of snow. Yet, it further highlights the dependency on the used temperature data set, especially in a model that calculates snow fall and snow melt based on temperature thresholds as ours.

Opposed to the MSC, the variability of W dominates TWSmod variations on IAV scales in the entire study domain, as clearly indicated by average CR = 0.63 (Fig. 7b). Inter-annual variations of SWEmod seem to be relevant only in regions that receive highest annual snow amounts, such as the Canadian Arctic Archipelago, the northern west coast of North America, North East Siberia and the northern West Siberian Plain. Due to a prolonged cold period there, the time span with rain fall and evapotranspiration is short, decreasing the occurrence of potential variability in W. However, even in these regions the influence of SWEmod is low compared to the MSC. This reduced importance of snow on inter-annual scales again agrees with the findings of Güntner et al. (2007).

Apart from that, and since we already showed a link between average TWSmod IAV and previous precipitation anomalies, and as precipitation represents the main model forcing data, we investigated the relative contribution of rain and snow fall to inter-annual variability of total precipitation (Fig. 8). Similar to the composition of TWSmod on local scale, rain anomalies prevail for most of the grid cells (mean CR = 0.68). This is consistent when snow fall is not scaled by $p_{sf}$ and thus suggests that the greater contribution of W to inter-annual variations of TWSmod on local scale relate to highly variable (liquid) summertime precipitation events. On contrary, monthly snow fall seems less variable between years, resulting in less pronounced variations in SWEmod compared to W. Exceptions are regions of high maximum SWEmod, that accordingly show a considerable relative contribution of snow to the inter-annual TWSmod variability.

### 3.3.3 Comparison of different scales

Figure 9 summarizes the above presented contributions to TWSmod variability across spatial and temporal scales. As explained in the previous sections, we obtained two scale dependent differences in the relative contribution to TWSmod variability: (1) in general between temporal scales, and (2) for inter-annual variability between spatial scales.

Regarding (1), Fig. 9 emphasizes again that seasonal variations of TWSmod are mostly determined by seasonal snow dynamics, while on inter-annual scales TWSmod variability mainly originates from variations in liquid water. As previously stated, determination by SWEmod dynamics on MSC scales relates to the pronounced magnitude of seasonal snow variations in northern mid-to-high latitudes. In comparison, average monthly changes in W were found to be minor and

additionally influenced by snow ablation. Thereby, the spatially integrated CR (black star) roughly agrees with the average of local contributions (dashed line).

Concerning IAV scales, we found that the determination of TWSmod variability by W relates to larger inter-annual variations in liquid precipitation compared to snow fall. However, considerable differences between spatial scales exist (Figure 9). Opposed to the MSC, the spatially integrated CR (black star) for IAV is not within the interquartile range of local CR. This indicates a relatively larger effect of SWEmod variations when looking on the spatially integrated values compared to local values. Liquid water storages are highly heterogeneous in space, mainly due to heterogeneous rainfall anomalies. On contrary, snow variability is affected by fewer factors, and mainly regulated by a range of temperature values that control freezing and melting. Temperature anomalies in turn show sizeable spatial coherence across large areas. To assess the spatial coherence of W compared to SWEmod, we calculated the proportion of total positive and total negative covariances among grid cells (Fig. 10).

For inter-annual variations of SWEmod, the sum of positive covariances prevails (Fig. 10a), whereas positive and negative values are more in balance for W (Fig. 10b). This suggests SWEmod anomalies to be more spatially coherent than anomalies of W. Thus, when spatially averaging, the more homogeneous snow anomaly patterns gain importance. On contrary, opposed anomalies of W compensate each other more strongly. This leads to a relatively larger influence of SWEmod to the spatially integrated inter-annual TWSmod variability compared to when analysing the local grid scale. Since positive covariation clearly dominates for temperature anomalies, the spatial coherence of SWEmod relates to their homogeneous patterns (Fig. 10c). Similar to W, positive covariances only slightly outweigh for year-to-year variations in rain fall (Fig. 10d). The same is true for snow fall (not shown). Therefore, the spatial coherence of SWEmod anomalies is less pronounced than for temperature, as snow is additionally influenced by snow fall anomalies. Regarding W anomalies, this indicates that the spatial heterogeneity in our model, which misses explicit information on soils, topography, etc., mainly results from inhomogeneous patterns in rain fall anomalies. Thereby, the slightly more balanced positive and negative covariations for W compared to rain fall can be ascribed to the additional impact of primarily radiation-driven evapotranspiration to SM.

In order to ensure that these results are not artificially caused by the forcing data, we did the same analysis running the model with rain and snow fall estimates of the WFDEI product (Weedon et al., 2014). Since we observed the same patterns, we assume our findings to be robust (S7.1).

### 3.4 Limitations of the approach

Although the model of this study reproduces observed hydrological patterns well and achieves comparable results to state-of-the-art models, its low complexity and the applied calibration approach are associated with limitations in terms of process understanding and predictive power.

First of all, the simple structure only allows inferences on represented processes, that likely include effects of fluxes and storages not considered explicitly. For example, the model does not resolve individual liquid water storages such as deep groundwater and surface water explicitly. As discussed previously, our delayed land runoff comprises various (intermediate)

storages and delay times, and thus cannot be associated with one distinct storage component. Even though soil moisture is distinguished from these slowly varying reservoirs, its quantity and pattern have not been directly validated. Future research is required to increase confidence by including remote sensing based data of soil moisture in calibration and/or validation. However, these satellite data still have limited value as the microwave signals can only capture moisture in the upper 5 cm of

soil and do not provide estimates under snow-cover and dense vegetation (Döll et al., 2015;Lettenmaier et al., 2015). Therefore, a multi-layer soil scheme is needed to compare model outputs to satellite derived soil moisture estimates, as for example successfully demonstrated by Albergel et al. (2017).

Further, the model does not include any human-induced changes of water storages, which yet contribute to observed TWS variability in many regions (Döll et al., 2015;Rodell et al., 2015). Other simplified or ignored hydrological processes include

the coincident occurrence of rain and snow fall, liquid water capacity of snow, interception, freeze/thaw dynamics within the soil, capillary rise and other surface-groundwater interactions, the effect of vegetation or other surface properties, as well as lateral flow from one grid cell to another. Especially in the downstream areas of large basins the latter represents a potential input that may significantly affect total TWS (Kim et al., 2009a), and thus may contribute to the discrepancy between TWSobs and TWSmod in some regions. Besides, the model does not account for spatial variability of topography and land

surface characteristics.

With regards to model parameter, we apply a global uniform parameter set and do not regionalize the parameters according to spatially distributed physio-geographical characteristics. In contrast, most macro-scale hydrological models include spatially distributed soil properties to define parameters related to infiltration, soil water holding capacity and percolation, as well as vegetation types to assess the effects of different plant functional types on evapotranspiration and canopy storage

(Sood and Smakhtin, 2015). Our model only implicitly considers the effects of vegetation for example on ET, but not its spatial variability, as the associated impacts are included in the observational constraint. Spatial variability of model parameters might affect the relative contributions of different storage components to TWS variability at different spatial scales. However, the comparison with eartH2Observe models, which generally involve spatial heterogeneity in model parameters, suggests that the main conclusions remain unchanged. Additionally, we want to highlight that spatial distribution

of model parameters depend on assumptions and some degree of simplification as well, and thus does not necessarily improve model performance compared to a global uniform parameter set obtained from multiple observational data. Further, as we encountered issues with parameter equifinality, especially between modelled snow melt and sublimation, future efforts should include a stronger utilization of runoff data in the calibration and validation process. This would help to better constrain between water fluxes to the atmosphere and liquid water fluxes, that can contribute to the runoff.

Finally, though the implemented cost function explicitly accounts for the uncertainty of the calibration data, additional uncertainties of other input data, their processing and characteristics remain partly unaddressed.

**Conclusion**

In this study, we assessed the relative contributions of snow pack versus soil and retained water variations to the variability of total terrestrial water storage (TWS) for northern mid-to-high latitudes. To do so, we constrained a parsimonious hydrological model with multi-criteria calibration against multiple Earth observation data streams including TWS from GRACE satellites and snow pack estimates from GlobSnow. The optimized model showed considerably good agreement with observed patterns of hydrological fluxes and states, and was found to perform comparable or better than simulations from state-of-the-art macro-scale hydrological models. This underlines the potential of simple hydrological models tied to observational data streams as powerful tools to diagnose and understand large scale water cycle patterns. Further, it highlights the benefits of considering multiple, complementary data constraints to overcome their individual shortcomings.

Consistent with previous studies, we found that seasonal TWS variations are dominated by the development of snow pack during winter months in most places of the mid-to-high northern latitudes. In contrast to this seasonal pattern, our study reveals that not snow but anomalies in liquid water storages, mainly comprising soil moisture, drive inter-annual TWS variations in almost the entire spatial domain. This counter-intuitive pattern was found to relate to larger rainfall anomalies as compared to snowfall anomalies.

Apart from the time-scale dependent dominant controls on TWS variations, we additionally observed different behaviour across spatial scales. In terms of seasonal variations, the spatially integrated contribution reflects the average of the spatial domain. However, and more interestingly, the relative contribution of snow pack variations to total TWS inter-annual anomalies appears to be larger when spatially integrated than at local scale. We found this pattern results from stronger spatial coherence of snow pack anomalies compared to anomalies in other storages, such that the latter cancel out more strongly than the former when calculating an average across large spatial domains. The stronger spatial coherence of snow pack anomalies seems to be driven by the nature of spatially coherent temperature anomalies that determine snow accumulation and melt. These findings imply that patterns from large scale integrated signals should not be associated with locally operating processes, since spatial covariations of climatic variables can confound the picture.

Overall, our study underlines the benefits of GRACE TWS as a model constraint, and moreover, stresses the importance of temporal and spatial scale when assessing the determinants of TWS variability. Clearly, insights obtained at one scale cannot be transferred to another, as is often (unintentionally) done. Hence, TWS variations in northern latitudes seem to be not merely subject to snow variability, but rather are driven by soil moisture on inter-annual scales - which may be of considerable importance when assessing long-term water availability in the context of climate changes.

**Authors contribution**

TT, SK and MJ designed the research in extensive collaboration with NC, AE and MF. CN processed and integrated runoff estimates from EU-OBS in model calibration. NC contributed to parameter estimation and uncertainty analysis. TT conducted the analysis. All co-authors contributed to the preparation of the manuscript.

## Competing interests

The authors declare that they have no conflict of interest.

## Acknowledgements

This research was carried out within the initiative for the development of SINDBAD, the Strategies to Integrate Data and Biogeochemical models, framework.

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

**Table 1.** Overview on data applied for meteorological forcing and multi-criteria calibration resp. model evaluation (NH: Northern Hemisphere).

| Variable | | Dataset | Coverage and resolution | | Reference |
|---|---|---|---|---|---|
| | | | Spatial | Temporal | |
| *Meteorological forcing* | | | | | |
| **P** | precipitation | GPCP 1dd v1.2 | 1° x 1° global | daily 1996–present | Huffman et al. (2000);Huffman and Bolvin (2013) |
| **T** | air temperature | CRUNCEP v6.1 | 0.5° x 0.5° global | daily 1901–2014 | Viovy (2015) |
| **Rn** | net radiation | CERES SYN1deg Ed3A | 1° x 1° global | 3-hourly 03/2000–05/2015 | Wielicki et al. (1996) |
| *Calibration and evaluation* | | | | | |
| **TWS** | terrestrial water storage anomalies | GRACE Tellus JPL-RL05M v2 | 0.5° x 0.5° global | monthly 2002–2016 | Watkins et al. (2015);Wiese (2015) |
| **SWE** | snow water equivalent | GlobSnow v2.0 | 0.25° x 0.25° non-alpine NH | daily 1979–2012 | Luojus et al. (2014) |
| **ET** | evapo-transpiration | FLUXCOM | 0.5° x 0.5° global | daily 1982–2013 | Tramontana et al. (2016) |
| **Q** | runoff | EU-RUN v1.1 | 0.5° x 0.5° Europe | monthly 1950–2015 | Gudmundsson and Seneviratne (2016) |

**Table 2.** Adjustable model parameters, their meaning, calibration range (theoretical range in brackets), optimized value including estimated uncertainty, and the corresponding equation in S1.

| Parameter | Description | Unit | Range (theoretical) | value | Optimized ± uncertainty (%) | | Eq. |
|---|---|---|---|---|---|---|---|
| **Snow** | | | | | | | |
| $p_{sf}$ | scaling factor for snow fall | - | 0–3 ($\infty$) | 0.67 | ± 1e$^{-3}$ | (< 1 %) | (S2) |
| $sn_c$ | minimum SWE that ensures complete snow cover of the grid | mm | 0–500 ($\infty$) | 80 | ± 19 | (24 %) | (S3) |
| $m_t$ | snow melt factor for T | mm K$^{-1}$ d$^{-1}$ | 0–10 | 2.63 | ± 0.26 | (10 %) | (S4) |
| $m_r$ | snow melt factor for Rn | mm MJ$^{-1}$ d$^{-1}$ | 0–3 | 0.90 | ± 0.05 | (6 %) | (S4) |
| $sn_a$ | sublimation resistance | - | 0–3 | 0.44 | ± 0.01 | (3 %) | (S5) |
| **Soil** | | | | | | | |
| $s_{exp}$ | shape parameter of runoff-infiltration curve | - | 0.1–5 | 1.46 | ± 0.02 | (2 %) | (S12) |
| $s_{max}$ | maximum soil water holding capacity | mm | 10–1000 (0–$\infty$) | 515 | ± 9 | (2 %) | (S12) |
| $et_a$ | alpha coefficient in Priestley-Taylor formula | - | 0–3 | 1.20 | ± 0.01 | (1 %) | (S14) |
| $et_{sup}$ | ET sensitivity / SM fraction available for ET | d$^{-1}$ | 0–1 | 0.02 | ± 6e$^{-5}$ | (< 1 %) | (S18) |
| **Runoff** | | | | | | | |
| $q_t$ | recession time scale for land runoff | d | 0.5 (0)–100 | 13 | ± 4 | (31 %) | (S20) |

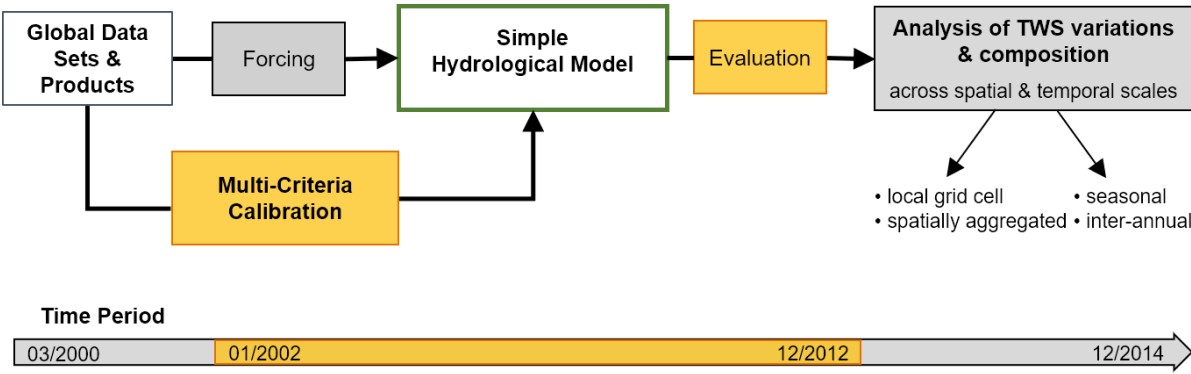

**Figure 1.** Experiment design and considered time periods for forcing/analysis (grey) and model calibration/evaluation (orange).

**Figure 2.** Schematic structure of the model with calculation of TWS. Boxes denote the water storages [mm]: snow water equivalent SWE, soil moisture SM, retained water RW, liquid water W and total terrestrial water storage TWS. Fluxes are represented by arrows. Red colour identifies forcing data: precipitation P [mm d$^{-1}$], air temperature T [°C] and net radiation Rn [MJ m$^{-2}$d$^{-1}$]; while green colour indicates variables constrained by observations: evapotranspiration ET [mm d$^{-1}$], runoff Q [mm d$^{-1}$], SWE [mm] and TWS [mm].

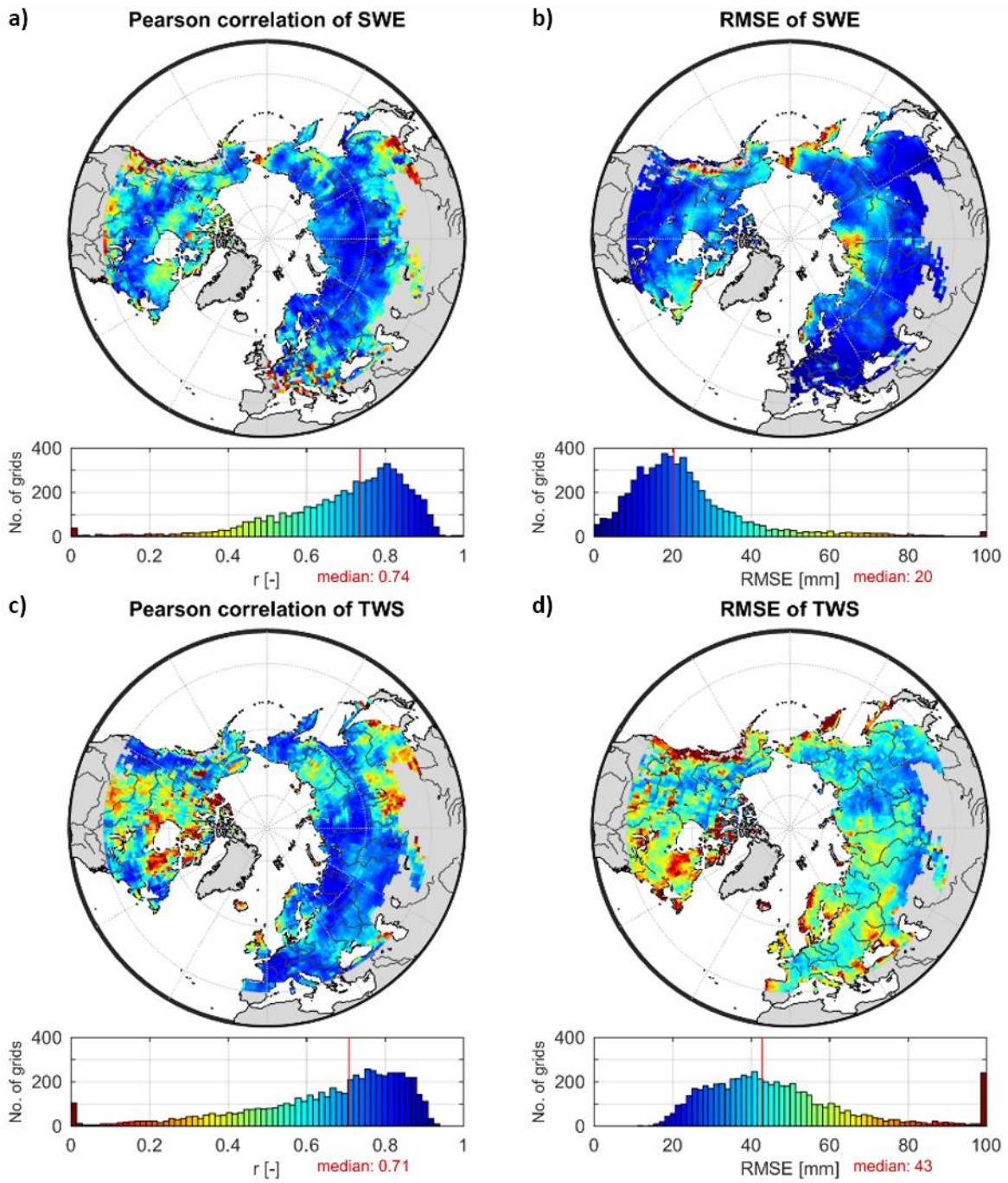

**Figure 3.** Pearson correlation coefficient r **(a,c)** and root mean square error RMSE **(b,d)** between monthly values of modelled SWE and GlobSnow SWE **(a,b)**, as well as modelled TWS and GRACE TWS **(c,d)** for the period 2002–2012 and for each 1° x 1° grid cell of the study domain. Values of r are truncated to the range 0-1 (**a,c**), and values of RMSE to the range 0-100 (**b,d**).

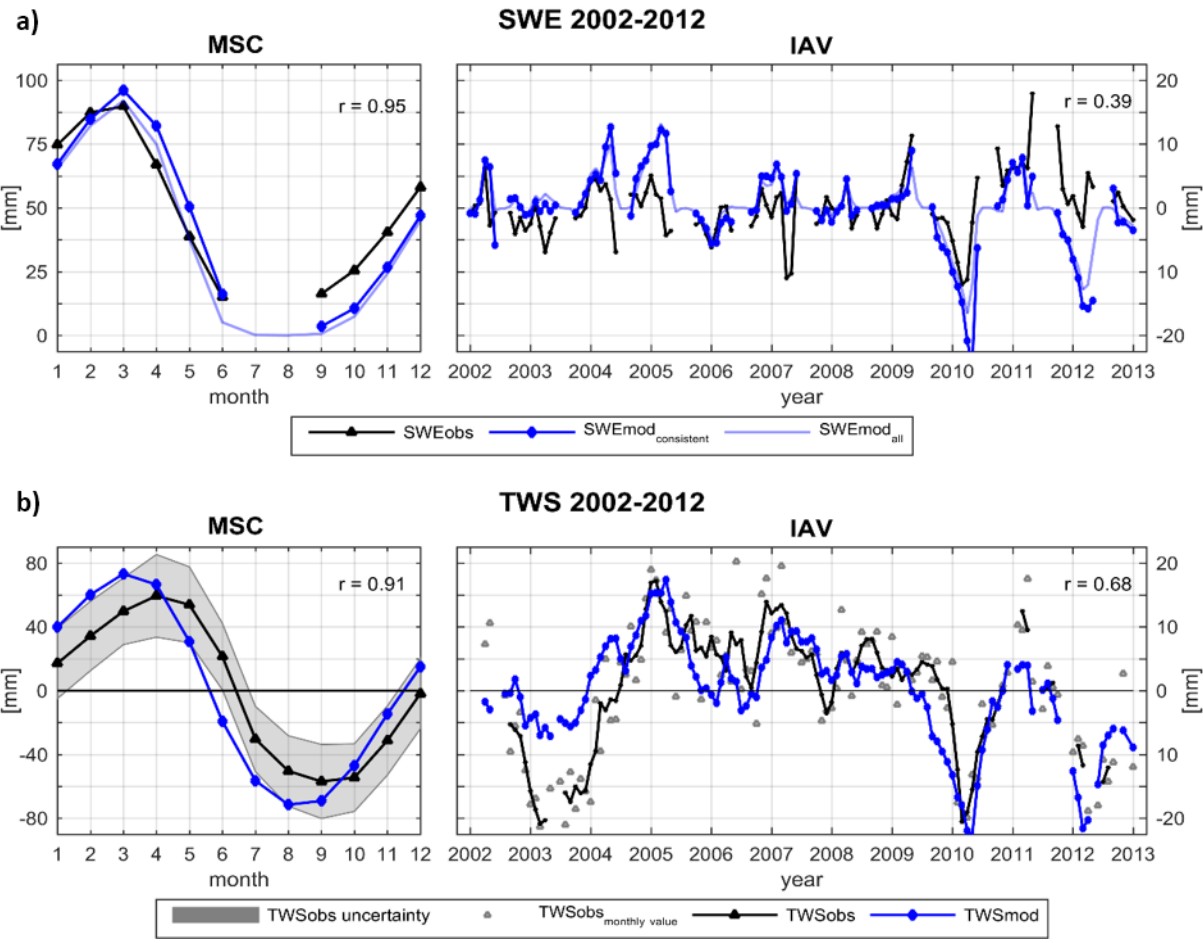

**Figure 4.** Spatially averaged mean seasonal cycle (MSC) of the period 2002–2012 as well as inter-annual variability (IAV, difference between monthly values and the MSC) for **a)** SWE and **b)** TWS. In a), SWEmod_consistent refers to modelled SWE considering only data points with available SWEobs, while SWEmod_all incorporates all time steps for all grids of the study domain. Correlation r is calculated only for consistent data point, respectively. In b) IAV, TWSobs_monthly value shows the original IAV of individual TWSobs months, while TWSobs and TWSmod are smoothed using a 3-month average moving window filter. Correlation r refers to the smoothed values. For the MSC in b) no smoothing is applied.

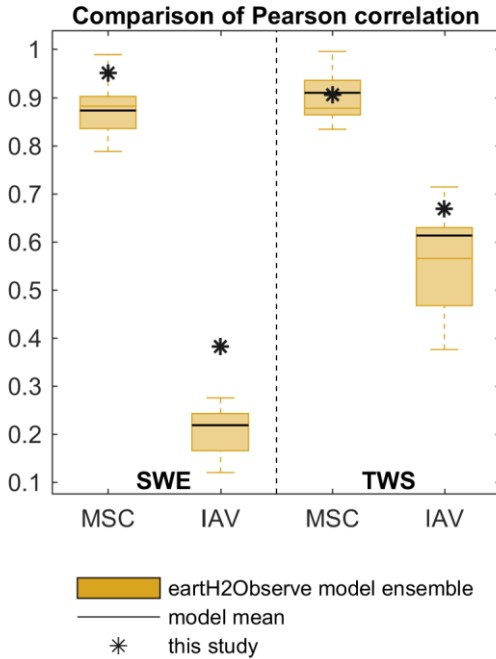

**Figure 5.** Pearson correlation for the spatially integrated SWE (left) and TWS (right) achieved by this study compared to the model ensemble of eartH2Observe dataset across temporal scales. In each box, the central orange line represents the median and the edges the 25 % and 75 % percentiles of the model ensemble, while the solid black line marks the performance of the ensemble mean.

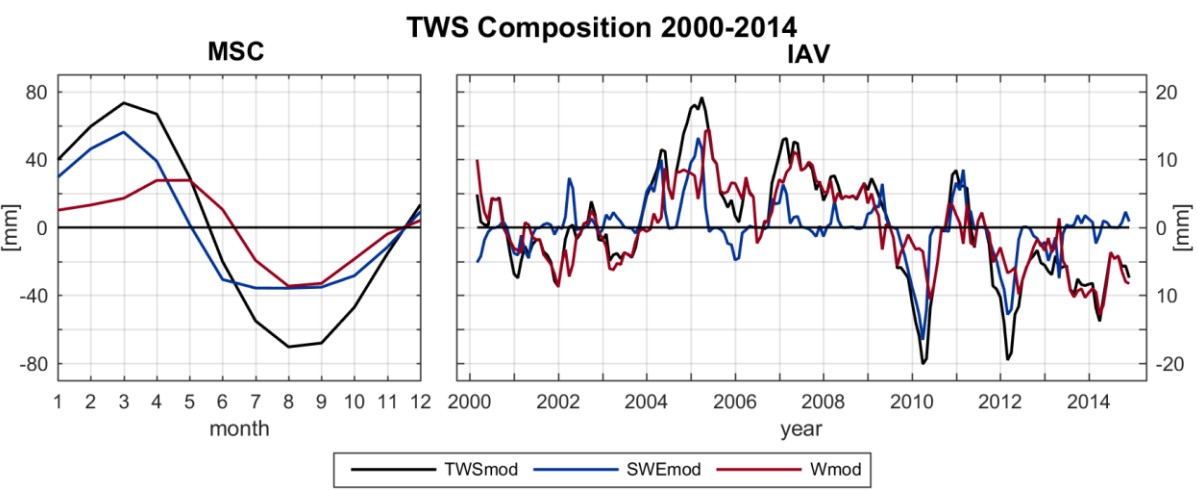

**Figure 6.** Spatially averaged mean seasonal cycle (MSC) of the period 2000–2014 as well as inter-annual variability (IAV, difference between monthly values and the MSC) for modelled TWS, SWE and W anomalies to the time-mean of 2000–2014.

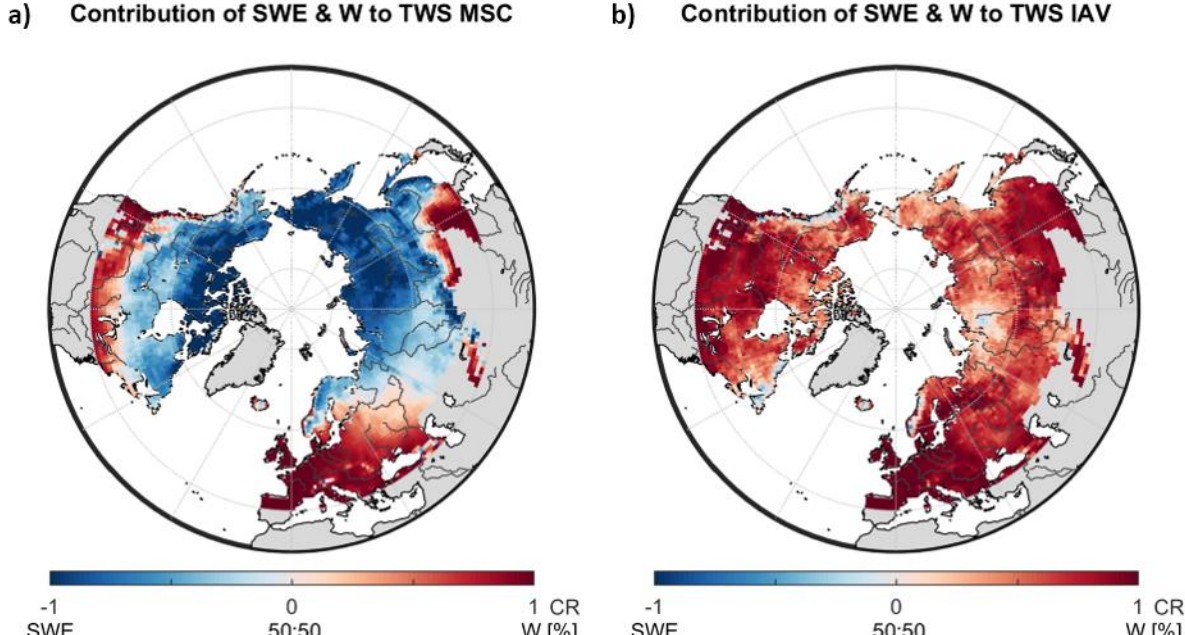

**a)** Contribution of SWE & W to TWS MSC

**b)** Contribution of SWE & W to TWS IAV

**Figure 7.** Relative contribution based on CR (Eq.(3)) of modelled snow (SWE) and liquid water (W) storage anomalies to **a)** mean seasonal variations from 2000–2014 of modelled TWS anomalies, and **b)** inter-annual variations of modelled TWS anomalies for each grid cell of the study domain, respectively.

Contribution of snow fall & rain fall to P IAV

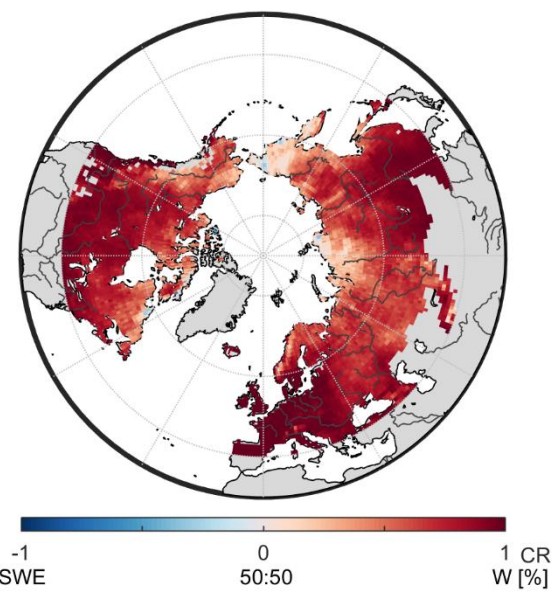

**Figure 8.** Relative contribution based on CR (Eq.(3)) of modelled snow fall and rain fall to total precipitation (P) anomalies on inter-annual (IAV) scales for each grid of the study domain.

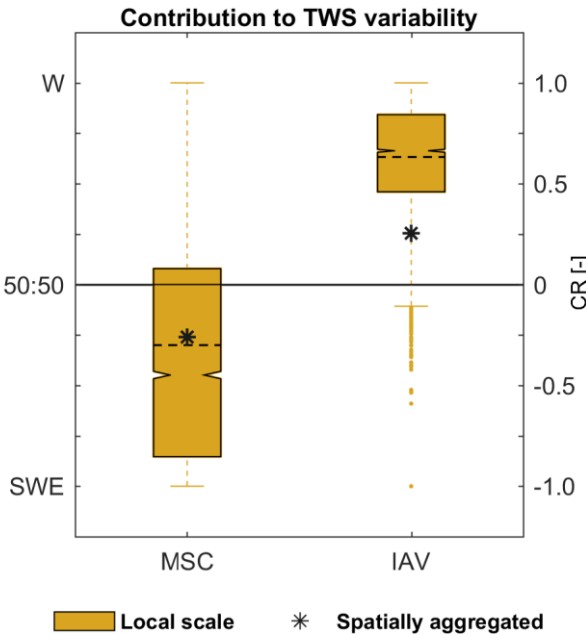

**Figure 9.** Relative contribution of snow (SWE) and liquid water (W) to TWS variability on different spatial (local grid scale, spatially integrated) and temporal (mean seasonal MSC, inter-annual IAV) scales based on CR (Eq.(3)). The boxplots represent the distribution of grid cell CR, with the dashed line marking the corresponding average. The star represents the CR calculated for the spatially integrated values.

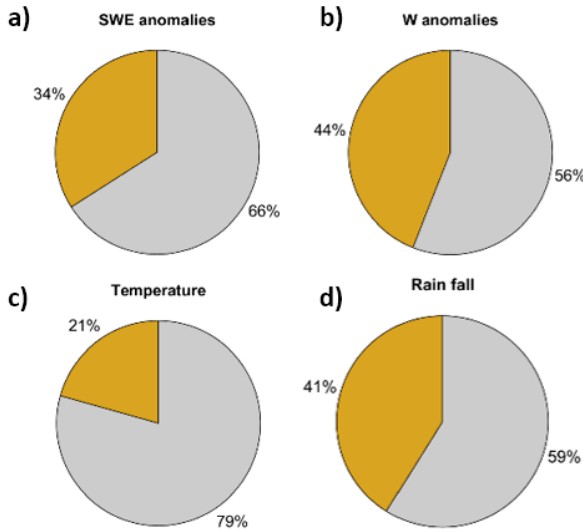

**Figure 101.** Proportion of total positive (grey) and negative (orange) covariances among grid cells for inter-annual variations of **a)** snow (SWE), **b)** liquid water storages (W), **c)** temperature, and **d)** rain fall.