# Peer review of "Understanding terrestrial water storage variations in northern latitudes across scales"

_Hydrology and Earth System Sciences, 2017_

## Referee Comment (RC1) · V. Humphrey (Referee) · 22 Feb 2018

This study aims to evaluate the relative contribution of snow versus liquid water in (total) terrestrial water storage changes in northern latitudes. In order to investigate this question, the authors construct a simple bucket-type hydrological model with 10 free parameters which they calibrate against four different datasets (satellite observations of terrestrial water storage from GRACE satellites, snow water equivalent from the GlobSnow product which combines satellite and ground observations, evapotranspiration from FLUXCOM which is based on an ensemble of machine learning methods calibrated with in-situ observations, and E-RUN estimates of gridded runoff also based on a machine learning model calibrated with in-situ observations). Following a short evaluation of the performance of the presented model (and a comparison with the

Earth2Observe ensemble of hydrological models), the main results are focused on distinguishing the respective contributions of snow and liquid storages to terrestrial water storage. Their analysis contrasts 1) local scale effects versus a spatially integrated average and 2) the mean seasonal cycle versus inter-annual variability. Consistent with previous studies, the authors find that the seasonal cycle is dominated by the snow component. The main finding of the paper is that liquid water storage clearly dominates inter-annual variability both at local scale and when considering a spatially integrated time series. They also find that the relative contribution of liquid water is weaker for the spatial integral compared to the local scale analysis. The authors argue that because snowpack evolution is primarily dependent on temperature (which has high spatial coherence (fig. 10)), this explains why the relative contribution of snowpack to large-scale inter-annual TWS variability is higher. In their conclusions, the authors comment on the usefulness of a simple hydrological modeling approach informed by multiple observational constraints. They suggest that long-term changes in water availability in northern latitudes might be driven by soil moisture rather than by snow dynamics.

This is a really good and well conducted paper. I find the results very interesting and worthy of publication in HESS. One can see that a lot of effort was invested in developing a custom hydrological model and this is reflected by the relatively important share of the methods and model evaluation sections in the paper. However, the authors manage to keep the results and discussion focused around the primary objective of quantifying the relative contribution of snow and non-snow storages to overall water storage variability.

I have four major comments/suggestions which I would like the authors to consider as well as some minor comments that are listed below.

**Major comments**

In their results, the authors find that the modeled seasonal cycle of TWS has a systematic lag compared to observations (model TWS preceding observed TWS). This

lag is also present in other models from the Earth2Observe ensemble. The analysis of the authors convincingly shows that their modeled snow storage seems to have a correct phase and is therefore not responsible for this lag between modeled and observed TWS. They also mention that adding delayed storage responses (as e.g. with a groundwater module) could not correct this effect either. I find this a major finding for the research community (which could be made more prominent in the conclusions) since it is often supposed that such model errors mainly stem from the lack of long-memory water storages and poor representation of snow dynamics. Here the authors conclude that neither of these seem responsible and that the origin of the phase lag in TWS must reside elsewhere, which brings me to my main suggestion below. One important limitation that the authors fail to mention is that there is no consideration of permafrost and liquid/solid phase transitions of soil moisture content. In the proposed model, soil moisture does not have temperature neither does it store energy. In reality, it is well known that freeze/thaw dynamics are also a dominant factor for water and energy fluxes in high latitudes. Freeze/thaw is the on and off switch for evapotranspiration and vegetation growth. However, a phase lag between the availability of energy and the ET response cannot be modeled with the current model setup (the alpha parameter only conditions ET amplitude). Potentially, a lot of ground heat flux might be required before ET can actually take place. In addition, from my understanding of the equations presented in the supplementary material, actual ET is not reduced in the case of snow cover neither is it dependent on vegetation growth. This might introduce a too early response of ET to net radiation compared to reality, leading to a fast rise of soil moisture depletion already in early spring. Later, soil moisture would become limiting already in mid-summer and ET would peak in June and start to reduce already in July (Fig S1). The reference below suggests a peak of vegetation growth in August for a boreal forest (from one FLUXNET site). The authors might consider exploring this direction and maybe check whether there is some evidence that FLUXCOM ET itself (the observational constraint) already contains such a phase lag. As this would require some additional work, it would also be fine if the authors prefer to simply mention this

as a hypothesis to explore.

Brown, S. M., Petrone, R. M., Chasmer, L., Mendoza, C., Lazerjan, M. S., Landhäusser, S. M., ... & Devito, K. J. (2014). Atmospheric and soil moisture controls on evapotranspiration from above and within a Western Boreal Plain aspen forest. Hydrological processes, 28(15), 4449-4462.

This leads me to my second main comment: The separation of TWS into liquid and snow water seems a bit misleading since the liquid phase might implicitly include some frozen water as well (frozen soil moisture). As mentioned by the authors, there is a mismatch between explicitly represented processes and observed processes (TWS includes frozen water) that may be compensated by adjustments in model parameters. The expression "liquid phase" is hence misused in my opinion and might very well lead to confusion. It might be more accurate to refer to snow versus non-snow changes as done for example in page 27 line 30. I think this terminology should be extended to the rest of the manuscript.

Third comment: Figure S7 is quite pre-occupying because it suggests a dependency of your results on the forcing dataset. For instance, the difference might be related to your partitioning between snowfall and rainfall (which was not applied when using WFDEI). One possibility to check if this comes from uncertainty in the precipitation data might be to compare the regional mean time series of the two products and look for large differences in 2005 and 2010. This would also indicate whether GPCP-1DD appears superior to WFDEI. In relation to this -> Line 11-12 page 26: this is a rather unsubstantiated statement. Please give it more weight, for instance by replicating key figures (e.g. Fig 9) in the supplementary material.

Fourth comment: You could make lines 24-29 of your abstract clearer. Upon first reading, I understood that snow dynamics dominate IAV on a large scale, which is not the case. It should be clearly said that "liquid water" dominates IAV at all spatial scales while snow dominates MSC at all spatial scales (Fig. 9). In addition, for IAV, the relative influence of snow increases with spatial aggregation due to the spatial coherence of T, a main driver of snowfall and snow melt. The wording "liquid water storages, comprising mainly of soil moisture" is also a bit misleading. It is not really clear what is implicitly incorporated in the soil moisture reservoir in order to fully reproduce TWS (as mentioned in the third comment). Güntner et al. 2007 provides a similar analysis based on WaterGAP. This would be an interesting point of comparison since they indicate a contribution for IAV of 33% snow, 27% soil and 12% groundwater and 28% surface water (!) for cold climates (their table 5). I think this reference should be discussed and compared with your results.

Güntner, A., Stuck, J., Werth, S., Döll, P., Verzano, K., & Merz, B. (2007). A global analysis of temporal and spatial variations in continental water storage. Water Resources Research, 43(5).

**Minor comments**

Your work is very new and promising in the sense that multiple remote sensing or machine-learning observation-derived products are used simultaneously for calibrating a hydrological model. This is not easy to do and a research direction worth to explore. The overall modeling framework however still relies on a very standard land surface model structure. One missed opportunity may be to have used these observational datasets not only to calibrate model parameters, but also to identify functional relationships directly from the data (as opposed to fitting the parameters of a pre-defined equation to the data). Such research might be suggested as one possible future direction in the discussion. Finally, the paper does not emphasize on the added value of using remote sensing products to constrain the model (except for a lower RMSE against observations, which is somewhat expected since other models were not calibrated with these observations). Could similar results have been obtained with the Earth2Observe ensemble ? (especially on IAV?) If not, this would better show the merit and relevance of the presented approach.

Liquid water is explicitly modeled as soil moisture + runoff routing but also likely includes river storage, lakes and wetlands implicitly (e.g. large water holding capacity mentioned on page 12, lines 7 and 19). This could also be made a bit clearer already in the model description in order to avoid some confusion later. Using a snow/non-snow terminology would also help resolving this.

It could be made clearer that runoff is currently only generated from infilitration limitation (e.g. no baseflow in Eq. S10). Also mention that this is partially compensated by the recession time scale parameter that delays runoff generated on a specific day. Likely because the model is evaluated at monthly scale, this only has a limited impact on model performance and this model parameter is the least constrained by observations.

Methods: You could make a better distinction between purely observational products, and observation-based upscaled products such as Tramontana et al. or Gudmundson et al. which also rely on the quality of the underlying forcing data.

Line 29-30, page 10: this assumption seems a bit dangerous given figure 6. Could you please document the degree to which this assumption is correct and if this might affect the results qualitatively (possibly in supplementary information)?

** in-text comments **

Line 29, page 2: and in addition there can be no retrieval of SM in snow-covered or highly vegetated regions.

Line 3, page 4: for clarity, maybe you could add an introductory sentence indicating that this whole section is meant to give an overview of the model setup.

Line 10, page 4: E-RUN, based on E-OBS

Line 10-14, page 7: I thought FLUXCOM was based on an ensemble of machine learning algorithms (e.g. not only random forest). Could you also briefly comment on the performance of FLUXCOM in snow regions and high latitudes? Any idea if FLUXCOM is already accounting for sublimation?

Page 9: Is there any reference for this cost function ?

Lines 17-20 page 9, I think this is indeed a very good idea!

Line 22, page 9, can you indicate where these commonly reported values can be found? (It's also fine if you decide to assume 10%)

Line 8, page 10 : typo

Line 22-23 page 11: interestingly however, this also contradicts Behrangi et al. 2017 for mountainous regions..

Behrangi, A., Gardner, A. S., Reager, J. T., & Fisher, J. B. (2017). Using GRACE to constrain precipitation amount over cold mountainous basins. Geophysical Research Letters, 44(1), 219-227.

Line 15 page 12 : maybe this rather small value is in relation with the relatively large soil water holding capacity.

Line 12-13 page 15: For instance, Humphrey et al. 2016 figure 6 shows that the central North America and Eastern Eurasia is rather dominated by IAV (which appears more difficult to model according to your figure 5).

Humphrey, V., Gudmundsson, L., & Seneviratne, S. I. (2016). Assessing global water storage variability from GRACE: Trends, seasonal cycle, subseasonal anomalies and extremes. Surveys in geophysics, 37(2), 357-395.

Line 16 page 12: the sentence is inaccurate : a recession time scale of x days does not mean that only runoff of the preceeding x days contributes to "total runoff" (check Orth et al. 2013).

Line 13, page 13 : maybe not necessary to say that these approaches are not commonly accepted as this might be a subjective statement in my opinion. The arguments you give just before (on overfitting) and the continental-scale focus of your study might be sufficient arguments. Another argument you could mention is that allowing locally

varying parameters would contaminate your conclusions: with locally dependent parameters, the differences in local-scale / large-scale contribution to IAV might due to the spatial dependency of parameters. But with your current setting, they can only be attributed to climate forcing. This is also why it makes a very clean experiment setup. This last point also calls for one caveat in the conclusion : your picture of the partitioning and scale-dependency of liquid versus snow might also change once you introduce spatial variability of the model parameters (e.g. snow melt factor might be very dependent on the vegetation cover, contrasting the responses of tundra versus boreal forests).

Page 14, lines 3-6 : essentially repeats page 13 line 10.

Figure 3. If values were truncated (e.g. Fig3d) this should be indicated in the legend and in labels.

Line 1 page 19 : TWSmod ?

Line 7 page 19 : typo in earth2observe

Line 5 page 10 : replace "grids" with "grid cells" idem on lines 5-6 page 20

Line 5 page 20 : was the use of a subset mentioned also in the methods ?

Line 6 page 21 : coincides

Figure 7: It would be nice to add units to the colorbars (in addition to qualitative labels), same in Figure 8.

Line 18 page 22: is "received" the adequate word?

Line 7 page 23: frozen soil is not modelled

Line 11 page 25: On first read I could not follow since you cannot invoke geographic characteristics when you have spatially constant model parameters. The only source of spatial variability is in the model forcing. This is mentioned but only later on page 26

line 5, maybe you could reformulate this in a way that avoids such a misunderstanding.

Line 7-9 page 26: note that ET is also influenced by Rnet which might also be less spatially coherent.

Line 11-12 page 27: and there can be no SM retrieval in snow-covered regions.

Line 15-18 page 27: solid/liquid phase transitions in soil moisture layers are another type of neglected effect relevant in the study domain as mentioned in the main comment.

Line 10 page 28: the fact that snowpack anomalies are "erased" each summer and partially transferred to soil moisture through snow melt also largely explains this pattern. Hence, soil moisture also by construction allows for a longer memory than snowpack. This could be made more prominent in the discussion as well.

Line 11-12 page 28: I would not qualify this as "diverging" since the sign is still the same (Figure 9). The non-snow storage only becomes less dominant when spatially integrated.

Line 23-25 page 28: adding this to the abstract would really explain better what you mean with the cryptic ending on line 32-34 page 1.

---

## Referee Comment (RC2) · Anonymous Referee #2 · 26 Feb 2018

The main objective of this study is to analyse the spatial and temporal variability of snow pack and liquid water (mainly soil moisture) components at mid to high Northern latitudes and their respective contributions to total water storage (TWS) variations. To do so, a parsimonious hydrological model adapted to this purpose was first developed and calibrated using Earth Observations datasets, including TWS, snow water equivalent (SWE), evapotranspiration and gridded runoff. A comprehensive description of the model is provided in the Supplementary Material and a rather deep analysis of the calibration procedure is proposed. The authors also made a great effort in analysing the performances of the model at different time scales (seasonal and interanual) and spatial scales (grid pixel and whole domain). Then the model is used over the 2000-2014 time period to evaluate the contribution of solid and liquid water components to

[Figure]

TWS variations at these different spatio-temporal scales. Main conclusions are that TWS variations are mainly driven by snow dynamics at seasonal scales, while liquid water dominates TWS variations at interanual scales. Before concluding, the authors discuss some limitations of the method.

The paper is well written and organized. The conclusions are consistent with results presented all along the manuscript. The analysis of the calibration results (in terms of parameter values) is appreciable. Also appreciable is the comparison of the model to state-of-the-art global hydrological models from the eartH2Observe project, showing that despite its simplified structure, the current model performs reasonably well. I have only one major comment and some minor remarks and suggestions developed in the following.

Major comment:

The need to develop a new model is not clearly stated, which is of prior importance since a large part of the paper is devoted to the presentation/validation of this model and the model outputs are used to draw the conclusions. Namely: - why not using existing models that show comparable performances and include more processes? - why not directly compare TWS and SWE from observations used here to calibrate the model? In that case, do the conclusions remain unchanged?

Minor comments:

P1L20: "... observed hydrological spatio-temporal patterns..."

P2L32: Some models explicitly simulate the upper soil layer using a multi-layer scheme (e.g., the ISBA land surface model, Decharme et al., 2011). In that case, satellite-derived soil moisture can be compared to model outputs, and even assimilated with positive impacts on the model performances (Albergel et al., 2017).

Decharme, B., Boone, A., Delire, C., & Noilhan, J. (2011). Local evaluation of the Interaction between Soil Biosphere Atmosphere soil multilayer diffusion scheme using

four pedotransfer functions. Journal of Geophysical Research Atmospheres, 116(20), 1–29.

Albergel, C., Munier, S., Leroux, D. J., Dewaele, H., Fairbairn, D., Barbu, A. L., . . . Calvet, J.-C. (2017). Sequential assimilation of satellite-derived vegetation and soil moisture products using SURFEX_v8.0: LDAS-Monde assessment over the Euro-Mediterranean area. Geoscientific Model Development, 10(10), 3889–3912.

P4L5: Which datasets are used to mask out such pixels?

2.2 Model description: if I understand correctly, incoming water from upstream grid cells are not accounted for. At the monthly time scale, I agree that this would be negligible at the pixel scale, but is it still true at the basin scale (e.g., the Ob river basin)?

P6L8: ". . . daily cumulated gridded precipitation. . ."

P6L11: ". . . that combines remotely-sensed precipitation. . ."

P7L2-4: Are these data assimilated into a snow model?

2.3 Input Data: Since EO uncertainties are an important aspect of the calibration process, I suggest the authors to add a figure showing maps of temporal averages of each uncertainties for each dataset. This could help interpreting the model performances as shown in Figures 3, S1 and S2.

P10L8: "Therefore. . ."

P11L4-9: This paragraph is unclear. Is it related to the smoothness of GRACE spatial patterns? In that sense, I think that for a better comparison with GRACE, modelled TWS should be first processed to remove high frequency spatial variability that is not observed by GRACE.

P11L22-24: Is the overestimation found by Behrangi et al. (2016) and Swenson (2010) quantitatively comparable to this study?

P11L26: "... if SWE > 80 mm (parameter snc)."

3.1 Model optimization: It would be interesting to discuss the values of the four cost terms in Eq. (2) obtained with the optimized parameters.

P14L12: Are "seasonal variations" equal to the "mean seasonal cycle"? We understand after (from the figures) that yes. In this case, very high correlation values are not really surprising. Bias and RMSE would be more suited.

Figure 3: It seems from figure 3(d) that large RMSEs are found in regions affected by the Postglacial Rebound (Eastern Canada and Scandinavia) and near coastlines (ocean signal contamination?).

P19L5: Do the authors have any possible explanation of the large negative anomaly in 2003 and why it is not captured by the model?

P23L4: The average value does not show that CR is positive over the entire domain.

P23L14: "... less variable at interanual time scale..."

---

## Author Comment (AC1) · 9 Apr 2018

We very much thank the anonymous referee #2 for the helpful comments and suggestions on our paper. Please find the author's response in the following.

\*Major Comment\*

**NEED TO DEVELOP A NEW MODEL**

The need to develop a new model is not clearly stated, which is of prior importance since a large part of the paper is devoted to the presentation/validation of this model and the model outputs are used to draw the conclusions. Namely: - why not using existing models that show comparable performances and include more processes? - why not directly compare TWS and SWE from observations used here to calibrate the

model? In that case, do the conclusions remain unchanged?

The reviewer made a good point that developing a new model should be better justified. As the reviewer suggested, existing models could have been used for our study in principle. We chose to implement our own version of a parsimonious model of water cycle processes which shares the common conceptualization of existing models and represents a recombination of established process formulations for conceptual and methodological reasons. Conceptually, simulations of simple models are advantageous with regards to interpretation and understanding of the responses. Furthermore, we think it is also useful to confront results of a simple model informed by observations with more complex and more 'physically-based' ones to elucidate the added value of increased model complexity or possibly to understand where the model requires more comprehensiveness. From a methodological point of view, the model-data fusion approach requires that the underlying model is parsimonious with respect to a) identifiability of model parameters, and b) computational tractability as thousands of simulations need to be performed during the model optimization. Unfortunately, to our understanding, both considerations are hardly achievable using most of the existing models. Additionally, the design is tailored by the globally available data and kept simple possible to provide the opportunity to identify the effect of the inclusion of the different data sets. To address this comment in the manuscript we now state in the introduction:

In this study, we therefore aim to investigate the contributions of snow compared to other (liquid) water reservoirs to spatio-temporal variations of TWS in the northern mid-to-high latitudes. To do so, we establish a model-data-fusion approach that integrates multiple Earth Observation based data streams including GRACE TWS along with estimates of snow water equivalent, evapotranspiration and runoff into a rather simple hydrological model. This model is designed as a combination of standard model formulations yet aims to maintain a low complexity in order to facilitate multi-criteria calibration and to focus on variables that can be constrained by observations. First, we explain the applied methods including the implemented model, the used data, [...]
Regarding the second question, a direct comparison of TWS and SWE from observations is an interesting suggestion that we considered thoroughly. There are two three obstacles with respect to the suggested analysis: 1) the SWE data from GlobSnow suffer from a saturation effect above SWE values of about 100mm. This causes that these systematic errors in the snow data directly propagate to and corrupts the inferred 'liquid' storage component if the difference to GRACE-TWS is calculated. 2) There are frequent gaps in the SWE data which can be either due to the absence of snow or missing data. The suggested analysis would therefore be biased to respective grid cells and times without gaps and could not yield a representative picture. 3) Besides errors in GlobSnow SWE that propagate to the inferred 'liquid' water storages, errors and uncertainties of the GRACE TWS transfer to the 'liquid' water storage as well. We concluded that a joint interpretation of GRACE-TWS and GlobSnow SWE within an appropriate model-data fusion approach as done in this manuscript is preferred. Nevertheless, we performed our analysis using GlobSnow SWE and GRACE TWS, as the referee suggested. We calculated liquid water as the difference between GRACE TWS and GlobSnow SWE and then compared the model results using the same data points as available from the observations (Fig. 1-3). On the interannual scale, we obtain similar conclusions when directly using the observations and when using the model. For the mean seasonal cycle, the main pattern persists as well, yet conclusions differ in some regions that likely suffer from saturation in GlobSnow SWE (e.g. Kamchatka) or in regions where permafrost and wetlands play a role (e.g. East Siberia). As the latter are not observed by GlobSnow, their contribution to observed TWS is included in calculated W. Additionally, the magnitude in GRACE TWS anomalies is in general much larger than the magnitude of GlobSnow SWE, and thus the magnitude in W based on these observations is larger than the magnitude in modelled W. Therefore, the relative contribution to TWS variability based on observations is shifted towards larger effect of liquid water storages as compared to the modelled results (Fig. 3).

\*Minor Comments\*
**USE OF SATELLITE DERIVED SOIL MOISTURE**

P2L32: Some models explicitly simulate the upper soil layer using a multi-layer scheme (e.g., the ISBA land surface model, Decharme et al., 2011). In that case, satellite derived soil moisture can be compared to model outputs, and even assimilated with positive impacts on the model performances (Albergel et al., 2017).

The authors are thankful for the reviewer's suggestion. We are considering the use of a multi-layer soil scheme with potential to assimilate satellite-derived soil moisture for future efforts, especially at the global scale, and mention this as outlook and potential improvement in the discussion of the revised manuscript.

**DELINEATION OF THE STUDY AREA**

P4L5: Which datasets are used to mask out such pixels?

We thank the reviewer for pointing out that our manuscript was missing this critical information, which will be included the revised manuscript:

We defined humid land surface based on an aridity index AI  $\geq$  0.65. Therefore we calculated AI as the ratio of precipitation and potential evapotranspiration (United Nations Environment, 1992), using the precipitation and potential evapotranspiration data that were also applied as model forcing (GPCP-1DD precipitation (Huffman et al., 2000) and potential evapotranspiration following the Priestley-Taylor formula based on CERES net radiation (Wielicki et al., 1996) and CRUNCEP v6.1 air temperature (Viovy, 2015)). To mask out grids with > 90 % permanent snow cover and > 50 % water fraction, we applied the SYNMAP land cover classification (Jung et al., 2006). This dataset has an original resolution of 1 km and was used to determine the fraction of land cover classes within each 1° x 1° grid cell.

Huffman, G. J., Adler, R., Morrissey, M. M., Bolvin, D., Curtis, S., Joyce, R., McGavock, B., and Susskind, J.: Global Precipitation at One-Degree Resolution from Multisatellite Observations, Journal of Hydrometeorology, 2, 36-50,
2000. Jung, M., Henkel, K., Herold, M., and Churkina, G.: Exploiting synergies of global land cover products for carbon cycle modeling, Remote Sensing of Environment, 101, 534-553, https://doi.org/10.1016/j.rse.2006.01.020, 2006. United Nations Environment, P.: World atlas of desertification / UNEP, United Nations Environment Programme, Accessed from http://nla.gov.au/nla.cat-vn624121, Edward Arnold, London ; Baltimore, 1992. Viovy, N.: CRU-NCEPv6.1 Dataset, http://dods.extra.cea.fr/data/p529viov/cruncep/, 2015. Wielicki, B. A., Barkstrom, B. R., Harrison, E. F., Lee, R. B. I., Smith, L. G., and Cooper, J. E.: Clouds and the Earths Radiant Energy System (CERES): An Earth Observing System Experiment, Bulletin of the Amercian Meteorological Society, 77, 853-868, 1996.

**ROUTING**

2.2 Model description: if I understand correctly, incoming water from upstream grid cells are not accounted for. At the monthly time scale, I agree that this would be negligible at the pixel scale, but is it still true at the basin scale (e.g., the Ob river basin)?

As the reviewer pointed out, routing effects can be significant for large basins, especially in humid regions, which manifests in differences between surface runoff and river discharge at a given location. To address this, we do not use measured river discharge of large basins in our model-data fusion approach, but rather monthly runoff estimates for the European region at grid scale. Regarding the effect of river routing on TWS, e.g. Kim et al. (2009) showed that the contribution of river storage to total TWS anomalies can be significant in the downstream regions of large basins. In northern high latitude catchments, this contribution is relatively smaller compared to continental tropical basins with large floodplains. Thus, we assume that although river storage is not explicitly represented in our model, the associated delay in surface runoff is sufficiently implicitly lumped into the delayed response of land runoff. Therefore, as the reviewer suggested, the effects of routing on the findings of this study can be expected to be small. We have clarified this in the revised manuscript by adding the following:
Model Description: The model does not account for lateral flow of water among grid cells and does not consider river routing explicitly. While the effect of the routing can be significant in large river basins of humid regions (Kim et al., 2009), it is negligible on the spatial scale of a grid cell (as also shown by small influence of delayed storage component), and at the temporal scale of monthly aggregated values. To ensure that the model calibration is not affected by river routing, we do not compare simulations to measured river discharge of large basins in our model-data fusion approach. Limitations of the approach: [..], as well as lateral flow from one grid cell to another. Especially in the downstream areas of large basins the latter represents a potential input that may significantly affect total TWS (Kim et al., 2009), and thus may propagate to the discrepancy between TWSobs and TWSmod in some regions.

Kim, H., Yeh, P. J. F., Oki, T., and Kanae, S.: Role of rivers in the seasonal variations of terrestrial water storage over global basins, Geophysical Research Letters, 36, doi:10.1029/2009GL039006, 2009.

**GLOBSNOW DATA**

P7L2-4: Are these data [observed snow depth and radar data] assimilated into a snow model?

Yes, to our understanding the GlobSnow SWE processing applies a semi-empirical snow emission model and an assimilation scheme to produce maps of SWE estimates based on observations from passive microwave remote sensing and weather station observations (Luojus et al., 2010). We state this in the revised manuscript.

Luojus, K., Pulliainen, J., Takala, M., Lemmetyinen, J., Derksen, C., and Wang, L.: GlobSnow Snow Water Equivalent (SWE) Product Guide, ESA, 2010.

MAPS OF TEMPORAL AVERAGE DATA UNCERTAINTIES

2.3 Input Data: Since EO uncertainties are an important aspect of the calibration process, I suggest the authors to add a figure showing maps of temporal averages of each
uncertainty for each dataset. This could help interpreting the model performances as shown in Figures 3, S1 and S2.

This is a helpful comment for making the manuscript comprehensive. We include the maps of the temporal averages of the uncertainty of observed TWS, ET and Q that are used for model calibration in the supplement of the revised manuscript (see belowFig. 4). As we apply a constant average uncertainty of 35 mm (as mentioned in line 26 page 9), an additional map is not included in the supplement.

**PARAGRAPH OF SPATIAL COHERENCE**

P11L4-9: This paragraph is unclear. Is it related to the smoothness of GRACE spatial patterns? In that sense, I think that for a better comparison with GRACE, modelled TWS should be first processed to remove high frequency spatial variability that is not observed by GRACE.

We see that the methodological paragraph on compensatory effects and spatial coherence was not clear enough. It is not related to the smoothness of GRACE spatial patterns but is meant to provide background information for the analyses to explain the different importance of TWS components to the total TWS across different spatial scales (local grid scale vs. spatially aggregated). We have revised the paragraph accordingly:

As this study intends to analyze the effects of storage components on TWS at different spatial scales (local grid scale and large (regional) spatial averages), the difference in spatial heterogeneities of these components has to be considered. Some storage components, e.g., soil moisture, have much larger spatial variability than others. Due to this large small-scale heterogeneity, the effect on larger regional scale might actually be minimal, as different local scale heterogeneities compensate each other when the regional averages are calculated (Jung et al., 2017). Thus, we assessed the spatial coherence of simulated patterns of SWE and W by calculating the proportion of total positive and total negative covariances among grid cells (Eq.(4,5) in Jung et al. (2017)).

HESSD
If the sum of positive covariances outweighs the sum of negative covariances, it implies some degree of spatial coherence of the anomalies. Spatial coherence of anomalies then causes a larger variance of the averaged anomalies compared to the sum of the variances of individual grid cells. This assessment of spatial coherence of SWE and W anomalies allows for understanding different contributions of SWE and W to TWS variability at local scale compared to the regional scale.

Jung, M., Reichstein, M., Schwalm, C. R., Huntingford, C., Sitch, S., Ahlstrom, A., Arneth, A., Camps-Valls, G., Ciais, P., Friedlingstein, P., Gans, F., Ichii, K., Jain, A. K., Kato, E., Papale, D., Poulter, B., Raduly, B., Rodenbeck, C., Tramontana, G., Viovy, N., Wang, Y. P., Weber, U., Zaehle, S., and Zeng, N.: Compensatory water effects link yearly global land CO2 sink changes to temperature, Nature, 541, 516-520, 10.1038/nature20780, 2017.

**OVERESTIMATION OF GPCP**

P11L22-24: Is the overestimation found by Behrangi et al. (2016) and Swenson (2010) quantitatively comparable to this study?

The reviewer pointed to an interesting comparison that was missing in the original manuscript. Due to the mismatch in the spatial and temporal domain, it yet is difficult to quantitatively compare the results reported in Behrangi et al. (2016) and in Swenson (2010) with this study in a precise manner. However, Behrangi et al. (2016) showed that average high-latitude annual precipitation of GPCP is 20 % higher compared to other precipitation products, and Swenson (2010) state that the GPCP undercatch correction is too large, resulting in too much cold season accumulation. This suggests that reducing GPCP snow fall in our study by 33 % seems quantitatively comparable. We include this statement in the revised manuscript.

Behrangi, A., Christensen, M., Richardson, M., Lebsock, M., Stephens, G., Huffman, G. J., Bolvin, D., Adler, R. F., Gardner, A., Lambrigtsen, B., and Fetzer, E.: Status of high-latitude precipitation estimates from observations and reanalyses, Jour-
nal of Geophysical Research: Atmospheres, 121, 4468-4486, 10.1002/2015jd024546, 2016. Swenson, S.: Assessing High-Latitude Winter Precipitation from Global Precipitation Analyses Using GRACE, Journal of Hydrometeorology, 11, 405-420, 10.1175/2009jhm1194.1, 2010.

**VALUES OF THE 4 COST TERMS**

3.1 Model optimization: It would be interesting to discuss the values of the four cost terms in Eq. (2) obtained with the optimized parameters.

We agree with the reviewer that the individual contributions to the total cost are a relevant methodological aspect. We have included a respective table in the supplement of the revised manuscript for completeness. To keep the manuscript concise, we will not add excessive discussion to this methodological detail, in particular since we present and discuss the evaluation of the model simulations against the individual data streams quite extensively in the manuscript.

Table S1 (Fig. 9) shows the cost terms achieved with the default and the optimized parameter set. Compared to the default parameter values, total costs clearly improve after calibration. The magnitude of the optimized values in general reflects the qualitative importance we assign to the individual data streams (as large cost values 'punish' the model during optimization), with the highest value for TWS, followed by SWE and Q and the smallest value for ET. These values represent a weighted Nash-Sutcliff efficiency of 0.37 (TWS), 0.44 (SWE), 0.57 (Q) and 0.80 (ET).

**CORRELATION FOR SEASONAL VARIATIONS**

P14L12: Are "seasonal variations" equal to the "mean seasonal cycle"? We understand after (from the figures) that yes. In this case, very high correlation values are not really surprising. Bias and RMSE would be more suited.

The reviewer is correct, 'seasonal variations' are used synonymously to 'mean seasonal cycle'. This has been clarified in the revised manuscript. We will also follow the
suggestion of the reviewer and include bias and RMSE metrics of ET and Q in the supplement of the revised manuscript.

**REGIONS OF LARGE RMSE OF TWS**

Figure 3: It seems from figure 3(d) that large RMSEs are found in regions affected by the Postglacial Rebound (Eastern Canada and Scandinavia) and near coastlines (ocean signal contamination?).

Yes, the reviewer is right, large RMSEs tend to occur in regions affected by Postglacial Rebound and near coastlines where the signal potentially is contaminated by the ocean. These limitations and errors of the GRACE TWS estimates are referred to in line 6-8 page 15 and are stated in relation to high RMSE in line 10-11 page 15:

[...] Second, although GRACE TWS passed through various pre-processing steps, the models to account e.g. for postglacial rebound or leakage between neighbouring grid cells introduce their own uncertainties and do not remove the effects completely. [...] This together is reflected in higher RMSE in arctic regions (e.g. surrounding the Hudson Bay), as well as in heterogeneous coastal and mountainous regions.

**NEGATIVE TWS ANOMALY IN 2003**

P19L5: Do the authors have any possible explanation of the large negative anomaly in 2003 and why it is not captured by the model?

This is a very interesting point, which we investigated further. If we isolate interannual variations by removing the trends in GRACE and modelled TWS the agreement with respect to 2003 gets substantially better, as indicated by higher correlation scores (Fig. 5). This suggests that the trend in GRACE TWS is to some extent either subject to observational issues or represents a process that is not captured by our and the earthH2Observe models, which don't reproduce the 2003 anomaly adequately either (see FigS4). In addition, there is a negative SWE anomaly of on average 5 mm (see Fig.4 a) indicated in the GlobSnow data, that is not captured by our model, suggesting Interactive comment
an issue with the precipitation forcing data. This not captured SWE anomaly appears to explain the remaining difference to the GRACE TWS anomaly in 2003 after detrending. The reason why this snow anomaly is not captured by the forcing remains unclear at this point – it persists when using the WFDEI forcing data set (Fig. 6). While the model reproduces the spatial pattern of the 2003 interannual TWS variability, the magnitude of observed TWS, especially in North America, is not captured by the forcing and thus by the model, either (Fig. 7). We add a paragraph to the revised manuscript on the discrepancy regarding the 2003 anomaly.

**AVERAGE VALUE OF CR**

P23L4: The average value does not show that CR is positive over the entire domain.

We agree with the reviewer here and add quantitative labels of CR to the respective figures in the revised manuscript.
**Contribution of SWEobs & Wobs to TWSobs MSC Contribution of SWEmod & Wmod to TWSmod MSC**

**Fig. 1.** Relative contribution based on CR of snow (SWE) and liquid water (W) storage anomalies to mean seasonal TWS anomalies based on observations (left) and based on the model (right)

---

## Author Comment (AC2) · 9 Apr 2018

On behalf of all co-authors we very much thank Vincent Humphrey for very thoughtful and constructive comments. We appreciate the details and clarity in his remarks, and we have addressed all major and minor comments in the following. Further suggestions regarding terminology, clarity of formulations and figures were gratefully received and will be included in the revised manuscript.

*Major Comments*

PHASE LAG OF SEASONAL TWS

In their results, the authors find that the modeled seasonal cycle of TWS has a systematic lag compared to observations (model TWS preceding observed TWS). This

lag is also present in other models from the Earth2Observe ensemble. The analysis of the authors convincingly shows that their modeled snow storage seems to have a correct phase and is therefore not responsible for this lag between modeled and observed TWS. They also mention that adding delayed storage responses (as e.g. with a groundwater module) could not correct this effect either. I find this a major finding for the research community (which could be made more prominent in the conclusions) since it is often supposed that such model errors mainly stem from the lack of long memory water storages and poor representation of snow dynamics. Here the authors conclude that neither of these seem responsible and that the origin of the phase lag in TWS must reside elsewhere, which brings me to my main suggestion below. One important limitation that the authors fail to mention is that there is no consideration of permafrost and liquid/solid phase transitions of soil moisture content. In the proposed model, soil moisture does not have temperature neither does it store energy. In reality, it is well known that freeze/thaw dynamics are also a dominant factor for water and energy fluxes in high latitudes. Freeze/thaw is the on and off switch for evapotranspiration and vegetation growth. However, a phase lag between the availability of energy and the ET response cannot be modeled with the current model setup (the alpha parameter only conditions ET amplitude). Potentially, a lot of ground heat flux might be required before ET can actually take place. In addition, from my understanding of the equations presented the supplementary material, actual ET is not reduced in the case of snow cover neither is it dependent on vegetation growth. This might introduce a too early response of ET to net radiation compared to reality, leading to a fast rise of soil moisture depletion already in early spring. Later, soil moisture would become limiting already in mid-summer and ET would peak in June and start to reduce already in July (Fig S1). The reference below suggests a peak of vegetation growth in August for a boreal forest (from one FLUXNET site). The authors might consider exploring this direction and maybe check whether there is some evidence that FLUXCOM ET itself (the observational constraint) already contains such a phase lag. As this would require some additional work, it would also be fine if the authors prefer to simply mention this

as a hypothesis to explore.

We are grateful for the suggestions and a detailed explanation of the potential causes of the systematic lag in the modeled TWS.

*Biases in ET and its effect on TWS: We have explored the relationship of potential biases in ET that may lead to a different timing in peaks of TWS. As already mentioned in the manuscript, we do not have ground heat flux and vegetation growth processes in the current model formulation, but we gratefully acknowledge it as an interesting opportunity for future investigations. In the current model, as the review correctly points out, actual ET is not reduced in the case of snow cover, which may lead to an early reduction of soil moisture and, consequently, TWS. To assess this effect, we scaled ET with the snow free fraction of a grid cell (1-FSC). Using the optimized parameter set presented in the manuscript together with the new scaling formulation of ET, there was a slight reduction of simulated ET in spring and a corresponding increase in July. This lead to a slight improvement of TWS timing (Fig. 1). On the other hand, when the model variant with snow cover scaling factor was optimized again, the marginal gain of performance was reduced (Fig. 2). This suggest that the observation data streams guide the model to optimal parameter values that would still result in the lag in the TWS. As a further check, we post-adjusted the simulated TWS with biases in ET simulation, representing the perfect ET simulation, but even that adjustment in the TWS was not enough to improve the lag in TWS.

*Permafrost and TWS variation: As the reviewer points out, our model does not consider the permafrost dynamics. In order to identify the potential associations of the lag in TWS simulations against occurrences of permafrost, we compared the lag against permafrost fraction from the circum-Arctic map of permafrost and ground ice conditions (Brown et al., 1997). There is a tendency that the regions with the largest negative lag have a higher permafrost fraction (Fig. 3). This is especially visible in regions with sporadic permafrost (smf, slr, shr), as well as isolated patches of permafrost with high ground extent and thick overburden (ihf). One can expect that the sporadic permafrost

is more active and may have larger influences in seasonal storage dynamics than more 'permanent' and larger permafrost. However, it should be noted that the ranges of permafrost fractions are large for both, short and long lags of TWS, suggesting a complex interaction between permafrost extent and its effect on lag in seasonal TWS dynamics as well as possible other factors related to the lag.

We include the main findings of the above two analysis on potential relationships between the lag in TWS and biases in ET and the effects of permafrost and freeze/thaw dynamics. We further highlight the limitation that some potentially relevant processes are not yet accounted for in the current model setup and add the following paragraph in the discussion of the revised manuscript.

Performance of the spatially integrated simulations: [. . .] The lag in TWS simulation can occur due to several mechanisms and processes that are not yet considered in the current model structure such as lateral flow and surface storages (wetland and lakes), vegetation processes, glacier melt, and human influence with dams and reservoirs. However, we don't observe a general and a systematic relationship with either elevation, land cover type, soil properties, and the occurrence of lakes and wetlands. There is a tendency that larger negative lags occur more frequently in regions with sporadic permafrost, but the ranges of permafrost fractions are large for both, short and long lags in TWS, suggesting a complex interaction between permafrost extent and its effect on lag in seasonal TWS dynamics. Finally, potential biases in timing of ET due to snow cover and/or vegetation processes may also affect the timing of depletion of SM and TWS. Additionally, high uncertainties of the precipitation forcing and GlobSnow SWE [. . .]

Limitations of the approach: [. . .] Other simplified or ignored hydrological processes include the coincident occurrence of rain and snow fall, liquid water capacity of snow, interception, freeze/thaw dynamics within the soil, capillary rise and other surface-groundwater interactions, the effect of vegetation growth, as well as lateral flow from one grid cell to another.

MISLEADING TERMINOLOGY SNOW VS. LIQUID WATER

The separation of TWS into liquid and snow water seems a bit misleading since the liquid phase might implicitly include some frozen water as well (frozen soil moisture). As mentioned by the authors, there is a mismatch between explicitly represented processes and observed processes (TWS includes frozen water) that may be compensated by adjustments in model parameters. The expression "liquid phase" is hence misused in my opinion and might very well lead to confusion. It might be more accurate to refer to snow versus non-snow changes as done for example in page 27 line 30. I think this terminology should be extended to the rest of the manuscript.

The reviewer makes a valid point that the terminology might be misleading, especially with regards to observation. In reality, some part of TWS also includes solid or frozen water. However, in our study, the terminology of 'snow' vs 'liquid water storages' are used in the context of model simulation in which we do not account for frozen water storages. In order to avoid misunderstanding, we elucidate that liquid water storages might implicitly include frozen water especially in the observation.

[. . .] The amount of water storages in retained land runoff (RW) and SM represents the liquid water storage (W). Frozen water, e.g. in soil, is not explicitly included in the model, yet might implicitly be accounted for in W after model calibration.

EFFECT OF PRECIPITATION FORCING

Figure S7 is quite pre-occupying because it suggests a dependency of your results on the forcing dataset. For instance, the difference might be related to your partitioning between snowfall and rainfall (which was not applied when using WFDEI). One possibility to check if this comes from uncertainty in the precipitation data might be to compare the regional mean time series of the two products and look for large differences in 2005 and 2010. This would also indicate whether GPCP-1DD appears superior to WFDEI. In relation to this -> Line 11-12 page 26: this is a rather unsubstantiated statement. Please give it more weight, for instance by replicating key figures (e.g. Fig 9) in the

supplementary material.

We thank the reviewer for pointing to Figure S7. Doing the analysis that he suggested, we found that in Figure S7 of the submitted manuscript the shown time periods of TWS forced by WFDEI were shifted relative the observations and the modelled TWS based on GPCP forcing. The updated results (Fig. 4) are included in the revised manuscript. The use of different precipitation forcing results in marginal difference in TWS simulations. TWSmodWFDEI shows a larger seasonal amplitude because the amount of wintertime precipitation (snowfall and rain fall) is higher, while summertime precipitation is lower than estimated by GPCP (Fig. 5). However, the key findings of the dominant storage component remain the same (Fig. 6). We include Figure 6 in the supplement of the revised manuscript.

CLARITY OF ABSTRACT

You could make lines 24-29 of your abstract clearer. Upon first reading, I understood that snow dynamics dominate IAV on a large scale, which is not the case. It should be clearly said that "liquid water" dominates IAV at all spatial scales while snow dominates MSC at all spatial scales (Fig. 9). In addition, for IAV, the relative influence of snow increases with spatial aggregation due to the spatial coherence of T, a main driver of snowfall and snow melt. The wording "liquid water storages, comprising mainly of soil moisture" is also a bit misleading. It is not really clear what is implicitly incorporated in the soil moisture reservoir in order to fully reproduce TWS (as mentioned in the third comment). Güntner et al. 2007 provides a similar analysis based on WaterGAP. This would be an interesting point of comparison since they indicate a contribution for IAV of 33% snow, 27% soil and 12% groundwater and 28% surface water (!) for cold climates (their table 5). I think this reference should be discussed and compared with your results.

We thank the referee for highlighting this lack of clarity in the abstract. We revise the abstract accordingly:

Consistent with previous studies, we show seasonal TWS variations are controlled by snow dynamics across all spatial scales in the northern mid-to-high latitudes. In contrast, we find that inter-annual TWS variations are dominated by liquid water storages across all spatial scales. The relative contribution of snow to interannual TWS variations, though, increases when the spatial domain over which the storages are averaged becomes larger. This is due to a stronger spatial coherence of snow dynamics, that are mainly driven by temperature, as opposed to spatially more heterogeneous liquid water anomalies, that cancel out when averaged over a larger spatial domain.

Further, as the referee suggested, we include a comparison with the results of Güntner et al. 2007 in the discussion of the revised manuscript.

*Minor Comments*

MODELLING FRAMEWORK

Your work is very new and promising in the sense that multiple remote sensing or machine-learning observation-derived products are used simultaneously for calibrating a hydrological model. This is not easy to do and a research direction worth to explore. The overall modeling framework however still relies on a very standard land surface model structure. One missed opportunity may be to have used these observational datasets not only to calibrate model parameters, but also to identify functional relationships directly from the data (as opposed to fitting the parameters of a pre-defined equation to the data). Such research might be suggested as one possible future direction in the discussion. Finally, the paper does not emphasize on the added value of using remote sensing products to constrain the model (except for a lower RMSE against observations, which is somewhat expected since other models were not calibrated with these observations). Could similar results have been obtained with the Earth2Observe ensemble? (especially on IAV?) I f not, this would better show the merit and relevance of the presented approach.

The referee is right in that our modelling framework still relies on a standard land sur-
face model structure in terms of which processes are included. We yet want to highlight that this modelling framework still allows for more flexibility in the responses because we do not strictly constrain the model parameters that are often fixed in land surface models. We agree that identifying functional relationships directly from the observations represents new challenges in modelling the Earth system, especially when the modelling community shifts towards a hyper-resolution modeling, for which the classical formulation at coarse resolutions might not be valid. With the current availability and inconsistencies in the observational data, we could not address the challenge in the current study. As pointed out, use of remote sensing data is advantageous in constraining the model over much larger spatial domains than using site-level or discharge measurements and thus improves the confidence in model results. The improved confidence, also reflected in the presented better performance metrics is also an important merit compared to the EartH2Observe ensemble. Despite, the lower performance metrics, the results of the EartH2Observe ensemble are in general similar to our study. As the referee suggested, we highlight the merits of using remote sensing data, as well as potential future research in identifying functional relationships directly from observations.

Comparison with the eartH2Observe model ensemble: [. . .] Compared to the model simulations in the EartH2Observe ensemble, our modeling framework assimilates information from more data streams, e.g. GRACE and GlobSnow data. Even though we only used a subset of 1000 random grid cells to constrain the model parameters, our model performs better than EartH2Observe ensemble over the whole domain (6050 grids). This improvement in model performance is also consistent among several modelled variables and not limited to storage components only. This suggests that remote sensing data, with larger spatial coverage than site measurements, have a large potential in improving hydrological simulations over a large domain. In addition, remote sensing data also hold potentials beyond the use as an observational constraint and can provide information on identifying and formulating functional relationships across several spatial and temporal scales.

**CLARITY OF MODELLED LIQUID WATER AND MODELLED RUNOFF GENERATION**

Liquid water is explicitly modeled as soil moisture + runoff routing but also likely includes river storage, lakes and wetlands implicitly (e.g. large water holding capacity mentioned on page 12, lines 7 and 19). This could also be made a bit clearer already in the model description in order to avoid some confusion later. Using a snow/non-snow terminology would also help resolving this. It could be made clearer that runoff is currently only generated from infiltration limitation (e.g. no baseflow in Eq. S10). Also mention that this is partially compensated by the recession time scale parameter that delays runoff generated on a specific day. Likely because the model is evaluated at monthly scale, this only has a limited impact on model performance and this model parameter is the least constrained by observations.

Thanks for pointing this out, we adjust the model description in the revised manuscript as following:

As land runoff is generated with an effective infiltration excess formulation, this excess runoff is essentially all the water that cannot be stored in soil water storage, and thus implicitly contains both, surface runoff as well as the percolation to deeper water storages such as groundwater. Therefore, we use an exponential delay function (Orth et al., 2013) to mimic runoff contributions from slow-varying storages, such as groundwater and surface water bodies. After model calibration, this retained land runoff (RW) is supposed to implicitly include the effects of several water pools that are not explicitly represented in the model (groundwater, lakes, wetlands and the river storage). The sum of RW and SM is then taken as the total liquid water storage (W).

**DISTINCTION OF OBSERVATIONAL PRODUCTS**

Methods: You could make a better distinction between purely observational products, and observation-based upscaled products such as Tramontana et al. or Gudmundson et al. which also rely on the quality of the underlying forcing data.

[Figure]

We see that such a distinction could help to underline the dependencies and uncertainties in the observational products. However, for each data product, information on its derivation is included in the original manuscript in the description of the input data. Besides, the line between purely observation based and upscaled isn't always that clear, as e.g. GlobSnow is based on a snow model, satellite data and site measurements, and the GRACE estimates rely on several models for data correction as well.

COVARIANCE OF SWE AND W

Line 29-30, page 10: this assumption seems a bit dangerous given figure 6. Could you please document the degree to which this assumption is correct and if this might affect the results qualitatively (possibly in supplementary information)?

We agree with the referee that the potential implication of this assumption should be discussed. We therefore include a short discussion (see below) on the effect of the covariances between SWE and W on TWS variability in the supplement of the revised manuscript.

Figure S9 compares the contribution of the combined SWE and W variances and the covariance of both storages to the total variance of the spatially aggregated TWSmod. On the interannual scale, 81 % of TWS variability is explained by the variances in SWE and W, suggesting that the covariance between SWE and W only has minor effect. This is underlined by high percentage of SWE and W variance on total TWSmod variance for all grids of the study domain (Fig. S9). On mean seasonal scales, the majority of spatially aggregated TWS variability is still explained by variances in SWE and W, but the contribution of the covariance increases. This can be expected, as the seasonal variation of snow storage affects the subsequent availability of liquid water storages through the snowmelt process. At the local scale, though, the percentage of SWE and W variance on total TWSmod variance remains high in regions where the dominance of either snow or liquid water components are clear (Fig. 7 of the manuscript). In regions where covariances of two storage components is larger, the contribution of two

storage components to TWS variability are similar resulting in a CR value of around 0. Therefore, we conclude that while the covariances of snow and liquid water can be remarkable on the seasonal scale over a large spatial domain, it does not affect or change the dominant components on the TWS.

\*In-Text Comments\*

FLUXCOM ET

Line 10-14, page 7: I thought FLUXCOM was based on an ensemble of machine learning algorithms (e.g. not only random forest). Could you also briefly comment on the performance of FLUXCOM in snow regions and high latitudes? Any idea if FLUXCOM is already accounting for sublimation?

The referee is right, FLUXCOM provides an ensemble of machine learning algorithms, but we only used the products from the random forest variant in this study. Even though FLUXCOM data have not been validated explicitly for snow-dominated regions, the cross validation of ET shows a good performance in most regions (Tramontana et al., 2016). In terms of sublimation processes, FLUXCOM conceptually includes sublimation processes as well, but the confidence in capturing such small fluxes is low due to lower signal to noise ratio in the underlying observations in FLUXNET sites. Therefore, we do not constrain modelled sublimation by FLUXCOM-based ET. We clarify this in the revised manuscript.

The ET product is based on FLUXCOM (www.fluxcom.org), i.e. upscaled estimates of latent energy that were derived by integrating local eddy covariance measurements of FLUXNET sites, remote sensing, and meteorological data using machine learning algorithms (Tramontana et al., 2016). In this study, we apply the Random Forest (Breiman, 2001) realization of FLUXCOM-RS+METEO (see Tramontana et al. 2016 for details). While the product captures seasonality and spatial patterns of mean annual fluxes well, predictions of inter-annual variations remain highly uncertain (Tramontana et al., 2016). In addition, the performance of FLUXCOM ET was found to be lower in

extreme environments that are not well represented by FLUXNET sites such as the arctic. An underestimation in the order of 10–20 % of ET can be expected owing to missing energy balance correction prior to upscaling for this respective FLUXCOM ET realization. To calculate ETobs [mm d-1], we assume a constant latent heat of vaporization of 2.45 MJ m-2.

Tramontana, G., Jung, M., Camps-Valls, G., Ichii, K., Raduly, B., Reichstein, M., Schwalm, C. R., Arain, M. A., Cescatti, A., Kiely, G., Merbold, L., Serrano-Ortiz, P., Sickert, S., Wolf, S., and Papale, D.: Predicting carbon dioxide and energy fluxes across global FLUXNET sites with regression algorithms, Biogeosciences Discussions, 1-33, 10.5194/bg-2015-661, 2016.

INACCURATE SENTENCE

Line 16 page 12: the sentence is inaccurate: a recession time scale of x days does not mean that only runoff of the preceeding x days contributes to "total runoff" (check Orth et al. 2013).

Thank you very much for pointing this out! We change the sentence accordingly to: Finally, the calibrated recession time scale that delays land runoff is 13 days (qt). Compared to much smaller alpine catchments for which Orth et al. (2013) reported qt of 2 days, the longer delay coefficients are reasonable at a large spatial resolution of 1° x 1° grids, because the elevation gradients are much smaller within a large spatial area.

GLOBAL UNIFORM PARAMETER VALUES

Line 13, page 13: maybe not necessary to say that these approaches are not commonly accepted as this might be a subjective statement in my opinion. The arguments you give just before (on overfitting) and the continental-scale focus of your study might be sufficient arguments. Another argument you could mention is that allowing locally varying parameters would contaminate your conclusions: with locally dependent parameters, the differences in local-scale / large-scale contribution to IAV might due to
the spatial dependency of parameters. But with your current setting, they can only be attributed to climate forcing. This is also why it makes a very clean experiment. This last point also calls for one caveat in the conclusion: your picture of the partitioning and scale-dependency of liquid versus snow might also change once you introduce spatial variability of the model parameters (e.g. snow melt factor might be very dependent on the vegetation cover, contrasting the responses of tundra versus boreal forests).

The referee is right, this statement seems to transport a quite subjective opinion, yet it's based on Beck et al. (2016) 'Due to the lack of a commonly accepted approach for parameter regionalization, hydrologic models typically applied at continental to global scales (hereafter called macroscale) rarely use regionalized parameters [. . .]'. Therefore, we reformulate the sentence accordingly:

[. . .] Since such approaches are not commonly accepted, macro-scale models mostly apply a priori parameter values based on empirical relationship or on expert knowledge that may lead to suboptimal model simulations (Beck et al., 2016;Sood and Smakhtin, 2015).

With his last comment, that the conclusions may change if we introduce spatial variability, the referee made a good point that is missing in our discussion. We include this possible caveat when discussing the limitations of the approach on page 27 line 24:

[. . .] Considering the spatial variability of model parameters might affect the relative contributions of different storage components to TWS variability at different spatial scales. However, the comparison with eartH2Observe models, which partly involve spatial heterogeneity in model parameters, suggests that the main conclusions should remain unchanged. Additionally, we want to highlight [. . .]

Beck, H. E., Dijk, A. I. J. M. v., Roo, A. d., Miralles, D. G., McVicar, T. R., Schellekens, J., and Bruijnzeel, L. A.: Global scale regionalization of hydrologic model parameters, Water Resources Research, 52, 3599-3622, 10.1002/2015WR018247, 2016. Sood, A., and Smakhtin, V.: Global hydrological models: a review, Hydrological Sciences

Journal, 60, 549-565, 10.1080/02626667.2014.950580, 2015.

TRUNCATION OF VALUES IN FIGURES

Figure 3. If values were truncated (e.g. Fig3d) this should be indicated in the legend and in labels.

In the legend of figures in the revised manuscript, we indicate if values were truncated.

QUANTITAVE LABELS IN FIGURES SHOWING CR

Figure 7: It would be nice to add units to the colorbars (in addition to qualitative labels), same in Figure 8.

We add quantitative labels of CR in the revised manuscript.
* * *
[Figure]

**Fig. 1.** Comparison of MSC and IAV of ETobs, ETmodorig (as in manuscript), ETmodETscal (parameter as in manuscript but actETscaled) and ETmodETscalOPTIMIZED (actET scaled and optimized)

[Figure]

**Fig. 2.** Comparison of MSC and IAV of TWSobs, TWSmodorig (as in manuscript), TWSmod-ETscal (parameter as in manuscript but actETscaled) and TWSmodETscalOPTIMIZED (actET scaled and optimized)

[Figure]

**Fig. 3.** TWS phase lag compared to the permafrost fraction of the grid cell (colors relate to the TWS lag class)

[Figure]

**Fig. 4.** Comparison of the mean seasonal cycle and interannual variability of TWSobs (GRACE), TWSmod (forced with GPCP precipitation) and TWSmodWFDEI (forced with WFDEI rain and snow fall)

[Figure]

**Fig. 5.** Comparison of the mean seasonal cycle and interannual variability of rain fall and snow fall from WFEI product and from GPCP (snow fall as in the optimized model)

[Figure]

**Fig. 6.** Relative contribution of snow (SWE) and liquid water (W) to TWS variability when forced with WFDEI snow and rainfall on different spatial and temporal scales

[Figure]

**Fig. 7.** Percentage composition of spatially aggregated TWSmod variance from the combined variances of SWE and W, and two times the covariance of SWE and W on mean seasonal and interannual scales

[Figure]

**Fig. 8.** Percentage of SWE and W variance on total TWSmod variance on mean seasonal (MSC) and interannual (IAV) scales

---

## Author Response (AR1)

**Author Response to Reviews of:**

**Understanding terrestrial water storage variations in northern latitudes across scales**

Tina Trautmann[1,2], Sujan Koirala[1], Nuno Carvalhais[1,3], Annette Eicker[4], Manfred Fink[5], Christoph Niemann[1,5], Martin Jung[1]
* * *
Dear Prof. Jean-Christophe Calvet,

Dear Vincent Humphrey and Anonymous Referee #2,

we appreciate the positive feedback on our paper and thank the editor and the two reviewers for their constructive comments and suggestions. Please find our detailed response to all comments in the following sections. The corresponding changes in the revised manuscript and supplement are highlighted at the end of this document.

Apart from that, we ask for permission to include a few additional changes that are marked in the revised manuscript as well.

Please, don't hesitate to contact us, if further clarifications are needed.

Kind regards,

Tina Trautmann
on behalf of all co-authors

**1. Referee #1 – Vincent Humphrey**

*RC: This study aims to evaluate the relative contribution of snow versus liquid water in (total) terrestrial water storage changes in northern latitudes. In order to investigate this question, the authors construct a simple bucket-type hydrological model with 10 free parameters which they calibrate against four different datasets (satellite observations of terrestrial water storage from GRACE satellites, snow water equivalent from the GlobSnow product which combines satellite and ground observations, evapotranspiration from FLUXCOM which is based on an ensemble of machine learning methods calibrated with in-situ observations, and E-RUN estimates of gridded runoff also based on a machine learning model calibrated with in-situ observations). Following a short evaluation of the performance of the presented model (and a comparison with the Earth2Observe ensemble of hydrological models), the main results are focused on distinguishing the respective contributions of snow and liquid storages to terrestrial water storage. Their analysis contrasts 1) local scale effects versus a spatially integrated average and 2) the mean seasonal cycle versus inter-annual variability. Consistent with previous studies, the authors find that the seasonal cycle is dominated by the snow component. The main finding of the paper is that liquid water storage clearly dominates inter-annual variability both at local scale and when considering a spatially integrated time series. They also find that the relative contribution of liquid water is weaker for the spatial integral compared to the local scale analysis. The authors argue that because snowpack evolution is primarily dependent on temperature (which has high spatial coherence (fig. 10)), this explains why the relative contribution of snowpack to large-scale inter-annual TWS variability is higher. In their conclusions, the authors comment on the usefulness of a simple hydrological modeling approach informed by multiple observational constraints. They suggest that long-term changes in water availability in northern latitudes might be driven by soil moisture rather than by snow dynamics.*

*This is a really good and well conducted paper. I find the results very interesting and worthy of publication in HESS. One can see that a lot of effort was invested in developing a custom hydrological model and this is reflected by the relatively important share of the methods and model evaluation sections in the paper. However, the authors manage to keep the results and discussion focused around the primary objective of quantifying the relative contribution of snow and non-snow storages to overall water storage variability.*

*I have four major comments/suggestions which I would like the authors to consider as well as some minor comments that are listed below.*

    *AR:* On behalf of all co-authors we very much thank Vincent Humphrey for very thoughtful and constructive comments. We appreciate the details and clarity in his remarks, and we have addressed all major and minor comments in the following. Further suggestions regarding terminology, clarity of formulations and figures were gratefully received and will be included in the revised manuscript.

**\*Major Comments\***

*1) PHASE LAG OF SEASONAL TWS*
*RC: In their results, the authors find that the modeled seasonal cycle of TWS has a systematic lag compared to observations (model TWS preceding observed TWS). This lag is also present in other models from the Earth2Observe ensemble. The analysis of the authors convincingly shows that their modeled snow storage seems to have a correct phase and is therefore not responsible for this lag between modeled and observed TWS. They also mention that adding delayed storage responses (as e.g. with a groundwater module) could not correct this effect either. I find this a major finding for the research community (which could be made more prominent in the conclusions) since it is often supposed that such model errors mainly stem from the lack of long memory water storages and poor representation of snow dynamics. Here the authors*

*conclude that neither of these seem responsible and that the origin of the phase lag in TWS must reside elsewhere, which brings me to my main suggestion below.*

*One important limitation that the authors fail to mention is that there is no consideration of permafrost and liquid/solid phase transitions of soil moisture content. In the proposed model, soil moisture does not have temperature neither does it store energy. In reality, it is well known that freeze/thaw dynamics are also a dominant factor for water and energy fluxes in high latitudes. Freeze/thaw is the on and off switch for evapotranspiration and vegetation growth. However, a phase lag between the availability of energy and the ET response cannot be modeled with the current model setup (the alpha parameter only conditions ET amplitude). Potentially, a lot of ground heat flux might be required before ET can actually take place.*

*In addition, from my understanding of the equations presented the supplementary material, actual ET is not reduced in the case of snow cover neither is it dependent on vegetation growth. This might introduce a too early response of ET to net radiation compared to reality, leading to a fast rise of soil moisture depletion already in early spring. Later, soil moisture would become limiting already in mid-summer and ET would peak in June and start to reduce already in July (Fig S1). The reference below suggests a peak of vegetation growth in August for a boreal forest (from one FLUXNET site).*

*The authors might consider exploring this direction and maybe check whether there is some evidence that FLUXCOM ET itself (the observational constraint) already contains such a phase lag. As this would require some additional work, it would also be fine if the authors prefer to simply mention this as a hypothesis to explore.*

**AR:** We are grateful for the suggestions and a detailed explanation of the potential causes of the systematic lag in the modeled TWS.

- Biases in ET and its effect on TWS:

We have explored the relationship of potential biases in ET that may lead to a different timing in peaks of TWS. As already mentioned in the manuscript, we do not have ground heat flux and vegetation growth processes in the current model formulation, but we gratefully acknowledge it as an interesting opportunity for future investigations. In the current model, as the review correctly points out, actual ET is not reduced in the case of snow cover, which may lead to an early reduction of soil moisture and, consequently, TWS. To assess this effect, we scaled ET with the snow free fraction of a grid cell (1-FSC). Using the optimized parameter set presented in the manuscript together with the new scaling formulation of ET, there was a slight reduction of simulated ET in spring and a corresponding increase in July. This lead to a slight improvement of TWS timing (Fig. 1). On the other hand, when the model variant with snow cover scaling factor was optimized again, the marginal gain of performance was reduced (Fig. 2). This suggest that the observation data streams guide the model to optimal parameter values that would still result in the lag in the TWS. As a further check, we post-adjusted the simulated TWS with biases in ET simulation, representing the perfect ET simulation, but even that adjustment in the TWS was not enough to improve the lag in TWS.

[Figure]

**Figure 1.** Comparison of the mean seasonal cycle and interannual variability of ETobs (FLUXCOM), ETmod$_{orig}$ (as in the manuscript), ETmod$_{ETscal}$ (parameter values as in the manuscript but scaling of actET with snow free fraction of each grid cell) and ETmod$_{ETscalOPTIMIZED}$ (parameter values calibrated for scaling of actET with snow free fraction of each grid cell).

[Figure]

**Figure 2.** Comparison of the mean seasonal cycle and interannual variability of TWSobs (GRACE), TWSmod$_{orig}$ (as in the manuscript), TWSmod$_{ETscal}$ (parameter values as in the manuscript but scaling of actET with snow free fraction of each grid cell) and TWSmod$_{ETscalOPTIMIZED}$ (parameter values calibrated for scaling of actET with snow free fraction of each grid cell).

- Permafrost and TWS variation:

As the reviewer points out, our model does not consider the permafrost dynamics. In order to identify the potential associations of the lag in TWS simulations against occurrences of permafrost, we compared the lag against permafrost fraction from the circum-Arctic map of permafrost and ground ice conditions (Brown et al., 1997). There is a tendency that the regions with the largest negative lag have a higher permafrost fraction (Fig. 3). This is especially visible in regions with sporadic permafrost (smf, slr, shr), as well as isolated patches of permafrost with high ground extent and thick overburden (ihf). One can expect that the sporadic permafrost is more active and may have larger influences in seasonal storage dynamics than more 'permanent' and larger permafrost. However, it should be noted that the ranges of permafrost fractions are large for both, short and long lags of TWS, suggesting a complex interaction between permafrost extent and its effect on lag in seasonal TWS dynamics as well as possible other factors related to the lag.

[Figure]

**Figure 3.** TWS phase lag compared to the permafrost fraction of the grid cell.

We have included the main findings of the above two analysis on potential relationships between the lag in TWS and biases in ET and the effects of permafrost and freeze/thaw dynamics in the revised manuscript. We have further highlighted the limitation that some potentially relevant processes are not yet accounted for in the current model setup.

*2) MISLEADING TERMINOLOGY SNOW VS. LIQUID WATER*

*RC: The separation of TWS into liquid and snow water seems a bit misleading since the liquid phase might implicitly include some frozen water as well (frozen soil moisture). As mentioned by the authors, there is a mismatch between explicitly represented processes and observed processes (TWS includes frozen water) that may be compensated by adjustments in model parameters. The expression "liquid phase" is hence misused in my opinion and might very well lead to confusion. It might be more accurate to refer to snow versus non-snow changes as done for example in page 27 line 30. I think this terminology should be extended to the rest of the manuscript.*

**AR:** The reviewer makes a valid point that the terminology might be misleading, especially with regards to observation. In reality, some part of TWS also includes solid or frozen water. However, in our study, the terminology of 'snow' vs 'liquid water storages' are used in the context of model simulation in which we do not account for frozen water storages. In order to avoid misunderstanding, we have elucidated that liquid water storages might implicitly include frozen water especially in the observation.

*3) EFFECT OF PRECIPITATION FORCING*

*RC: Figure S7 is quite pre-occupying because it suggests a dependency of your results on the forcing dataset. For instance, the difference might be related to your partitioning between snowfall and rainfall (which was not applied when using WFDEI). One possibility to check if this comes from uncertainty in the precipitation data might be to compare the regional mean time series of the two products and look for large differences in 2005 and 2010. This would also indicate whether GPCP-1DD appears superior to WFDEI. In relation to this -> Line 11-12 page 26: this is a rather unsubstantiated statement. Please give it more weight, for instance by replicating key figures (e.g. Fig 9) in the supplementary material.*

**AR:** We thank the reviewer for pointing to Fig. S7 (original supplement). Doing the analysis that he suggested, we found that in Fig. S7 of the submitted manuscript the shown time periods of TWS forced by WFDEI were shifted relative the observations and the modelled TWS based on GPCP forcing. The updated results (Fig. 4) are included in the revised

manuscript. The use of different precipitation forcing results in marginal difference in TWS simulations. TWSmod$_{WFDEI}$ shows a larger seasonal amplitude because the amount of wintertime precipitation (snowfall and rain fall) is higher, while summertime precipitation is lower than estimated by GPCP (Fig. 5). However, the key findings of the dominant storage component remain the same (Fig. 6). We have included Fig. 6 in the supplement of the revised manuscript.

[Figure]

**Figure 4.** Comparison of the mean seasonal cycle and interannual variability of TWSobs (GRACE), TWSmod (forced with GPCP precipitation) and TWSmod$_{WFDEI}$ (forced with WFDEI rain and snow fall).

[Figure]

**Figure 5.** Comparison of the mean seasonal cycle and interannual variability of rain fall and snow fall from WFEI product and from GPCP (snow fall as in the optimized model: if T < 0°C and reduced to 67 % of original GPCP precipitation).

[Figure]

**Figure 6.** Relative contribution of snow (SWE) and liquid water (W) to TWS variability when forced with WFDEI snow and rainfall on different spatial (local grid scale, spatially integrated) and temporal (mean seasonal MSC, inter-annual IAV) scales based on CR (Eq.(3)). The boxplots represent the distribution of grid cell CR, with the dashed line marking the corresponding average. The star represents the CR calculated for the spatially integrated values.

*4) CLARITY OF ABSTRACT*

***RC:*** *You could make lines 24-29 of your abstract clearer. Upon first reading, I understood that snow dynamics dominate IAV on a large scale, which is not the case. It should be clearly said that "liquid water" dominates IAV at all spatial scales while snow dominates MSC at all spatial scales (Fig. 9). In addition, for IAV, the relative influence of snow increases with spatial aggregation due to the spatial coherence of T, a main driver of snowfall and snow melt. The wording "liquid water storages, comprising mainly of soil moisture" is also a bit misleading. It is not really clear what is implicitly incorporated in the soil moisture reservoir in order to fully reproduce TWS (as mentioned in the third comment).*
*Güntner et al. 2007 provides a similar analysis based on WaterGAP. This would be an interesting point of comparison since they indicate a contribution for IAV of 33% snow, 27% soil and 12% groundwater and 28% surface water (!) for cold climates (their table 5). I think this reference should be discussed and compared with your results.*

   ***AR:*** We thank the referee for highlighting this lack of clarity in the abstract. We have revised the abstract accordingly. Further, as the referee suggested, we have included a comparison with the results of Güntner et al. 2007 in the discussion of the revised manuscript.

**\*Minor Comments\***

*5) MODELLING FRAMEWORK*

***RC:*** *Your work is very new and promising in the sense that multiple remote sensing or machine-learning observation-derived products are used simultaneously for calibrating a hydrological model. This is not easy to do and a research direction worth to explore. The overall modeling framework however still relies on a very standard land surface model structure. One missed opportunity may be to have used these observational datasets not only to calibrate model parameters, but also to identify functional relationships directly from the data (as opposed to fitting the parameters of a pre-defined equation to the data). Such research might be suggested as one possible future direction in the discussion. Finally, the paper does not emphasize on the added value of using remote sensing products to constrain the model (except for a lower RMSE*

*against observations, which is somewhat expected since other models were not calibrated with these observations). Could similar results have been obtained with the Earth2Observe ensemble? (especially on IAV?) If not, this would better show the merit and relevance of the presented approach.*

**AR:** The referee is right in that our modelling framework still relies on a standard land surface model structure in terms of which processes are included. We yet want to highlight that this modelling framework still allows for more flexibility in the responses because we do not strictly constrain the model parameters that are often fixed in land surface models. We agree that identifying functional relationships directly from the observations represents new challenges in modelling the Earth system, especially when the modelling community shifts towards a hyper-resolution modeling, for which the classical formulation at coarse resolutions might not be valid. With the current availability and inconsistencies in the observational data, we could not address the challenge in the current study.

As pointed out, use of remote sensing data is advantageous in constraining the model over much larger spatial domains than using site-level or discharge measurements and thus improves the confidence in model results. The improved confidence, also reflected in the presented better performance metrics is also an important merit compared to the EartH2Observe ensemble. Despite, the lower performance metrics, the results of the EartH2Observe ensemble are in general similar to our study.

As the referee suggested, we have highlighted the merits of using remote sensing data, as well as potential future research in identifying functional relationships directly from observations.

**6) CLARITY OF MODELLED LIQUID WATER AND MODELLED RUNOFF GENERATION**

**RC:** *Liquid water is explicitly modeled as soil moisture + runoff routing but also likely includes river storage, lakes and wetlands implicitly (e.g. large water holding capacity mentioned on page 12, lines 7 and 19). This could also be made a bit clearer already in the model description in order to avoid some confusion later. Using a snow/non-snow terminology would also help resolving this.*

*It could be made clearer that runoff is currently only generated from infiltration limitation (e.g. no baseflow in Eq. S10). Also mention that this is partially compensated by the recession time scale parameter that delays runoff generated on a specific day. Likely because the model is evaluated at monthly scale, this only has a limited impact on model performance and this model parameter is the least constrained by observations.*

**AR:** Thanks for pointing this out, we have adjusted the model description in the revised manuscript.

**7) DISTINCTION OF OBSERVATIONAL PRODUCTS**

**RC:** *Methods: You could make a better distinction between purely observational products, and observation-based upscaled products such as Tramontana et al. or Gudmundson et al. which also rely on the quality of the underlying forcing data.*

**AR:** We see that such a distinction could help to underline the dependencies and uncertainties in the observational products. However, for each data product, information on its derivation is included in the original manuscript in the description of the input data. Besides, the line between purely observation based and upscaled isn't always that clear, as e.g. GlobSnow is based on a snow model, satellite data and site measurements, and the GRACE estimates rely on several models for data correction as well.

**8) COVARIANCE OF SWE AND W**

**RC:** *Line 29-30, page 10: this assumption seems a bit dangerous given figure 6. Could you please document the degree to which this assumption is correct and if this might affect the results qualitatively (possibly in supplementary information)?*

*AR:* We agree with the referee that the potential implication of this assumption should be discussed. We therefore have included a short discussion on the effect of the covariances between SWE and W on TWS variability in the supplement of the revised manuscript.

**\*In-Text Comments\***

*9) **RC:** Line 29, page 2: and in addition, there can be no retrieval of SM in snow-covered or highly vegetated regions.*

    *AR:* added.

*10) **RC:** Line 3, page 4: for clarity, maybe you could add an introductory sentence indicating that this whole section is meant to give an overview of the model setup.*

    *AR:* added.

*11) **RC:** Line 10, page 4: E-RUN, based on E-OBS*

    *AR:* changed.

*12) FLUXCOM ET*

**RC:** *Line 10-14, page 7: I thought FLUXCOM was based on an ensemble of machine learning algorithms (e.g. not only random forest). Could you also briefly comment on the performance of FLUXCOM in snow regions and high latitudes? Any idea if FLUXCOM is already accounting for sublimation?*

    *AR:* The referee is right, FLUXCOM provides an ensemble of machine learning algorithms, but we only used the products from the random forest variant in this study. Even though FLUXCOM data have not been validated explicitly for snow-dominated regions, the cross validation of ET shows a good performance in most regions (Tramontana et al., 2016). In terms of sublimation processes, FLUXCOM conceptually includes sublimation processes as well, but the confidence in capturing such small fluxes is low due to lower signal to noise ratio in the underlying observations in FLUXNET sites. Therefore, we do not constrain modelled sublimation by FLUXCOM-based ET. We have clarified this in the revised manuscript.

    Tramontana, G., Jung, M., Camps-Valls, G., Ichii, K., Raduly, B., Reichstein, M., Schwalm, C. R., Arain, M. A., Cescatti, A., Kiely, G., Merbold, L., Serrano-Ortiz, P., Sickert, S., Wolf, S., and Papale, D.: Predicting carbon dioxide and energy fluxes across global FLUXNET sites with regression algorithms, Biogeosciences Discussions, 1-33, 10.5194/bg-2015-661, 2016.

*13) **RC:** Page 9: Is there any reference for this cost function?*

    *AR:* Unfortunately, there is no other reference for the cost function as it is applied in this paper.

*14) **RC:** Lines 17-20 page 9: I think this is indeed a very good idea!*

    *AR:* Thanks.

*15) **RC:** Line 22, page 9: can you indicate where these commonly reported values can be found? (It's also fine if you decide to assume 10%).*

    *AR:* We have assumed the 10 % and indicate this in the revised manuscript.

*16)* ***RC:*** *Line 8, page 10: typo*

   ***AR:*** changed.

*17) REDUCTION OF SNOW FALL*
***RC:*** *Line 22-23 page 11: interestingly however, this also contradicts Behrangi et al. 2017 for mountainous regions…*
*Behrangi, A., Gardner, A. S., Reager, J. T., & Fisher, J. B. (2017). Using GRACE to constrain precipitation amount over cold mountainous basins. Geophysical Research Letters, 44(1), 219-227.*

   ***AR:*** Yes, Behrangi et al. 2017 found winter time precipitation is likely underestimated by popular precipitation products, including GPCP, compared to GRACE. However, their study focusses on two basins of the Tibetan Plateau, a region of which only the most northern parts with relatively less elevation and heterogenetic topography are covered in our study domain. On contrary, Behrangi et al. (2016) and Swenson (2010), both showed an overestimation of precipitation by global precipitation products compared to purely observation based estimates in high-latitudes. Since high latitudes cover a larger fraction of the study domain, the reduction of snow fall by the calibrated global $p_{sf}$ parameter rather reflects these regions instead of characteristics valid for the Tibetan Plateau.

   Behrangi, A., Christensen, M., Richardson, M., Lebsock, M., Stephens, G., Huffman, G. J., Bolvin, D., Adler, R. F., Gardner, A., Lambrigtsen, B., and Fetzer, E.: Status of high-latitude precipitation estimates from observations and reanalyses, Journal of Geophysical Research: Atmospheres, 121, 4468-4486, 10.1002/2015jd024546, 2016.
   Swenson, S.: Assessing High-Latitude Winter Precipitation from Global Precipitation Analyses Using GRACE, Journal of Hydrometeorology, 11, 405-420, 10.1175/2009jhm1194.1, 2010

*18)* ***RC:*** *Line 15 page 12: maybe this rather small value is in relation with the relatively large soil water holding capacity.*

   ***AR:*** The reviewer is correct, the parameter value of the soil water holding capacity influences the parameter for evapotranspiration and runoff generation.

*19) DOMINANCE OF IAV IN NA, E AURASIA*
 ***RC:*** *Line 12-13 page 15: For instance, Humphrey et al. 2016 figure 6 shows that the central North America and Eastern Eurasia is rather dominated by IAV (which appears more difficult to model according to your figure 5).*
*Humphrey, V., Gudmundsson, L., & Seneviratne, S. I. (2016). Assessing global water storage variability from GRACE: Trends, seasonal cycle, subseasonal anomalies and extremes. Surveys in geophysics, 37(2), 357-395.*

   ***AR:*** We thank the reviewer for pointing to this reference. We have included the it in the revised manuscript.

*20) INACCURATE SENTENCE*
***RC:*** *Line 16 page 12: the sentence is inaccurate: a recession time scale of x days does not mean that only runoff of the preceeding x days contributes to "total runoff" (check Orth et al. 2013).*

   ***AR:*** Thank you very much for pointing this out! We have changed the sentence in the revised manuscript accordingly.

*21) GLOBAL UNIFORM PARAMETER VALUES*
***RC:*** *Line 13, page 13: maybe not necessary to say that these approaches are not commonly accepted as this might be a subjective statement in my opinion. The arguments you give just before (on overfitting) and the continental-scale focus of your study might be sufficient arguments. Another argument you could mention is that allowing locally varying parameters*

*would contaminate your conclusions: with locally dependent parameters, the differences in local-scale / large-scale contribution to IAV might due to the spatial dependency of parameters. But with your current setting, they can only be attributed to climate forcing. This is also why it makes a very clean experiment. This last point also calls for one caveat in the conclusion: your picture of the partitioning and scale-dependency of liquid versus snow might also change once you introduce spatial variability of the model parameters (e.g. snow melt factor might be very dependent on the vegetation cover, contrasting the responses of tundra versus boreal forests).*

*AR:* The referee is right, this statement seems to transport a quite subjective opinion, yet it's based on Beck et al. (2016) 'Due to the lack of a commonly accepted approach for parameter regionalization, hydrologic models typically applied at continental to global scales (hereafter called macroscale) rarely use regionalized parameters […]'. Therefore, we reformulated the sentence in the revised manuscript.

With his last comment, that the conclusions may change if we introduce spatial variability, the referee made a good point that is missing in our discussion. We have included this possible caveat when discussing the limitations of the approach.

Beck, H. E., Dijk, A. I. J. M. v., Roo, A. d., Miralles, D. G., McVicar, T. R., Schellekens, J., and Bruijnzeel, L. A.: Global-scale regionalization of hydrologic model parameters, Water Resources Research, 52, 3599-3622, 10.1002/2015WR018247, 2016.

*22)* *RC:* *Page 14, lines 3-6: essentially repeats page 13 line 10.*

*AR:* We have removed the repeated lines.

*23)* *TRUNCATION OF VALUES IN FIGURES*
*RC:* *Figure 3. If values were truncated (e.g. Fig3d) this should be indicated in the legend and in labels.*

*AR:* In the legend of figures in the revised manuscript, we indicate if values were truncated.

*24)* *RC:* *Line 1 page 19: TWSmod?*

*AR:* clarified.

*25)* *RC:* *Line 7 page 19: typo in earth2observe*

*AR:* changed.

*26)* *RC:* *Line 5 page 10: replace "grids" with "grid cells" idem on lines 5-6 page 20*

*AR:* changed.

*27)* *RC:* *Line 5 page 20: was the use of a subset mentioned also in the methods?*

*AR:* Thanks for pointing to this. The use of a subset of grid cells for model calibration is mentioned in the methods of the revised manuscript now.

*28)* *RC:* *Line 6 page 21: coincides*

*AR:* changed.

*29)* **RC:** *QUANTITAVE LABELS IN FIGURES SHOWING CR*

*Figure 7: It would be nice to add units to the colorbars (in addition to qualitative labels), same in Figure 8.*

    **AR:** We have added quantitative labels of CR in the revised manuscript.

*30)* **RC:** *Line 18 page 22: is "received" the adequate word?*

    **AR:** changed.

*31)* **RC:** *Line 7 page 23: frozen soil is not modelled*

    **AR:** removed.

*32) MISUNDERSTANDING OF SPATIAL VARIABILITY*

**RC:** *Line 11 page 25: On first read I could not follow since you cannot invoke geographic characteristics when you have spatially constant model parameters. The only source of spatial variability is in the model forcing. This is mentioned but only later on page 26 line 5, maybe you could reformulate this in a way that avoids such a misunderstanding.*

    **AR:** Thanks for pointing to this possible misunderstanding. We intended the sentence to refer to general relations, not to the modelled variables. In order to avoid such a misunderstanding, we have clarified the sentence in the revised manuscript.

*33) SPATIAL COHERENCE OF Rnet*

**RC:** *Line 7-9 page 26: note that ET is also influenced by Rnet which might also be less spatially coherent.*

    **AR:** The reviewer is right, Rn influences ET as well. However, as **Figure 7** suggests, inter-annual variations of net radiation in the study domain have a spatial coherence similar to temperature anomalies.

[Figure]

**Figure 7.** Proportion of total positive (grey) and negative (orange) covariances among grid cells for inter-annual variations of net radiation.

*34)* **RC:** *Line 11-12 page 27: and there can be no SM retrieval in snow-covered regions.*

    **AR:** Thanks, we have added this fact in the revised manuscript.

*35) **RC:** Line 15-18 page 27: solid/liquid phase transitions in soil moisture layers are another type of neglected effect relevant in the study domain as mentioned in the main comment.*

**AR:** Thanks, we have added this fact in the revised manuscript.

*36) **RC:** Line 10 page 28: the fact that snowpack anomalies are "erased" each summer and partially transferred to soil moisture through snow melt also largely explains this pattern. Hence, soil moisture also by construction allows for a longer memory than snowpack. This could be made more prominent in the discussion as well.*

**AR:** Thanks for highlighting this, we have added it to the revised manuscript.

*37) **RC:** Line 11-12 page 28: I would not qualify this as "diverging" since the sign is still the same (Figure 9). The non-snow storage only becomes less dominant when spatially integrated.*

**AR:** changed.

*38) **RC:** Line 23-25 page 28: adding this to the abstract would really explain better what you mean with the cryptic ending on line 32-34 page 1.*

**AR:** The reviewer made a good point. We have added the essence of this lines to the abstract.

**2. Anonymous Referee #2**

**RC:** *The main objective of this study is to analyse the spatial and temporal variability of snow pack and liquid water (mainly soil moisture) components at mid to high Northern latitudes and their respective contributions to total water storage (TWS) variations. To do so, a parsimonious hydrological model adapted to this purpose was first developed and calibrated using Earth Observations datasets, including TWS, snow water equivalent (SWE), evapotranspiration and gridded runoff. A comprehensive description of the model is provided in the Supplementary Material and a rather deep analysis of the calibration procedure is proposed. The authors also made a great effort in analyzing the performances of the model at different time scales (seasonal and interannual) and spatial scales (grid pixel and whole domain). Then the model is used over the 2000-2014 time period to evaluate the contribution of solid and liquid water components to TWS variations at these different spatio-temporal scales. Main conclusions are that TWS variations are mainly driven by snow dynamics at seasonal scales, while liquid water dominates TWS variations at interannual scales. Before concluding, the authors discuss some limitations of the method.*
*The paper is well written and organized. The conclusions are consistent with results presented all along the manuscript. The analysis of the calibration results (in terms of parameter values) is appreciable. Also appreciable is the comparison of the model to state-of-the-art global hydrological models from the eartH2Observe project, showing that despite its simplified structure, the current model performs reasonably well. I have only one major comment and some minor remarks and suggestions developed in the following.*

**AR:** We very much thank the anonymous referee #2 for the helpful comments and suggestions on our paper. Please find the author's response in the following.

**\*Major Comment\***

*1) NEED TO DEVELOP A NEW MODEL*

**RC:** *The need to develop a new model is not clearly stated, which is of prior importance since a large part of the paper is devoted to the presentation/validation of this model and the model outputs are used to draw the conclusions. Namely: - why not using existing models that show comparable performances and include more processes? -why not directly compare TWS and SWE from observations used here to calibrate the model? In that case, do the conclusions remain unchanged?*

    **AR:** The reviewer made a good point that developing a new model should be better justified. As the reviewer suggested, existing models could have been used for our study in principle. We chose to implement our own version of a parsimonious model of water cycle processes which shares the common conceptualization of existing models and represents a recombination of established process formulations for conceptual and methodological reasons. Conceptually, simulations of simple models are advantageous with regards to interpretation and understanding of the responses. Furthermore, we think it is also useful to confront results of a simple model informed by observations with more complex and more 'physically-based' ones to elucidate the added value of increased model complexity or possibly to understand where the model requires more comprehensiveness. From a methodological point of view, the model-data fusion approach requires that the underlying model is parsimonious with respect to a) identifiability of model parameters, and b) computational tractability as thousands of simulations need to be performed during the model optimization. Unfortunately, to our understanding, both considerations are hardly achievable using most of the existing models. Additionally, the design is tailored by the globally available data and kept simple possible to provide the opportunity to identify the effect of the inclusion of the different data sets.

To address this comment in the manuscript we have adjusted the introduction in the revised manuscript.

Regarding the second question, a direct comparison of TWS and SWE from observations is an interesting suggestion that we considered thoroughly. There are three obstacles with respect to the suggested analysis: 1) the SWE data from GlobSnow suffer from a saturation effect above SWE values of about 100mm. This causes that these systematic errors in the snow data directly propagate to and corrupts the inferred 'liquid' storage component if the difference to GRACE-TWS is calculated. 2) There are frequent gaps in the SWE data which can be either due to the absence of snow or missing data. The suggested analysis would therefore be biased to respective grid cells and times without gaps and could not yield a representative picture. 3) Besides errors in GlobSnow SWE that propagate to the inferred 'liquid' water storages, errors and uncertainties of the GRACE TWS transfer to the 'liquid' water storage as well. We concluded that a joint interpretation of GRACE-TWS and GlobSnow SWE within an appropriate model-data fusion approach as done in this manuscript is preferred.

Nevertheless, we performed our analysis using GlobSnow SWE and GRACE TWS, as the referee suggested. We calculated liquid water as the difference between GRACE TWS and GlobSnow SWE and then compared the model results using the same data points as available from the observations (Fig. 8-10). On the interannual scale, we obtain similar conclusions when directly using the observations and when using the model. For the mean seasonal cycle, the main pattern persists as well, yet conclusions differ in some regions that likely suffer from saturation in GlobSnow SWE (e.g. Kamchatka) or in regions where permafrost and wetlands play a role (e.g. East Siberia). As the latter are not observed by GlobSnow, their contribution to observed TWS is included in calculated W. Additionally, the magnitude in GRACE TWS anomalies is in general much larger than the magnitude of GlobSnow SWE, and thus the magnitude in W based on these observations is larger than the magnitude in modelled W. Therefore, the relative contribution to TWS variability based on observations is shifted towards a larger effect of liquid water storages as compared to the modelled results (Fig. 8).

**Contribution of SWEobs & Wobs to TWSobs MSC    Contribution of SWEmod & Wmod to TWSmod MSC**

[Figure]

**Figure 8.** Relative contribution based on CR of snow (SWE) and liquid water (W) storage anomalies to mean seasonal TWS anomalies based on observations (left) and based on the model considering the same data points (right).

**Contribution of SWEobs & Wobs to TWSobs IAV    Contribution of SWEmod & Wmod to TWSmod IAV**

[Figure]

**Figure 9.** Relative contribution based on CR of snow (SWE) and liquid water (W) storage anomalies to interannual TWS anomalies based on observations (left) and based on the model considering the same data points (right).

[Figure]

**Figure 10.** Relative contribution of snow (SWE) and liquid water (W) to TWS variability on different spatial and temporal scales based on observations (left) and based on the model considering the same data points (right).

**\*Minor Comments\***

*2)* ***RC:*** *P1L20: "…observed hydrological spatio-temporal patterns…"*

    ***AR:*** changed.

*3) USE OF SATELLITE DERIVED SOIL MOISTURE*

***RC:*** *P2L32: Some models explicitly simulate the upper soil layer using a multi-layer scheme (e.g., the ISBA land surface model, Decharme et al., 2011). In that case, satellite derived soil moisture can be compared to model outputs, and even assimilated with positive impacts on the model performances (Albergel et al., 2017).*

    ***AR:*** The authors are thankful for the reviewer's suggestion. We are considering the use of a multi-layer soil scheme with potential to assimilate satellite-derived soil moisture for future efforts, especially at the global scale, and mention this as outlook and potential improvement in the discussion of the revised manuscript.

*4) DELINEATION OF THE STUDY DOMAIN*

***RC:*** *P4L5: Which datasets are used to mask out such pixels?*

    ***AR:*** We thank the reviewer for pointing out that our manuscript was missing this critical information, which has been included the revised manuscript.

*5) ROUTING*

***RC:*** *2.2 Model description: if I understand correctly, incoming water from upstream grid cells are not accounted for. At the monthly time scale, I agree that this would be negligible at the pixel scale, but is it still true at the basin scale (e.g., the Ob river basin)?*

*AR:* As the reviewer pointed out, routing effects can be significant for large basins, especially in humid regions, which manifests in differences between surface runoff and river discharge at a given location. To address this, we do not use measured river discharge of large basins in our model-data fusion approach, but rather monthly runoff estimates for the European region at grid scale. Regarding the effect of river routing on TWS, e.g. Kim et al. (2009a) showed that the contribution of river storage to total TWS anomalies can be significant in the downstream regions of large basins. In northern high latitude catchments, this contribution is relatively smaller compared to continental tropical basins with large floodplains. Thus, we assume that although river storage is not explicitly represented in our model, the associated delay in surface runoff is sufficiently implicitly lumped into the delayed response of land runoff. Therefore, as the reviewer suggested, the effects of routing on the findings of this study can be expected to be small. We have clarified this in the revised manuscript.

Kim, H., Yeh, P. J. F., Oki, T., and Kanae, S.: Role of rivers in the seasonal variations of terrestrial water storage over global basins, Geophysical Research Letters, 36, doi:10.1029/2009GL039006, 2009.

*6) RC: P6L8: "…daily cumulated gridded precipitation…"*

*AR:* changed.

*7) RC: P6L11: "…that combines remotely-sensed precipitation…"*

*AR:* changed.

*8) GLOBSNOW DATA*
*RC: P7L2-4: Are these data [observed snow depth and radar data] assimilated into a snow model?*

*AR:* Yes, to our understanding the GlobSnow SWE processing applies a semi-empirical snow emission model and an assimilation scheme to produce maps of SWE estimates based on observations from passive microwave remote sensing and weather station observations (Luojus et al., 2010). We state this in the revised manuscript.

Luojus, K., Pulliainen, J., Takala, M., Lemmetyinen, J., Derksen, C., and Wang, L.: GlobSnow Snow Water Equivalent (SWE) Product Guide, ESA, 2010.

*9) MAPS OF TEMPORAL AVERAGE DATA UNCERTAINTIES*
*RC: 2.3 Input Data: Since EO uncertainties are an important aspect of the calibration process, I suggest the authors to add a figure showing maps of temporal averages of each uncertainty for each dataset. This could help interpreting the model performances as shown in Figures 3, S1 and S2.*

*AR:* This is a helpful comment for making the manuscript comprehensive. We include the maps of the temporal averages of the uncertainty of observed TWS, ET and Q that are used for model calibration in the supplement of the revised manuscript. As we apply a constant average uncertainty of 35 mm (as mentioned in line 26 page 9, original manuscript), an additional map is not included in the supplement.

*10) RC: P10L8: "…Therefore..."*

*AR:* changed.

*11) PARAGRAPH OF SPATIAL COHERENCE*

**RC:** *P11L4-9: This paragraph is unclear. Is it related to the smoothness of GRACE spatial patterns? In that sense, I think that for a better comparison with GRACE, modelled TWS should be first processed to remove high frequency spatial variability that is not observed by GRACE.*

**AR:** We see that the methodological paragraph on compensatory effects and spatial coherence was not clear enough. It is not related to the smoothness of GRACE spatial patterns but is meant to provide background information for the analyses to explain the different importance of TWS components to the total TWS across different spatial scales (local grid scale vs. spatially aggregated). We have clarified the paragraph in the revised manuscript.

*12) OVERESTIMATION OF GPCP*

**RC:** *P11L22-24: Is the overestimation found by Behrangi et al. (2016) and Swenson (2010) quantitatively comparable to this study?*

**AR:** The reviewer pointed to an interesting comparison that was missing in the original manuscript. Due to the mismatch in the spatial and temporal domain, it yet is difficult to quantitatively compare the results reported in Behrangi et al. (2016) and in Swenson (2010) with this study in a precise manner. However, Behrangi et al. (2016) showed that average high-latitude annual precipitation of GPCP is 20 % higher compared to other precipitation products, and Swenson (2010) state that the GPCP undercatch correction is too large, resulting in too much cold season accumulation. This suggests that reducing GPCP snow fall in our study by 33 % seems quantitatively comparable. We have included this statement in the revised manuscript.

Behrangi, A., Christensen, M., Richardson, M., Lebsock, M., Stephens, G., Huffman, G. J., Bolvin, D., Adler, R. F., Gardner, A., Lambrigtsen, B., and Fetzer, E.: Status of high-latitude precipitation estimates from observations and reanalyses, Journal of Geophysical Research: Atmospheres, 121, 4468-4486, 10.1002/2015jd024546, 2016.

Swenson, S.: Assessing High-Latitude Winter Precipitation from Global Precipitation Analyses Using GRACE, Journal of Hydrometeorology, 11, 405-420, 10.1175/2009jhm1194.1, 2010.

*13)* **RC:** *P11L26: "…if SWE > 80 mm (parameter snc) …"*

**AR:** changed.

*14) VALUES OF THE 4 COST TERMS*

**RC:** *3.1 Model optimization: It would be interesting to discuss the values of the four cost terms in Eq. (2) obtained with the optimized parameters.*

**AR:** We agree with the reviewer that the individual contributions to the total cost are a relevant methodological aspect. We have included a respective table in the supplement of the revised manuscript for completeness. To keep the manuscript concise, we don't add excessive discussion to this methodological detail, in particular since we present and discuss the evaluation of the model simulations against the individual data streams quite extensively in the manuscript.

*15) CORRELATION FOR SEASONAL VARIATIONS*

**RC:** *P14L12: Are "seasonal variations" equal to the "mean seasonal cycle"? We understand after (from the figures) that yes. In this case, very high correlation values are not really surprising. Bias and RMSE would be more suited.*

*AR:* The reviewer is correct, 'seasonal variations' are used synonymously to 'mean seasonal cycle'. This has been clarified in the revised manuscript. We will also follow the suggestion of the reviewer and include RMSE metrics of ET and Q in the supplement of the revised manuscript.

*16) REGIONS OF LARGE RMSE OF TWS*

*RC: Figure 3: It seems from figure 3(d) that large RMSEs are found in regions affected by the Postglacial Rebound (Eastern Canada and Scandinavia) and near coastlines (ocean signal contamination?).*

*AR:* Yes, the reviewer is right, large RMSEs tend to occur in regions affected by Postglacial Rebound and near coastlines where the signal potentially is contaminated by the ocean. These limitations and errors of the GRACE TWS estimates are referred to in line 6-8 page 15 and are stated in relation to high RMSE in line 10-11 page 15 (original manuscript):

"[…] Second, although GRACE TWS passed through various pre-processing steps, the models to account e.g. for postglacial rebound or leakage between neighbouring grid cells introduce their own uncertainties and do not remove the effects completely. […] This together is reflected in higher RMSE in arctic regions (e.g. surrounding the Hudson Bay), as well as in heterogeneous coastal and mountainous regions."

*17) NEGATIVE TWS ANOMALY IN 2003*

*RC: P19L5: Do the authors have any possible explanation of the large negative anomaly in 2003 and why it is not captured by the model?*

*AR:* This is a very interesting point, which we investigated further. If we isolate interannual variations by removing the trends in GRACE and modelled TWS the agreement with respect to 2003 gets substantially better, as indicated by higher correlation scores (Fig. 11). This suggests that the trend in GRACE TWS is to some extent either subject to observational issues or represents a process that is not captured by our and the earthH2Observe models, which don't reproduce the 2003 anomaly adequately either (see Fig. S6, revised supplement). In addition, there is a negative SWE anomaly of on average 5 mm (see Fig. 4a, manuscript) indicated in the GlobSnow data, that is not captured by our model, suggesting an issue with the precipitation forcing data. This not captured SWE anomaly appears to explain the remaining difference to the GRACE TWS anomaly in 2003 after detrending. The reason why this snow anomaly is not captured by the forcing remains unclear at this point – it persists when using the WFDEI forcing data set (Fig. 12). While the model reproduces the spatial pattern of the 2003 interannual TWS variability, the magnitude of observed TWS, especially in North America, is not captured by the forcing and thus by the model, either (Fig. 13).
We have added a paragraph to the revised manuscript on the discrepancy regarding the 2003 anomaly.

[Figure]

**Figure 11.** Comparison of the mean seasonal cycle and interannual variability of TWSobs (GRACE), TWSmod (forced with GPCP precipitation) and TWSmodwFDEI (forced with WFDEI rain and snow fall).

[Figure]

**Figure 12.** Mean seasonal cycle and interannual variability of original (orig) and detrended (DT) TWSobs (GRACE) and TWSmod (forced with GPCP precipitation).

[Figure]

**Figure 13.** Average IAV of winter 2002/2003 (December, January, February) of observed and modelled TWS.

[Figure]

**Figure 14.** Average IAV of winter 2002/2003 (December, January, February) of observed and modelled SWE.

[Figure]

**Figure 15.** Average IAV of winter 2002/2003 (December, January, February) of modelled rain fall and snow fall (based on GPCP forcing).

[Figure]

**Figure 16.** Average IAV of winter 2002/2003 (December, January, February) of WFDEI rain fall and snow fall data.

*18)* **RC:** *P23L4: The average value does not show that CR is positive over the entire domain.*

    *AR:* We agree with the reviewer here and add quantitative labels of CR to the respective figures in the revised manuscript.

*19)* **RC:** *P23L14: "…less variable at interannual time scale…"*

      *AR:* changed

REVISED MANUSCRIPT including marked changes
* * *

[revised manuscript text omitted]
 sub-grid heterogeneity of topography and land surface characteristics. except for the fractional snow cover used to estimate snow melt and sublimation.

With regards to model parameter, we apply a global uniform parameter set and do not regionalize the parameters according to spatially distributed physio-geographical characteristics. In contrast, most macro-scale hydrological models include spatially distributed soil properties to define parameters related to infiltration, soil water holding capacity and percolation, as well as vegetation types to assess the effects of different plant functional types on evapotranspiration and canopy storage (Sood and Smakhtin, 2015). In contrast, ourOur model only implicitly considers the effects of vegetation for example on ET, but not its spatial variability, as the associated impacts are included in the observational constraint. Spatial variability of model parameters might affect the relative contributions of different storage components to TWS variability at different spatial scales. However, the comparison with eartH2Observe models, which generally involve spatial heterogeneity in model parameters, suggests that the main conclusions remain unchanged. Additionally, However, we want to highlight that spatial distribution of model parameters depend on assumptions and some degree of simplification as well, and thus does not necessarily represent reality better thanimprove model performance compared to the a global uniform parameter set obtained from multiple observational data. Further, as we encountered issues with parameter equifinality, especially between modelled snow melt and sublimation, future efforts should include a stronger utilization of runoff data in the calibration and validation process. This would help to better constrain between water fluxes to the atmosphere and liquid water fluxes, that can contribute to the runoff.

Finally, though the implemented cost function explicitly accounts for the uncertainty of the calibration data, additional uncertainties of other input data, their processing and characteristics remain partly unaddressed.

**Conclusion**

In this study, we assessed the relative contributions of snow pack versus soil and retained water variations to the variability of total terrestrial water storage (TWS) for northern mid-to-high latitudes. To do so, we constrained a parsimonious hydrological model with multi-criteria calibration against multiple Earth observation data streams including TWS from GRACE satellites and snow pack estimates from GlobSnow. The optimized model showed considerably good agreement with observed patterns of hydrological fluxes and states, and was found to perform comparable or better than simulations from state-of-the-art macro-scale hydrological models. This underlines the potential of simple hydrological models tied to observational data streams as powerful tools to diagnose and understand large scale water cycle patterns. Further, it highlights the benefits of considering multiple, complementary data constraints to overcome their individual shortcomings.

Consistent with previous studies, we found that seasonal TWS variations are dominated by the development of snow pack during winter months in most places of the mid-to-high northern latitudes. In contrast to this seasonal pattern, our study reveals that not snow but anomalies in liquid water storages, mainly comprising soil moisture, drive inter-annual TWS

Kommentiert [TT48]: referee #1 / comment 34

Kommentiert [TT49]: referee #2 / comment 3

Kommentiert [TT50]: referee #1 / comment 1
referee #1 / comment 35

Kommentiert [TT51]: referee #2 / comment 5

[revised manuscript text omitted]

55  21-821-2017, 2017.

**Supplement**

5   **S1 Detailed model description and formulas**

The model consists of three components: (1) a snow component that simulates accumulation and ablation of snow, (2) a soil water component to calculate soil moisture, evapotranspiration and land runoff, and (3) a runoff component that derives total runoff. All modelled fluxes and states correspond to the spatio-temporal resolution of the forcing data, which in this study is a 1° x 1° latitude/longitude grid and daily time steps.

10   The following describes all implemented processes and equations in detail.

**S1.1 Snow Component**

Snow storage is implemented as a simple accumulation and melt approach, which further is extended by consideration of sublimation and fractional snow cover. The snow storage as described by the snow water equivalent SWE [mm] at time t [d] is calculated as mass balance:

$$SWE_t = SWE_{t-1} + SF_t - ETSub_t - M_t \tag{S1}$$

where $SWE_{t-1}$ [mm] is the snow water equivalent of the preceding time step which is increased by snowfall $SF_t$ [mm d$^{-1}$] and reduced by the amount of sublimation $ETSub_t$ [mm d$^{-1}$] and snow melt $M_t$ [mm d$^{-1}$].

All precipitation P [mm d$^{-1}$] is assumed to fall as snow at temperatures below 0 °C. Since precipitation estimates, especially

20   during the cold season, are known for biases due to substantial under-catch (Rudolf and Rubel, 2005;Seo et al., 2010), P is scaled using the parameter $p_{sf}$ to derive SF at time t:

$$SF_t = p_{sf} \cdot P_t \quad | \ T < 0°C \tag{S2}$$

In order to incorporate sub-grid variability, the fraction of the grid cell covered by snow is computed following the H-

25   TESSEL approach (Balsamo et al., 2009;ECMWF, 2014):

$$FSC_t = \min\left(\frac{SWE_{t-1}}{sn_c}, 1\right) \tag{S3}$$

with fractional snow cover FSC [-] at time t being linearly dependent on $SWE_{t-1}$ of the preceding time step and the parameter $sn_c$ [mm] being the minimum SWE that ensures complete snow coverage of the grid cell.

30   Further, snow melt M and sublimation ETSub are assumed to only emerge from snow covered area by using FSC as scaling factor in the calculation of these fluxes.

Snow melt M occurs when snow storage is present and temperature exceeds melting temperature. Based on the restricted degree-day radiation balance approach described by Kustas et al. (1994), melt M [mm d$^{-1}$] at time t depends on temperature

35   $T_t$ [°C] and net radiation $Rn_t$ [MJ m$^{-2}$ d$^{-1}$]:

$$M_t = (m_t \cdot T_t + m_r \cdot Rn_t) \cdot FSC_t \quad | \; T > 0°C \tag{S4}$$

where the degree-day factor $m_t$ [mm °C$^{-1}$] and the radiation factor $m_r$ [mm MJ$^{-1}$] control the melt rate.

The derivation of snow sublimation ETSub is adapted from the approach implemented in the GLEAM model. This technique is based on the Priestley and Taylor (1972) formula, which calculates evaporation rate as latent heat flux LE [MJ m$^{-2}$ d$^{-1}$] based on the available energy Rn [MJ m$^{-2}$ d$^{-1}$], ground heat flux G [MJ m$^{-2}$ d$^{-1}$]) and a dimensionless coefficient $sn_a$ that parameterizes evaporation-resistance. LE at time t is derived by

$$LE_t = \left( sn_a \cdot \frac{\Delta sn_t}{\Delta sn_t + \gamma sn_t} \cdot (Rn_t - G) \right) \cdot FSC_t \tag{S5}$$

with $\Delta sn_t$ being the slope of the temperature/saturated vapor pressure curve [kPa K$^{-1}$] and $\gamma sn_t$ representing the psychrometric constant [kPa K$^{-1}$]. Both, $\Delta sn$ and $\gamma sn$, are modified for snow covered areas according to Murphy and Koop (2005).

They calculate $\Delta sn_t$ as a function of $T_t$ [K] (Eq. (S6)), and $\gamma sn_t$ as a function of atmospheric pressure Pair [kPa], specific heat of air at constant pressure $c_p$ [MJ kg$^{-1}$ K$^{-1}$], the ratio molecular weight of water vapor/dry air MW and latent heat of sublimation of ice $\lambda sn$ [MJ kg$^{-1}$] (Eq. (S7))).

$$\Delta sn_t = \left( \frac{5723.265}{T_t^2} + \frac{3.53069}{T_t - 0{,}00728332} \right) \cdot e^{9.550426 - \frac{5723.265}{T_t} + 3.53068 \cdot \ln(T_t) - 0{,}00728332 \cdot T_t} \tag{S6}$$

$$\gamma sn_t = \frac{Pair \cdot c_p}{MA \cdot \lambda sn_t} \tag{S7}$$

In Eq.(S7), Pair is assumed to be time- and space-invariant with a uniform value of 101.3 kPa and $c_p = 0.001$ MJ kg$^{-1}$ K$^{-1}$. MA is a constant of 0.622 and $\lambda sn$ is defined by Murphy and Koop (2005) as a function of $T_t$ [K]. With a molecular mass of water of 18.01528 g mol$^{-1}$, $\lambda sn$ can be calculated as:

$$\lambda sn_t = \left( 46782.5 + 35.8925 \cdot T_t - 0.07414 \cdot T_t^2 + 541.5 \cdot e^{-\left( \frac{T_t}{123.75} \right)^2} \right) \cdot \frac{0.001}{18.01528} \tag{S8}$$

Since snow-covered ecosystems can be assumed to be unstressed due to the sufficient availability of water, LE corresponds to actual sublimation ETSub (Miralles et al., 2011). And ETSub [mm d$^{-1}$] can be converted from LE through division by $\lambda sn$:

$$ETSub_t = \frac{LE_t}{\lambda sn_t} \tag{S9}$$

Altogether, the model calculates ETSub as a function of $T_t$, $Rn_t$, Pair, G, $sn_a$ and $FSC_t$. While $T_t$, $Rn_t$ and $FSC_t$ are variable in time and space and depend on input data, the approach postulates constant Pair = 101.3 kPa and G = 0 MJ m$^{-2}$ d$^{-1}$.

**S1.2 Soil component**

The central part of the model is the soil water component, which distributes input from rain fall and snow melt to soil water storage SM [mm], actual evapotranspiration ET [mm d$^{-1}$] and land runoff Qs [mm d$^{-1}$].

Like snow, the calculation of soil water storage as represented by soil moisture SM [mm] at time t follows the mass balance

$$SM_t = SM_{t-1} + In_t - ET_t \tag{S10}$$

with $SM_{t-1}$ [mm] being the soil moisture of the preceding time step which is increased by infiltration $In_t$ [mm d$^{-1}$] and reduced by actual evapotranspiration $ET_t$ [mm d$^{-1}$].

On the one hand, the amount of infiltration In [mm d$^{-1}$] depends on the possible inflow IW [mm d$^{-1}$], which is the sum of rain fall RF (precipitation P if T ≥ 0°C) and snow melt M at time t:

$$IW_t = RF_t + M_t \tag{S11}$$

10 On the other hand, a part of IW may not infiltrate due to current soil moisture conditions but contribute to (direct) land runoff Qs [mm d$^{-1}$]. To estimate the partitioning of IW into SM and Qs, Qs at time t is calculated after Bergström (1995) as:

$$Qs_t = IW_t \cdot \left(\frac{SM_{t-1}}{s\_max}\right)^{S_{exp}} \tag{S12}$$

In Eq. (S12) $Qs_t$ depends on the inflow $IW_t$, the runoff coefficient $s_{exp}$ and the actual soil moisture $SM_{t-1}$ compared to its
15 maximum water holding capacity $s_{max}$. Thus, no land runoff occurs if the soil water storage is empty and all IW is allocated to land runoff if the soil is completely saturated. Between these points, $s_{exp}$ determines the amount of inflow that converts to Qs. While low values of $s_{exp}$ lead to a high amount of Qs even if the soil moisture deficit is low (e.g. low SM/$s_{max}$ ratio), higher values of $s_{exp}$ increase the proportion of IW that infiltrates.
Infiltration In at time t is derived in accordance to the law of conservation of mass as:

$$In_t = IW_t - Qs_t \tag{S13}$$

Potential evapotranspiration potET [mm d$^{-1}$] at time t is derived from net radiation Rn [MJ m$^{-2}$ d$^{-1}$] and air temperature T [°C] according to the Priestley-Taylor formula (Priestley and Taylor, 1972), where $et_a$ is the alpha coefficient:

$$potET_t = et_a \cdot \left(\frac{\Delta_t}{\Delta_t + \gamma_t} \cdot \frac{Rn_t}{\lambda_t}\right) \tag{S14}$$

where $\Delta_t$ is the slope of the temperature/saturated vapor pressure curve [kPa K$^{-1}$], $\lambda_t$ the latent heat of vaporization [MJ kg$^{-1}$] and $\gamma_t$ the psychrometric constant [kPa K$^{-1}$].
The slope of the saturated vapor pressure curve $\Delta_t$, as well as the latent heat of vaporization $\lambda_t$ are functions of T at time t:

$$\Delta_t = \frac{4098 \cdot 0.611 \cdot e^{\frac{17.27 \cdot T_t}{T_t + 237.3}}}{(T_t \cdot 237.3)^2} \tag{S15}$$

$$\lambda_t = 2.501 - (2.361 \cdot 10^{-3}) \cdot T_t \tag{S16}$$

Analogue to Eq. (S7), $\gamma_t$ depends on a constant atmospheric pressure Pair of 101.3 kPa, the specific heat of air at constant pressure $c_p$ [MJ kg$^{-1}$ K$^{-1}$], the constant MA and the latent heat of vaporization $\lambda_t$:

$$\gamma_t = \frac{Pair \cdot c_p}{MA \cdot \lambda_t} \tag{S17}$$

In order to avoid complete depletion of the soil water storage and to account for cohesion of water in the soil matrix, only a fraction of soil moisture after infiltration is assumed to be available for evapotranspiration. We express the sensitivity of

5   evapotranspiration to available water similar to Teuling et al. (2006) by the parameter $et_{sup}$. Thus, $et_{sup}$ determines the portion of the sum of infiltration $In_t$ [mm d$^{-1}$] and soil moisture $SM_{t-1}$ [mm], that represents evapotranspiration supply supET [mm d$^{-1}$] at time t:

$$supET_t = et_{sup} \cdot (SM_{t-1} + In_t) \tag{S18}$$

10  Finally, actual evapotranspiration ET [mm d$^{-1}$] at time t is derived by comparing $potET_t$ [mm d$^{-1}$] and $supET_t$ [mm d$^{-1}$]:

$$actET_t = min(potET_t, supET_t) \tag{S19}$$

**S1.3 Runoff component**

As total runoff comprises fast direct runoff as well as delayed interflow and base flow, it's appropriate to consider retardation (Orth et al., 2013). Accordingly, total runoff Q [mm d$^{-1}$] at time t results from the accumulated effects of all land

15  runoff Qs [mm d$^{-1}$] generated during the preceding 60 time steps:

$$Q_t = \sum_{i=0}^{60} Qs_{t-i} \cdot \underbrace{\left[ e^{-\frac{i}{q_t}} - e^{-\frac{i+1}{q_t}} \right]}_{\text{delay component}} \tag{S20}$$

where the recession time scale $q_t$ [d] determines how quickly land runoff is transformed into streamflow. In theory, an

20  infinite number of time steps would be necessary to ensure that all generated Qs is transformed into Q. However, the arbitrary number of 60 days allows accounting for > 99 % of Qs (Orth et al., 2013), as long as $q_t$ is below 13 days. To allow longer recession times when calibrating the model and still account for > 99 % of Qs within the 60 days-window, the delay component of Eq. (S20) is scaled with its sum.

Introducing temporal delay leads to retention of a portion of Qs, and thus to an additional, temporal storage of retained water

25  RW [mm]. The change of retained water storage ΔRW [mm d$^{-1}$] at time t can be inferred using the water balance:

$$0 = P_t - actET_t - Q_t + \Delta TWS_t \tag{S21}$$

with the change of total water storage ΔTWS [mm d$^{-1}$] resulting from

$$\Delta TWS = (SWE_t - SWE_{t-1}) + (SM_t - SM_{t-1}) + W_t \tag{S22}$$

so that solving Eq. (S21) and Eq. (S22)

$$\Delta RW_t = actET_t + Q_t - P_t - (SWE_t - SWE_{t-1}) - (SM_t - SM_{t-1}) \tag{S23}$$

The amount of retained water RW [mm] at time t then results from

$$RW_t = RW_{t-1} + \Delta RW_t \tag{S24}$$

Finally, the integrated terrestrial water storage TWS [mm] at time t represents the sum of all storage components:

$$TWS_t = SWE_t + SM_t + RW_t \tag{S25}$$

**S2 Uncertainty of the observational constraints**

Maps of the temporal average uncertainties of observed TWS, ET and Q that are used for model calibration are shown in **Figure S1**. For observed SWE a constant average uncertainty of 35 mm is applied.

[Figure]

**Figure S1.** Mean uncertainty of monthly TWSobs [mm], and of the mean seasonal cycle of ETobs [mm d⁻¹] and Qobs [mm d⁻¹] used for model calibration. Values are truncated to 50 mm resp. 10 mm.

**S3 Cost terms**

Table S1 shows the cost terms achieved with the default and the optimized parameter set. Compared to the default parameter values, total costs clearly improve after calibration. The shown optimized values represent a weighted Nash-Sutcliff efficiency of 0.37 (TWS), 0.44 (SWE), 0.57 (Q) and 0.80 (ET) (weighted Nash-Sutcliff = 1 – cost value).

**Table S1.** Cost values obtained with the default and the optimized model parameters using Eq. (1).

| parameter values | TWS | SWE | ET | Q | total |
|---|---|---|---|---|---|
| default | 0.84 | 0.54 | 0.15 | 1.00 | 2.55 |
| optimized | 0.63 | 0.56 | 0.20 | 0.43 | 1.82 |

Kommentiert [TT53]: referee #2 / comment 9

Kommentiert [TT54]: referee #1 / comment 23

Kommentiert [TT55]: referee #2 / comment 14

**S4 Model performance regarding evapotranspiration and runoff**

[Figure]

**Figure S2.** Spatially averaged mean seasonal cycle (MSC) of the period 2002–2012 and inter-annual variability (IAV, difference between monthly values and the MSC) for ETmod and FLUXCOM based ETobs.

[Figure]

**Figure S3.** Spatially averaged mean seasonal cycle (MSC) of the period 2002–2012 and inter-annual variability (IAV, difference between monthly values and the MSC) for Qmod and EU-grid runoff Qobs. Qmod$_{consistent}$ solely considers grid cells that coincide with Qobs, while Qmod$_{all}$ is based on modelled runoff for all grids of the study domain.

[Figure]

**Figure S4.** RMSE of the mean seasonal cycle of simulated and observed a) ET [mm month$^{-1}$] and b) Q [mm month$^{-1}$]. RMSE values have been truncated to the range 0–30 (a) resp. 0-50 (b).

**Kommentiert [TT56]:** referee #2 / comment 15
referee #1 / comment 23

5 **S5 Phase shift in mean seasonal TWS**

[Figure]

**Figure S5.** Grid wise phase lag [months] between mean seasonal TWSobs and TWSmod. Negative values indicate preceding of the model compared to GRACE TWS.

**S6 Comparison with eartH2Observe models**

[Figure]

**Figure S6.** Comparison of spatially averaged observed (obs) **a)** SWE (GlobSnow) and **b)** TWS (GRACE) to simulations of this study (mod) and eartH2Observe models (incl. ensemble mean) in terms of average mean seasonal cycle (MSC) and inter-annual variability (IAV). MSC is calculated for the period 2002–2012, and IAV represents the difference of monthly values from the MSC. Only data points consistent between all models and the respective observational data are considered.

[Figure]

**Figure S7.** RMSE for the spatially averaged SWE (left) and TWS (right) achieved by our model compared to the model ensemble of eartH2Observe models and the ensemble mean across temporal scales.

[Figure]

**Figure S8.** Comparison of a) RMSE and b) Pearson correlation r for monthly SWE and TWS time series simulated with the eartH2Observe models, the model ensemble mean (model mean) and by our model (mod).

**S7 Uncertainty due to forcing and calibration data**

**S7.1 Comparison to WFDEI precipitation forcing**

To assess the uncertainty in TWSmod and SWEmod that emerges from the choice of precipitation forcing, we calibrated our model in the same manner as before, yet used rain fall and snow fall estimates from the reanalysis based WFDEI product (Weedon et al. 2014) instead of GPCP-1DD precipitation data. Since precipitation is likely the most uncertain input data (Herold et al. 2015, Schellekens et al. 2017), we did not change the temperature and net radiation data sets. The global meteorological WFDEI data for land area is generated by applying the Water and Global Change (WATCH) forcing data methodology to ERA-Interim reanalysis data (Dee et al. 2011). The advantage of the WFDEI product is that it already provides separate values for snow and rain fall, as diagnosed by the reanalysis (Weedon et al. 2014). Therefore, it is not necessary to partition precipitation based on a temperature threshold within the model. We rather applied the provided rain and snow fall estimates directly, and also desisted from scaling snow fall.

Regarding the MSC, we obtained similar model performance in terms of SWE and TWS for both, the spatially averaged dynamics (Figure S9, Figure S10) and the spatial pattern (not shown). Although the dynamics and thus the correlation coincidence, we obtain a higher amplitude in TWSmod when using WFDEI as forcing compared to the original TWSmod (and TWSobs). This higher amplitude relates to larger seasonal snow accumulation in SWEmod$_{\text{WFDEI}}$, because the scaling parameter for snow fall is not calibrated. In terms of IAV, the correlation between observation and WFDEI forced model is comparable for both, TWS and SWE. However, the key findings (**Figure S11**) remain the same as with GPCP precipitation forcing.

[Figure]

**Figure S9.** Comparison of the spatially averaged mean seasonal cycle (MSC) and inter-annual variability (IAV, difference between monthly values and the MSC) of observed SWE (SWEobs), modelled SWE (SWEmod), and modelled SWE based on WFDEI precipitation forcing (SWEmod$_{WFDEI}$). SWEmod consistent and SWEmod$_{WFDEI}$ consistent refers to modelled SWE considering only data points with available SWEobs, while SWEmod all and SWEmod$_{WFDEI}$ all incorporates all time steps for all grids of the study domain.

[Figure]

**Figure S10.** Comparison of the spatially averaged mean seasonal cycle (MSC) and inter-annual variability (IAV, difference between monthly values and the MSC) of observed TWS (TWSobs), modelled TWS (TWSmod), and modelled TWS based on WFDEI precipitation forcing (TWSmod$_{WFDEI}$). For IAV, TWSobs$_{monthly\ value}$ shows the original IAV of individual TWSobs months, while TWSobs, TWSmod and TWSmod$_{WFDEI}$ are smoothed using a 3-month average moving window filter. Pearson correlation r refers to the smoothed values. For the MSC no smoothing is applied.

**Kommentiert [TT57]:** referee #1 / comment 3

[Figure]

**Figure S11.** Relative contribution (based on CR (Eq.(3))) of snow (SWE) and liquid water (W) to TWS variability on different spatial (local grid scale, spatially integrated) and temporal (mean seasonal MSC, inter-annual IAV) scales when forced with WFDEI rain and snow fall. The boxplots represent the distribution of grid cell CR, with the dashed line marking the corresponding average. The star represents the CR calculated for the spatially integrated values.

**Kommentiert [TT58]:** referee #1 / comment 3

**S7.2 Comparison to other GRACE solutions**

In this study we used TWS estimates from the JPL mascon RL05 product for model calibration and evaluation (Watkins et al., 2015;Wiese, 2015). However, various GRACE solutions for TWS from different institutions and using different processing approaches exist. To assess the potential uncertainty resulting from the choice of TWS solution, we compared modelled TWS (mod) and the JPL mascon solution ($JPL_{masc}$) with other solutions based on different processing approaches. They include the mascon product from the Center of Space Research (CSR at the University of Texas) ($CSR_{masc}$) (Save et al., 2016), as well as three RL05 solutions based on spherical harmonics provided by JPL, CSR and GeoforschungsZentrum (GFZ) (Swenson and Wahr, 2006;Landerer and Swenson, 2012;Swenson, 2012). As recommended, we also considered the average of the latter three ($Avg_{JPL/CSR/GFZ}$). All TWS estimates were taken as anomalies to the respective time-mean of 2002–2012, and scaled with the provided gain factors (except for $CSR_{masc}$ that does not require scaling (Save et al., 2016). For comparison, we calculated the spatial average mean seasonal cycle (MSC) and inter-annual variability (IAV) across all grid cells of the study domain (**Figure S12**).

Thereby, we find that the spatial average MSC of all GRACE TWS estimates agrees in its dynamics, albeit minor differences in the solutions' amplitudes exist (by ±15 mm). This results in comparable correlation and RMSE with modelled TWS. As the signal itself is noisier on IAV scales, the GRACE solutions show broader variability for IAV than at MSC scales as well. However, the qualitative pattern between the solutions remains, and modelled TWS is not closer to one specific solution or another during the entire time period. Therefore, the uncertainty evolving from the choice of GRACE solution used for model calibration can be assumed to be minor.

[Figure]

**Figure S12.** Comparison of the spatially averaged mean seasonal cycle (MSC) and inter-annual variability (IAV, difference between monthly values and the MSC) of modelled TWS (mod) and observed TWS of different GRACE solutions.

**S8 Covariances between SWE and W**

**Kommentiert [TT59]:** referee #1 / comment 8

**Figure S13** and **Figure S14** compare the contribution of the combined SWE and W variances and the covariance of both storages to the total variance of the spatially aggregated TWSmod. On the interannual and spatially aggregated scale, 81 % of TWS variability is explained by the variances in SWE and W, suggesting that the covariance between SWE and W only has minor effect. This is underlined by high percentage of SWE and W variance on total TWSmod variance for all grids of the study domain (**Figure S14**). On mean seasonal scales, the majority of spatially aggregated TWS variability is still explained by variances in SWE and W, but the contribution of the covariance increases. This can be expected, as the seasonal variation of snow storage affects the subsequent availability of liquid water storages through the snowmelt process. At the local scale, though, the percentage of SWE and W variance on total TWSmod variance remains high in regions where the dominance of either snow or liquid water components are clear (Fig. 7 of the manuscript). In regions where covariances of two storage components is larger, the contribution of two storage components to TWS variability are similar resulting in a CR value of around 0. Therefore, we conclude that while the covariances of snow and liquid water can be remarkable on the seasonal scale over a large spatial domain, it does not affect or change the dominant components on the TWS.

[Figure]

**Figure S13.** Percentage of SWE and W variance on total TWSmod variance on mean seasonal (MSC) and interannual (IAV) scales.

[Figure]

**Figure S14.** Percentage composition of spatially aggregated TWSmod variance from the combined variances of SWE and W, and two times the covariance of SWE and W on mean seasonal (MSC) and interannual (IAV) scales.

**References Supplement**

Balsamo, G., Beljaars, A., Scipal, K., Viterbo, P., van den Hurk, B., Hirschi, M., and Betts, A. K.: A Revised Hydrology for the ECMWF Model: Verification from Field Site to Terrestrial Water Storage and Impact in the Integrated Forecast System, Journal of Hydrometeorology, 10, 623-643, 10.1175/2008jhm1068.1, 2009.

Bergström, S.: The HBV model, in: Computer models of watershed hydrology., edited by: Singh, V., 443-476, 1995.

ECMWF: IFS Documentation - Cy40r1: Part IV: Physical Processes, in, European Centre for Medium-Range Weather Forecasts, Reading, England, 2014.

Kustas, W. P., Rango, A., and Uijlenhoet, R.: A simple energy budget algorithm for the snowmelt runoff model, Water Resources Research, 30, 1515-1527, 1994.

Landerer, F. W., and Swenson, S. C.: Accuracy of scaled GRACE terrestrial water storage estimates, Water Resources Research, 48, 10.1029/2011wr011453, 2012.

Miralles, D., Holmes, T., De Jeu, R., Gash, J., Meesters, A., and Dolman, A.: Global land-surface evaporation estimated from satellite-based observations, Hydrology and Earth System Sciences, 15, 453-469, 2011.

Murphy, D., and Koop, T.: Review of the vapour pressures of ice and supercooled water for atmospheric applications, Quarterly Journal of the Royal Meteorological Society, 131, 1539-1565, 2005.

Orth, R., Koster, R. D., and Seneviratne, S. I.: Inferring soil moisture memory from streamflow observations using a simple water balance model, Journal of Hydrometeorology, 14, 1773-1790, 2013.

Priestley, C., and Taylor, R.: On the assessment of surface heat flux and evaporation using large-scale parameters, Monthly Weather Review, 100, 81-92, 1972.

Rudolf, B., and Rubel, F.: Global Precipitation, in: Observed Global Climate edited by: Hantel, M., Geophysics Springer, Berlin, 567, 2005.

Save, H., Bettadpur, S., and Tapley, B. D.: High-resolution CSR GRACE RL05 mascons, Journal of Geophysical Research: Solid Earth, 121, 7547-7569, 10.1002/2016jb013007, 2016.

Seo, K. W., Ryu, D., Kim, B. M., Waliser, D. E., Tian, B., and Eom, J.: GRACE and AMSR-E-based estimates of winter season solid precipitation accumulation in the Arctic drainage region, Journal of Geophysical Research: Atmospheres, 115, 2010.

Swenson, S., and Wahr, J.: Post-processing removal of correlated errors in GRACE data, Geophysical Research Letters, 33, 10.1029/2005gl025285, 2006.

Swenson, S.: GRACE monthly land water mass grids NETCDF RELEASE 5.0 CA, USA, http://dx.doi.org/10.5067/TELND-NC005, 2012.

Teuling, A. J., Seneviratne, S. I., Williams, C., and Troch, P. A.: Observed timescales of evapotranspiration response to soil moisture, Geophysical Research Letters, 33, 10.1029/2006gl028178, 2006.

Watkins, M. M., Wiese, D. N., Yuan, D. N., Boening, C., and Landerer, F. W.: Improved methods for observing Earth's time variable mass distribution with GRACE using spherical cap mascons, Journal of Geophysical Research, 120, 2648-2671, 2015.

Wiese, D. N.: GRACE monthly global water mass grids NETCDF RELEASE 5.0 Ver. 5.0 Mascon Ver. 2, PO.DAAC, CA, USA, 2015.